# Multi-frequency altimetry snow depth estimates over heterogeneous snow-covered Antarctic summer sea ice – Part I: C/S-, Ku-, and Ka-band airborne observations

Renée M. Fredensborg Hansen[1,2,3], Henriette Skourup[1], Eero Rinne[3], Arttu Jutila[4], Isobel R. Lawrence[5,6], Andrew Shepherd[6], Knut V. Høyland[2], Jilu Li[7], Fernando Rodriguez-Morales[7], Sebastian B. Simonsen[1], Jeremy Wilkinson[8], Gaelle Veyssiere[8], Donghui Yi[9], René Forsberg[1], and Taniâ G. D. Casal[10]

[1]Department of Geodesy and Earth Observation, The National Space Institute (DTU Space), The Technical University of Denmark (DTU), Kgs. Lyngby, Denmark
[2]Department of Civil and Environmental Engineering, Norwegian University of Science and Technology (NTNU), Trondheim, Norway
[3]Department of Arctic Geophysics, The University Centre in Svalbard (UNIS), Longyearbyen, Norway
[4]Finnish Meteorological Institute (FMI), Helsinki, Finland
[5]ESA – ESRIN, European Space Agency, Frascati, Rome, Italy
[6]Centre for Polar Observation and Modelling, Department of Geography and Environmental Science, Northumbria University, Newcastle, UK
[7]University of Kansas, Center for Remote Sensing and Integrated Systems (CReSIS), USA
[8]British Antarctic Survey, UK
[9]GST Inc., Laboratory for Satellite Altimetry, Center for Satellite Applications and Research, NOAA, College Park, MD, USA
[10]ESA – ESTEC, European Space Agency, Noordwijk, Netherlands

**Correspondence:** Renée M. Fredensborg Hansen (rmfha@dtu.dk)

**Abstract.** The recent alignment of CryoSat-2 to maximise orbital coincidence with the Ice, Cloud, and land Elevation Satellite-2 (ICESat-2) over the Southern Ocean and Antarctica in July 2022, known as the CryoSat-2 and ICESat-2 Resonance Campaign (CRYO2ICE), provided an opportunity to validate these satellites over land and sea ice. This was achieved through a simultaneous airborne campaign which underflew the near-coincident CryoSat-2 and ICESat-2 orbits in December 2022 and carried, amongst other instrumentation, Ka-, Ku-, C/S-band radars and a scanning near-infrared lidar. This campaign resulted in the first multi-frequency radar evaluation of snow penetration over sea ice along near-coincident orbits. The airborne observations (at footprints of 5 m) revealed limited penetration of the snowpack at both Ka- and Ku-band, with the primary scattering occurring either at the air-snow interface or inside the snowpack for both frequencies. On average, the Ka- and Ku-band scattering interface was 0.2 to 0.3 m above that for C/S-band's primary scatter, where the average snow depth using C/S-band reached around $0.5 \pm 0.05$ m depending on re-trackers and combinations used. Interestingly, when the primary peak in the received signal occurs within the snowpack or at the air-snow interface, some scatter contributions are still present from the sea-ice interface at the Ku-band. This suggests a potential for snow depth to be derived from Ku-band signals alone by co-identifying these respective peaks in the waveform. Furthermore, it contradicts the assumption of a single scattering interface primarily contributing to the backscatter at Ku-band (and, to some extent, Ka-band) on airborne scales. The validity of this assumption needs further evaluation using former campaigns covering different sea ice conditions and seasons. With the unique combina-

tion of sensors and methods evaluated here, a shortcoming is the limited validation that can take place without strategically placed coincident in situ efforts. We call for coincident field initiatives as part of future validation campaigns considering the observational capabilities of air- and spaceborne sensors when deciding on appropriate sampling strategies.

## 1 Introduction

Snow on sea ice plays a crucial role in the climate system of the polar regions with its insulating properties and high albedo, regulating sea ice growth and melt (Webster et al., 2018; Sturm et al., 2002). Beyond climatic importance and influence on the sea-ice-albedo–feedback, snow depth plays a key role in the retrieval of sea ice thickness from satellite altimetry, as snow loading must be accounted for when assuming hydrostatic equilibrium. Hence, accurate and timely large-scale estimates of snow depth on sea ice are crucial. In particular, with relatively warm air temperatures even during freeze-up, strong winds,

and heavy precipitation, the Antarctic sea ice is covered by an exceptionally heterogeneous snow cover (Massom et al., 2001). These conditions result in a wide variety of snow metamorphism occurring, complicating signatures observed using remote sensing methods. In particular, warm temperatures and variable atmospheric conditions can lead to, e.g., large snow grain sizes, internal ice layers, or ensure the presence of liquid water within the snowpack (Webster et al., 2018). Such conditions strongly interfere with the electromagnetic radiation from satellite observations. Coupled with a predominantly seasonal and

relatively thin ice cover (Worby et al., 2008), the ice floes tend to depress below the water level due to the excessive loading of the snowpack, giving rise to snow-ice formation once flooded (Arndt et al., 2024). Finally, percolation of meltwater to the snow-ice interface during the austral summer can initiate the formation of superimposed ice (Arndt et al., 2021). Combined with the absence of strong surface melt, these conditions permit the survival of a year-round, highly complex, and diversified snow cover (Arndt and Paul, 2018).

Snow depth has been derived from altimeters by using a measure of the air-snow (a-s) interface and the snow-ice (s-i) interface (Kwok and Maksym, 2014) either from coincident surface elevations, range estimates, or freeboards (height of ice or ice + snow relative to the water level, depending on the frequency of the altimeter). Commonly, the a-s interface is identified using laser observations that reflect at (or very close to) the snow surface (Kacimi and Kwok, 2022, 2020), where the freeboard observations computed from lasers are referred to as the total freeboard (snow + ice). Ka-band altimeters are also usually

assumed to have negligible penetration into the snow cover and are therefore often used as an estimate of a-s interface as well. Estimating the s-i interface has commonly been achieved by using Ku-band radars, where laboratory experiments have shown that Ku-band signals can penetrate to the s-i interface under cold and dry conditions (Beaven et al., 1995). Based on these experiments, Ku-band signals are often assumed to penetrate the snow cover, providing radar freeboards which are directly converted to sea ice freeboard after accounting for the slower wave propagation speed (e.g., Hendricks, 2022; Rinne and Hendricks, 2023; Mallett et al., 2020). Several studies (e.g., Nab et al., 2023; Willatt et al., 2023, 2010, 2011; De Rijke-

Thomas et al., 2023; Armitage and Ridout, 2015; Rösel et al., 2021; King et al., 2018; Nandan et al., 2017, 2020, 2023) have disputed this assumption based on comparisons using ground-based, airborne, and spaceborne observations, arguing that various interactions between the atmosphere, ice, and snow, such as snow metamorphism, redistribution, brine wicking, or

flooding, can significantly limit the penetration of Ku-band waves. Nonetheless, this dual-frequency approach remains one of the few Earth observation methods to derive snow depth over sea ice, and it is currently one of the driving factors and main objectives of the future dual-frequency polar radar altimetry mission, Copernicus Polar Ice and Snow Topography Altimeter (CRISTAL), planned for launch in 2027/2028 (Kern et al., 2020). For further investigation of the dual-frequency approach, ESA changed the orbit of their polar Ku-band radar altimetry mission, CryoSat-2, in July 2020 to align periodically with NASA's polar laser altimetry mission, the Ice, Cloud and land Elevation Satellite-2 (ICESat-2) for the Northern Hemisphere. This alignment, known as the CRYO2ICE Resonance Campaign, was adjusted again in July 2022 to maximise orbits in the Southern Hemisphere. So far, near-coincident laser and radar observations have been collected over both hemispheres during the past four years with orbit maximisation efforts enabling the evaluation of snow depth estimates along-track across both hemispheres for at least two consecutive years (Fredensborg Hansen et al., 2024).

To the best of our knowledge, no studies have fully evaluated the application of airborne multi-frequency altimetry for snow depth retrieval using lidar, Ka-, Ku-band and snow radars (with different ranges of frequencies having been flown, but commonly covering C/S-band and more recently, also Ku-band) simultaneously in either the Arctic or the Antarctic since there until now has not been a campaign employing the full suite of instruments available. Most airborne studies have so far focused on (a) the Arctic snow depth using snow radars or combinations of snow radar and lidar (e.g., Kurtz and Farrell, 2011; Newman et al., 2014; Rösel et al., 2021); (b) Ku-band radars combined with either ground-based reference observations or airborne lidars (e.g., Willatt et al., 2011; De Rijke-Thomas et al., 2023; King et al., 2018); or, (c) deriving snow depth from snow radar, Ka- and/or Ku-band for the purpose of validating satellite observations (e.g., Garnier et al., 2021; Armitage and Ridout, 2015). Furthermore, while a variety of snow radar retrieval algorithms are available for Arctic applications (see e.g., Kwok et al., 2017, for an inter-comparison of snow radar retrieval algorithms applied to spring conditions), only a few studies have computed airborne snow depth and evaluated their applicability over Antarctic sea ice cover (e.g., Fons and Kurtz, 2019; Kwok and Kacimi, 2018; Kwok and Maksym, 2014; Galin et al., 2012; Panzer et al., 2013). Here, Fons and Kurtz (2019) presented some of the first airborne Ku-band results, showing that the main scattering horizon occurs closer to or at the a-s interface, indicating that the Ku-band scattering is also affected by the a-s interface.

Most observations acquired with either microwave snow radar sounders, Ka/Ku-band radars or lidars from airborne survey campaigns have been conducted through NASA's Operation IceBridge (OIB, 2009–2019), ESA's CryoSat Validation Experiment (CryoVEx, 2001–onwards) or Alfred Wegener Institute's (AWI) IceBird (2009–onwards). For sea ice, studies using NASA OIB have primarily relied on observations from a combination of lidar, snow radar, and optical sensors, which limits the potential for studies on Ka- and Ku-band signal penetration. 2009, 2010, 2011 and 2012–2016 campaigns carried several versions of a dedicated Ku-band radar at different frequency ranges (MacGregor et al., 2021), however, few studies utilise this data; e.g., De Rijke-Thomas et al. (2023) used the 2016 Arctic campaign over landfast ice for dedicated analysis of airborne data, and Landy et al. (2021) for evaluation of estimated sea surface height anomalies and radar freeboards using an optimised decorrelation framework using 2011, 2012, and 2014 campaigns. During the Arctic 2015 spring campaign, a Ka-band radar demonstrator was flown (MacGregor et al., 2021); however, to our knowledge, this data has not been further evaluated over sea ice. Lastly, campaigns after 2017 carried a combined 2–18 GHz snow radar (MacGregor et al., 2021; Rodriguez-Morales

et al., 2020), which also covers the Ku-band frequency range, but sub-banding to evaluate the data at their specific bands at a
cost of range resolution has not been fully explored. By analysing the radar returns over discrete sub-bands, Yan et al. (2017)
demonstrated that the snow-ice interface appeared stronger over the 2–8 GHz sub-band versus the 12–18 GHz band, albeit at
the expense of a degradation in vertical resolution with respect to the full 2–18 GHz. A difference at the air-snow interface was
observed due to different surface roughness impacting differently depending on frequency. Furthermore, the above study noted
that the interfaces were ambiguous for snow depths of less than 0.1 m, a direct effect of the degraded range resolution.

CryoVEx originally acquired Ku-band and airborne laser scanner observations over both Arctic and Antarctic land and sea
ice to validate CryoSat-2, using the CryoSat-2 airborne simulator Airborne Synthetic aperture radar (SAR)/Interferometric
Radar System (ASIRAS). Based on the results of using SARAL/AltiKa (French-Indian Ka-band spaceborne pulse-limited
radar altimeter in orbit since 2014) in combination with CryoSat-2 for snow depth retrieval (Guerreiro et al., 2016; Armitage
and Ridout, 2015), efforts were made to include an airborne Ka-band sensor on the campaigns. Since spring 2017, CryoVEx
has carried combinations of Ka- and Ku-band originally as separate instruments, with ASIRAS and KAREN, and later through
the combined Center for Remote Sensing and Integrated Systems (CReSIS) Ka/Ku-band radar (flown on the CryoVEx 2019
summer and Cryo2IceEx 2022 spring campaigns, where both radars provide separate observations at their dedicated frequency
bands). This instrumentation palette provides observations from lidars, Ka-, and Ku-band along the same orbit, the only cam-
paigns to fully employ and provide Ka-band airborne observations, from which both Ka- and Ku-band penetration can be
further evaluated. Finally, the CryoVEx in collaboration with the Natural Environment Research Council (NERC) Drivers and
Effects of Fluctuations in sea Ice in the ANTarctic (DEFIANT) project completed an airborne campaign over Antarctic land
and sea ice in December 2022 which under-flew a CRYO2ICE orbit (Fig. 1), a campaign that for the first time carried the full
instrument suite including a snow radar, Ka- and Ku-band radars, and a scanning infrared lidar.

The AWI IceBird campaigns have been flying routinely in the Arctic since 2009, currently aiming for campaigns twice a
year, i.e. during the Arctic sea-ice winter maximum in April and the summer minimum in August (Jutila et al., 2022a). The
payload includes an airborne electromagnetic induction sounding sensor for total (snow + ice) thickness observations, along
with a lidar and a CReSIS snow radar (2–18 GHz, since 2017) to derive the total freeboard and the snow depth, respectively.
Currently, a method to derive snow depth from this particular snow radar has been developed; however only been tested for the
Arctic (Jutila et al., 2022b). One Antarctic campaign with a sea-ice component has been conducted, but further analysis of the
applicability of the methodology is pending.

With this two-part study, we bridge several gaps. In Part I, we evaluate microwave penetration into Antarctic snow cover
at Ka- and Ku-band and the applicability of a C/S-band snow radar, with comparison to lidar airborne observations. Here, we
present the first in-depth evaluation of airborne coincident Ka-, Ku-, snow radar (C/S-band) and lidar observations over sea ice.
We note that the study is limited by the logistical challenges of airborne campaigns, and as such, the observations were acquired
during the early Antarctic summer season. Hence, the physical conditions of the snow and ice pack on Antarctic sea ice likely
limit the full evaluation of the various retrieval algorithms and data available for all surfaces, sea ice and snow conditions. From
the evaluation of the airborne data, we then compare airborne snow depth estimates derived based on traditional hypotheses
of penetration with near-coincident spaceborne radar (CryoSat-2) and laser (ICESat-2) orbits along a dedicated CRYO2ICE

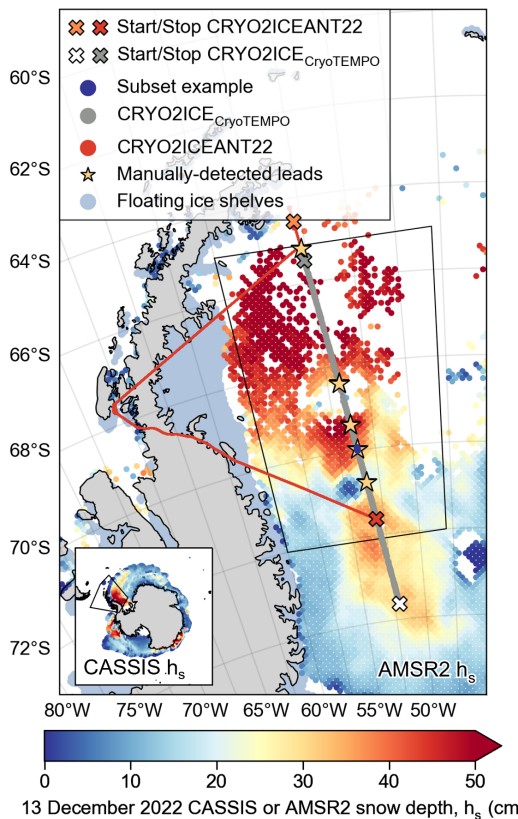

**Figure 1.** Cryo2IceEx and NERC DEFIANT 2022 (CRYO2ICEANT22) campaign with under-flight on 13 December 2022 along CryoSat-2 and ICESat-2 (CRYO2ICE) orbit with the location of subset example used throughout the study and location of manually-detected leads for offset calibration overlaid the AMSR2 snow depth product (for the main figure) and the CASSIS (Centre for Polar Observation and Modelling (CPOM) Antarctic Snow on Sea Ice Simulation) product (for inset), see data availability section for reference to data products (which are further utilised in Part II). Note the CryoTEMPO product refers to a specific processing chain used for CryoSat-2 observations, which are further evaluated in Part II. The black outline in the inset denotes the map extent shown as the main figure, black outline in the main figure denotes the area of interest for ERA5 precipitation and temperature calculations (see also Fig. 2). Floating ice shelves are provided in the NSIDC-0780 Antarctic regional mask data product (Meier and Stewart, 2023) at 6.25 km (using the NASA classification).

orbit, which is currently the only CRYO2ICE validation under-flight carried out with a full suite of instruments able to evaluate

penetration into snow. This is presented in Part II.

**Table 1.** Radar parameters. If only one value per parameter is provided per row, the same value stands for all radars. Values in parenthesis denote average and standard deviation.

| | C/S-band | Ku-band | Ka-band |
|---|---|---|---|
| Frequency-band | 2–8 GHz | 12–18 GHz | 32–38 GHz |
| Wavelength, $\lambda$ | 15–4 cm | 2.5–1.6 cm | 0.9–0.8 cm |
| Center frequency, $f_c$ | 5 GHz | 15 GHz | 35 GHz |
| Center wavelength, $\lambda_c$ | 6 cm | 2 cm | 0.8 cm |
| Pulse length | | 250 $\mu$s | |
| Pulse-repetition-frequency, PRF | | 3.125 kHz | |
| Bandwidth, $B$ | | 6 GHz | |
| Sampling frequency | | 125 MHz | |
| Antenna beam width[a] (cross- $\times$ along-track), $\beta$ | 2 GHz: $84° \times 76°$ <br> 8 GHz: $31° \times 31°$ | 12 GHz: $21° \times 22°$ <br> 18 GHz: $15° \times 15°$ | $17° \times 18°$ |
| Range resolution (Eq. A1) | | 0.04 m in free space (full bandwidth)[b] | |
| Nominal altitude, $H$ | | $\sim$300 m ($298 \pm 22$ m)[c] | |
| Nominal velocity, $v$ | | 330 km h$^{-1}$ $\equiv$ 91.6 m s$^{-1}$ | |
| Along-track diameter, effective SAR (Eq. A3) | 3.84 m | 1.28 m | 0.55 m |
| Cross-track diameter, $\sigma_{\text{Fresnel-limited}}$ (Eq. A6) | 6.0 m | 3.46 m | 2.27 m |
| Cross-track diameter, $\sigma_{\text{pulse-limited}}$ (Eq. A7) | | 9.49 m | |
| Cross-track diameter, $\sigma_{\text{beam-limited}}$ (Eq. A8) | $\beta_{2\text{ GHz, }84°}$: 540.24 m <br> $\beta_{8\text{ GHz, }31°}$: 166.39 m | $\beta_{12\text{ GHz, }22°}$: 116.63 m <br> $\beta_{18\text{ GHz, }15°}$: 78.99 m | $\beta_{18°}$: 95.03 m <br> $\beta_{17°}$: 89.67 m |
| Along-track spacing | | $\sim$4.5 m | |
| Transmit power | | 100 mW | |

[a]Manufacturer of Ka-band antennae does not provide a number versus frequency, only for the full range. [b]Often, processing is completed for a smaller bandwidth resulting in a slightly coarser resolution. [c]Based on the elevation variable in the CReSIS data profile.

## 2 Data

### 2.1 Cryo2IceEx/NERC DEFIANT 2022 (CRYO2ICEANT22) airborne campaign

ESA continued their validation programme CryoVEx in collaboration with the Technical University of Denmark (DTU) National Space Institute (DTU Space), CReSIS, University of Leeds, and the British Antarctic Survey (BAS) under the ESA Cryo2IceEx and NERC DEFIANT projects in December 2022 (29 November to 20 December). The 2022 airborne campaign (dubbed CRYO2ICEANT22) focused on the validation of multi-frequency altimetry observations along coordinated CryoSat-2 and ICESat-2 orbits over both land and sea ice, including the dedicated CRYO2ICE orbit over sea ice in the Weddell Sea. The survey flights were carried out using a BAS DASH-7 aircraft from Rothera Research Station. The instrument package included Ka-, Ku-, and C/S-band radar altimeters from CReSIS; an airborne laser scanner (ALS) of the type Riegl LMS Q-240i-80;

five dual-frequency GNSS Javad Delta receivers; a high-precision inertial navigation system (INS) of the type iMAR (Jensen, 2024); four GoPro 9/10 cameras; an Eppley radiometer with pyranometer and infrared thermometer; and GNSS-R (reflectometry). For this study, we focus only on the radar and lidar observations that have been pre-processed using data from the GNSS receivers and iMAR INS as input. We only utilise visible imagery from the nadir-looking GoPro as qualitative validation of our surface classification.

During the campaign, there was no in situ component for local observations of snow depth or ice conditions on sea ice, nor were other reference observations available to provide input on the conditions (e.g., active AWI's snow depth and ice mass balance buoys during December 2022 (ID 2017S54, 2022S110, 2022T88) did not drift far enough from the Neumayer III research station to be covered by the orbit, see Grosfeld et al., 2016). Thus, we have to rely on ERA5 reanalysis estimates (see Fig. 2, Section 2.2 and Section 5) of precipitation and air temperature for the discussion on radar penetration and the impact on snow conditions.

### 2.1.1 Ku/Ka radar and C/S snow radar

The CReSIS Ka- (32–38 GHz), Ku- (12–18 Ghz) and C/S-band (2–8 GHz) radars were provided by the University of Kansas, and allow for resolving various layers (e.g., snow versus ice over sea ice) depending on surface conditions (Rodriguez-Morales et al., 2014). The radars operate in frequency-modulated-continuous-wave (FMCW) mode over a wide bandwidth, resulting in a fine (a few centimetres) vertical resolution depending on the bandwidth. Depending on the surface conditions, C/S-band has a vertical resolution of 0.02–0.04 m (in snow), whereas Ku- and Ka-band have a vertical resolution of approximately 0.04 m (Rodríguez-Morales et al., 2021; Panzer et al., 2013). For the campaign, the radars can be accommodated in different modes, where the default mode is full bandwidth which supports simultaneous acquisition of all three radars in their full frequency range at altitudes up to ∼1,200 ft (∼365 m) above ground level (AGL), with an optimal operating altitude at 1,000 ft (300 m) and a minimum altitude of 800 ft (240 m). At nominal flight altitudes, the diameter of the footprint of the radars is in the order of 5 m. For a detailed description of the pre-processing steps resulting in the radar waveforms used in the study, see Appendix A.

During the under-flight, the radars were turned on (off) before (after) reaching the intended CRYO2ICE orbit ground track to ensure that the instruments were running as expected when at the location. During these stretches, as well as during calibration manoeuvres, the radars are likely to experience signal degradation due to various flight manoeuvres (e.g., turns, too high altitude), which should be filtered out from the science data. Due to the segmentation of the CReSIS data into frames and the activation of truncation procedures (see Appendix A3), the size of the waveform parameter changes significantly during such manoeuvres (the number of range bins increasing from approximately 1,200 to 25,000, or even more). For processing a full flight and combining all frames into one file, we consider only the frames where the aircraft was aligned with the intended satellite orbit (frames 74–232 selected manually, see extent in Fig. 1) and within the altitude of the default mode, resulting in range bins size in the order of ∼1,200, and an along-track distance coverage of 792 km.

### 2.1.2 Airborne laser scanner (ALS)

The Riegl LMS Q-240i-80 ALS, provided by BAS, uses three rotating mirrors, resulting in parallel scan lines on the surface with a maximum scan angle of $80°$. It operates in near-infrared (NIR) with a wavelength of 904 nm assumed to reflect at the a-s interface. While NIR (904 nm) waves are generally assumed to reflect from first-encountered surfaces, they are likely to penetrate several centimetres under optimal cold, low-density surface snow (Deems et al., 2013). However, we assume that under these conditions, such potential penetration is negligible. The ALS has a pulse-repetition-frequency of 10 kHz and a sampling configuration of 40 scan lines per second with 251 shots per line, providing regular coverage of the surface at nominal flight speeds. This configuration provides 1 observation per $m^2$ with an illuminated footprint of close to 0.7 m in diameter at a nominal flight height of 300 m AGL and ground speeds on 250 km h$^{-1}$. This corresponds to an across-track swath width of approximately 400 m, and a laser shot-to-shot accuracy of a few centimetres. The ALS point clouds were combined with GNSS-INS solutions using precise point positioning (PPP) to provide ellipsoidal heights. Ellipsoidal heights are elevations relative to the reference ellipsoid, in this case, World Geodetic System 1984 (WGS 84). This is computed as the range obtained from re-trackers (Section 3.1) subtracted from the altitude of the air- or spacecraft. Precise orbit and clock information, provided routinely by the International Geodetic Service (IGS), were used. This allows for accurate solutions even for flights with long baselines. Offset angles between ALS and INS measurements were calculated and applied using standard calibration procedures (Skourup et al., 2024, currently under final review at ESA per September 2024).

### 2.1.3 Visual cameras

The DASH-7 was equipped with a nadir-looking GoPro 10 with an image size of 5568 x 4176 pixels and a resolution of 72 dpi. The GoPro uses an internal GNSS mode and is set to capture images at 0.5-second intervals, with a field-of-view corresponding to approximately the ALS swath width. This setup ensures along-track overlapping images. The calibration procedures of the cameras are similar to those for the ALS.

### 2.2 ERA5 2 m air temperature and precipitation

Ideally, for the validation of snow depth derived from airborne Ka/Ku/C/S-band combinations, in situ observations would be collected to provide a ground truth to compare against. However, during this campaign, there was no in situ sea-ice component. Instead, we utilise estimates of precipitation and temperature from the ERA5 reanalysis data set within approximately $\pm1$ month of the under-flight date to evaluate the potential impact of atmospheric events. ERA5 provides estimates of several atmospheric, land, and oceanic climate variables (Hersbach et al., 2023). Here, we utilise the 2 m air temperature, total precipitation, and snowfall parameters, all available on a regular $0.25° \times 0.25°$ grid, averaged to daily resolution from 1 November 2022 to 31 December 2022. We use these observations to evaluate the temperature and precipitation conditions before, during, and after the campaign by computing the minimum, maximum, and average values of each parameter within a bounding box around the airborne flight path covering sea ice (see Fig. 2).

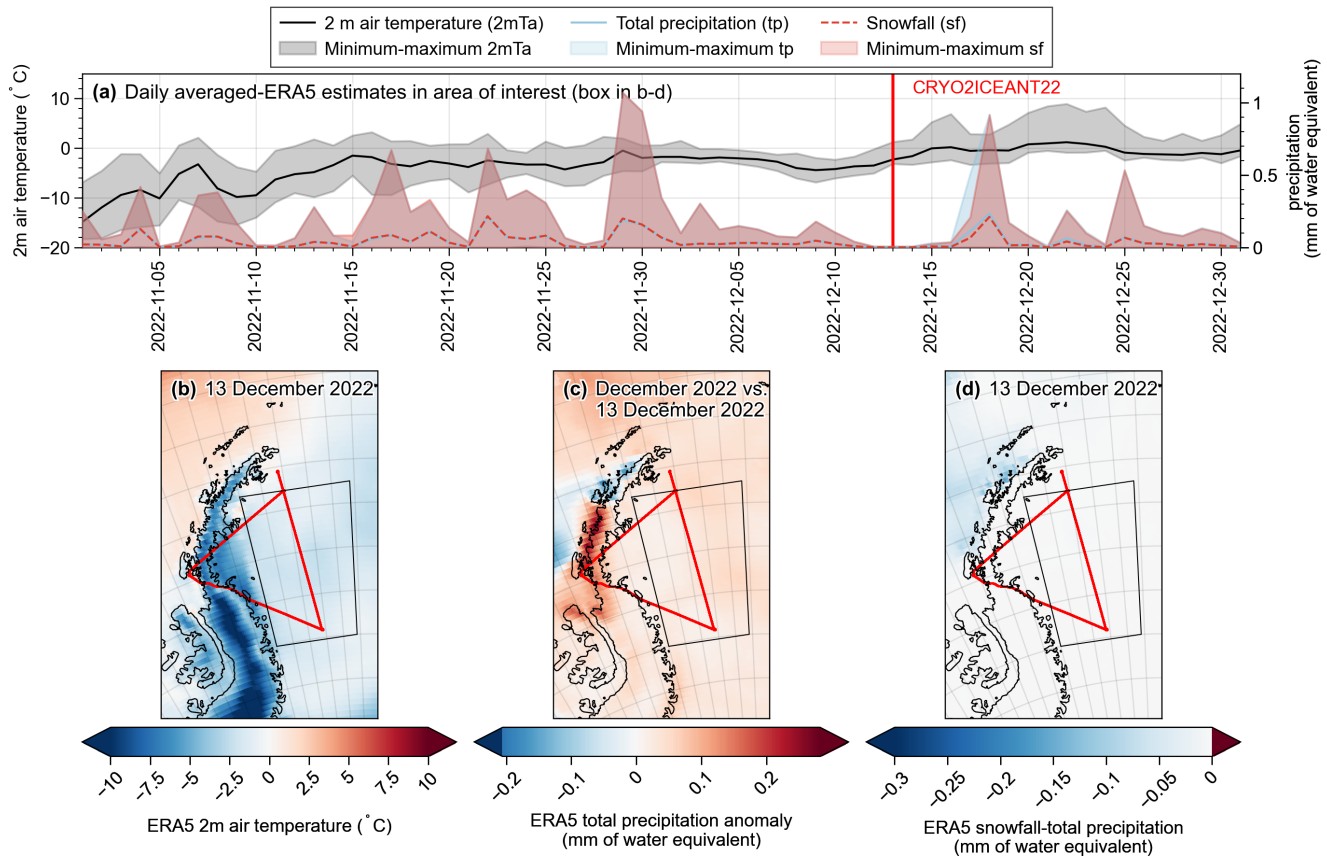

**Figure 2.** (a) ERA5 daily averaged 2 m air temperature and precipitation (total precipitation and snowfall) for 1 November to 31 December 2022, computed for the area marked by the black outline in (b-d). (b) 2 m air temperature on 13 December 2022, (c) total precipitation anomalies, computed as the difference on 13 December compared to the December 2022 average, and (d) the difference between snowfall and total precipitation on 13 December 2022. The CRYO2ICEANT22 under-flight on 13 December 2022 is outlined in red (b-d and Fig. 1).

## 3  Methodology

### 3.1  Surface retrieval methodology (re-trackers) for airborne radar altimeters

To extract the surface elevation from radar altimeters, one must identify the precise point on the radar echo (power spectrum, i.e., the waveform) that corresponds to the surface being measured (the re-tracking point) using a re-tracker. Various re-trackers are used in altimetry, and based on the measurement technique (snow radar or Ka/Ku-band radars), one may either apply a re-tracker to extract a single surface scattering horizon as the dominant interface or multiple scattering horizons based on the assumption that the radar echo contains information about several interfaces. The multiple-interface approach is the main

assumption applied to observations from FMCW snow radars, where the goal is to re-track the a-s and s-i interfaces. In contrast,

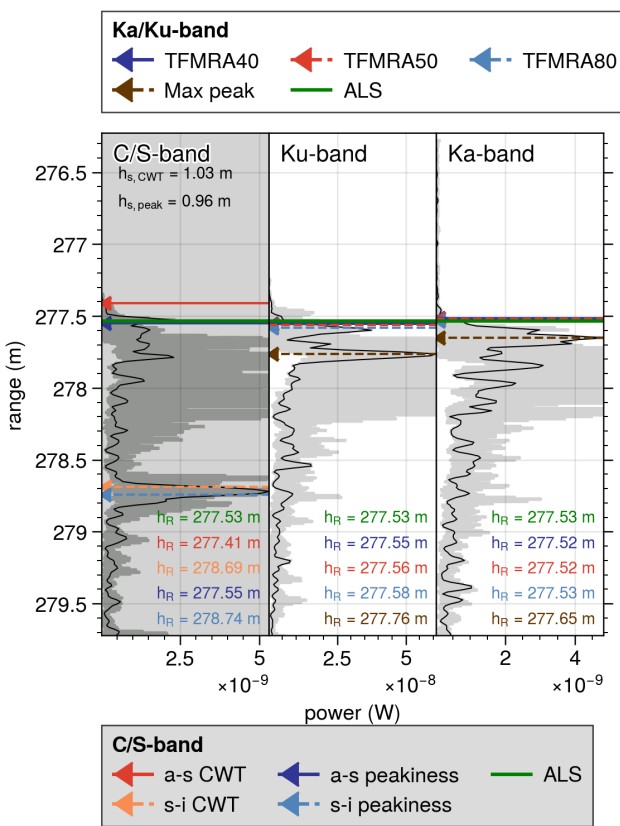

**Figure 3.** Individual waveform examples (in black) for each radar (Ku-, Ka-, or C/S-band) are provided, including the location of re-tracking points using various re-trackers (depending on the radar). The corrected range at different re-tracking horizons ($h_R$) and derived snow depths ($h_s$) for multiple scattering horizons using different re-trackers are also presented. In addition, the $\pm 5$ closest neighbouring waveforms (in grey) are shown to illustrate local variations in the waveforms. Here, Ka- and C/S-band have been aligned with the Ku-band's $z$-axis (using the lever arms provided in Appendix B) with $\Delta_{z,C/S} = 0.02$ m and $\Delta_{z,Ka} = -0.05$ m, where $z$ denotes the downward direction from the aircraft (or the range).

for conventional altimeters, such as Ku-band and Ka-band radars, the aim is to re-track the surface from which most backscatter is reflected, assuming one interface dominates the signal. Here, we specify the re-trackers used in this study for the extraction of single and multiple scattering horizons.

### 3.1.1 Single scattering horizons

Threshold-first-maximum-retracker-algorithm (TFMRA) is commonly used for satellite radar altimetry observations acquired over sea ice, where studies have investigated and currently apply different re-tracking thresholds (40%, 50%, 70%, or 80%, see e.g., Ricker et al., 2014; Tilling et al., 2018; Lawrence et al., 2018; Garnier et al., 2021; Rinne and Hendricks, 2023; Hendricks,

2022). For TFMRA, we re-track only points above the thermal noise level, determined as the average of the first 50 bins of the waveform. Three re-tracking thresholds are applied (40%, 50%, and 80%) to the first maximum. The first maximum is identified using a 30% threshold of the maximum value, and the re-tracking threshold is then applied to the first maximum to derive the range to the surface and, ultimately, the elevation. While the most commonly used empirical re-trackers for spaceborne sea ice altimetry employ TFMRA at 50% or 70%, recent airborne studies have investigated using a threshold-maximum-retracker-algorithm at 70% (De Rijke-Thomas et al., 2023). This algorithm uses the maximum power of the waveform to identify the surface, regardless of the origin of the backscatter. We also extract the maximum power here to evaluate the origin of the dominant scattering interface, although we do not use a threshold on the leading edge of the absolute maximum (similarly to ground-based studies of Willatt et al., 2023, where they also utilise the maximum amplitude, within a specific range-window of expected surface scattering to reduce the impact of sidelobes, as the main scattering interface).

### 3.1.2 Multiple scattering horizons

Currently, numerous retrieval methods for extracting a-s and s-i interfaces for snow radars exist, and several have been inter-compared over Arctic sea ice (Kwok et al., 2017). However, only two algorithms are currently provided as open-access, hence we only process the snow radar with those re-trackers.

Newman et al. (2014) presented the continuous wavelet transform (CWT) re-tracker for the detection of snow and ice interfaces from the CReSIS snow radar systems. This re-tracker is capable of identifying interfaces independent of radar systems and without relying on specified thresholds. Instead, CWT aims to detect two interfaces on the leading edge of the waveform by first identifying the range bin where the power first rises above the noise floor on the leading edge of the waveform. This range bin is expected to represent the illumination of the first interface (a-s). To identify the s-i interface, the range bin on the leading edge exceeding a level of noise and an estimate of radar clutter is selected. This bin is expected to represent the illumination of an interface with the highest power reflection coefficient (the theoretical s-i interface). This is achieved by preconditioning the echogram to make these locations more distinguishable. Since the s-i interface, under relatively undeformed and cold conditions, shall contribute most to the power distribution, no modification of the echogram is applied for re-tracking. However, for detecting the a-s interface, the logarithm of the radar echogram is taken to make the theoretical location of the a-s interface more distinguishable. The discretized version of the CWT is applied to find the two interfaces by the use of Haar wavelets, which are scaled step functions designed primarily to find sudden transitions within signals. The coefficients of the wavelets are summed to detect an interface, identified as the value where the summed coefficients are maximised. Here, the benefit is that it is independent of a set of fixed thresholds and that it is not affected by changes in transmitted power and received noise, which vary depending on the radar system. Based on evaluation over first- and multi-year Arctic sea ice, uncertainties were recorded at ~0.06 m over low-topographic targets (Newman et al., 2014).

Jutila et al. (2022b) developed the peakiness re-tracker (PEAK) specifically for the CReSIS 2–18 GHz snow radar system used on the AWI IceBird campaigns. This re-tracker enables snow depth retrievals even in more complex snowpacks, particu-larly in cases where the a-s interface is the main scattering surface. It is based on left- and right-handed pulse peakiness (PP) parameters following Ricker et al. (2014) as well as logarithmic and linear scale power thresholds. To detect the a-s interface,

the normalised waveform is analysed using the logarithmic scale similar to Newman et al. (2014). The first peak that is above the noise level by a user-defined power threshold and satisfies a user-defined left-hand peakiness threshold is defined as the a-s interface. The s-i interface is located in a similar way but using the normalised linear scale waveform, linear scale power threshold and right-hand peakiness threshold analysing up to five peaks. The last peak satisfying the thresholds is identified as the s-i interface, whether it is the maximum peak or not. If more than five peaks satisfy the thresholds, the waveform is regarded as ambiguous, and no interface locations are returned. The uncertainty of the retrieved snow depth was estimated to be 0.04 m (or 18% of the snow depth) over level and land-fast first-year sea ice. A detailed description can be found in Jutila et al. (2022b). A disadvantage of this method is its dependence on user-defined thresholds, which ideally should be verified with coincident ground-based measurements to ensure its accuracy.

Both re-trackers are applied using the open-source Python package pySnowRadar (King et al., 2020), where for CWT the following default parameters are used: one reference snow layer and a CWT precision of 10. For PEAK, the following default parameters are used: a logarithmic peak threshold of 0.6, a linear peak threshold of 0.2, and both left and right peakiness thresholds are set to 20. An example of derived ranges using the re-trackers presented herein on the airborne observations for all three radars is shown in Fig. 3. It is important to note that residual range sidelobes, which have been reported as an issue in previous studies (Kwok and Haas, 2015; Kwok and Maksym, 2014), have been significantly reduced through the use of the current version of the CReSIS processing chain. Additional steps can be taken to remove snow depths over highly deformed ice (provided through a topography parameter) using coincident lidar observations, however this has not been applied in our study due to inconsistencies between ALS and the a-s interfaces identified by the snow radars (see Section 4.1).

## 3.2 Pre-processing of airborne surface elevations and surface discrimination

Additional processing must be applied to the airborne observations (a) to derive a nadir lidar profile along the swath for comparison with the radar observations, (b) to align the lidar and radars by applying a calibration offset to account for internal delays in the radar system, and (c) to distinguish between floes and leads for comparing snow depth estimates only on floes.

### 3.2.1 Deriving nadir lidar profile along ALS swath

We extract a nadir lidar profile as the average of ALS observations within a search radius of 2.5 m from the centre location of each radar observation (latitude and longitude); see an example of the location of the nadir profile in Fig. 4a. This methodology has also been implemented in De Rijke-Thomas et al. (2023) and Jutila et al. (2022b). The nadir ALS profile allows for evaluation of where the radar scattering occurs within the snowpack along the flight track, assuming that the lidar scatters at the a-s interface. We note here that in the presence of broken floes or a highly heterogeneous ice cover (e.g., leads with small ice floes in between), the lidar will favour scattering from the ice floes as opposed to specular surfaces such as leads (which has also been discussed by Hendricks et al., 2010). Hence, in the case of such surfaces, the nadir lidar profile may reflect a different ice surface than the radar, as the radar instead will be dominated by specular returns from leads. This is further discussed in Section 4.3 and Section 5. Laser observations where the absolute difference of the radar elevations (derived using TFMRA50) and nadir laser profile are more than 3 m are noted as not-a-number.

**Table 2.** Manually detected leads from airborne observations used to compute the calibration offset ($\Delta$) between ALS and the radar system for each frequency band. The last row includes the sum of all observations and the average offset of each frequency band. The offsets shown here were calculated with TFMRA50 for Ka- and Ku-bands, and the PEAK re-tracker at the a-s interface (discussed further in Section 3.2.2). $N$ denotes the number of individual observations used per lead identified.

| | $N$ | latitude$_{max}$ (degrees N) | latitude$_{min}$ (degrees N) | $\Delta$C/S (m) | $\Delta$Ku (m) | $\Delta$Ka (m) |
|---|---|---|---|---|---|---|
| #1 | 9 | -65.1901 | -65.1908 | 4.47 | 0.38 | 0.15 |
| #2 | 11 | -68.4051 | -68.4059 | 4.52 | 0.31 | 0.18 |
| #3 | 9 | -68.4087 | -68.4094 | 4.57 | 0.36 | 0.24 |
| #4 | 21 | -69.3859 | -69.3875 | 4.56 | 0.35 | 0.22 |
| #5 | 20 | -69.3892 | -69.3907 | 4.56 | 0.35 | 0.23 |
| #6 | 8 | -70.7583 | -70.7589 | 4.56 | 0.36 | 0.24 |
| #7 | 18 | -69.9682 | -69.9696 | 4.56 | 0.35 | 0.24 |
| All | 96 | – | – | 4.54 | 0.34 | 0.22 |

### 3.2.2 Offset calibration to align radars with lidar

To account for internal delays in cables and electronics, a calibration offset ($\Delta$) between the ALS and radars (with the radar observations subtracted from the lidar observations) must be computed to obtain aligned absolute surface heights. We compute this offset using manually identified specular surfaces, i.e., leads along the flight track, where the laser and radars are expected to reflect from the same surface. We manually identify seven leads ($\sim$100 observations) and compute the mean offset per detected lead (see Table 2) from the ellipsoidal heights. From the seven mean offsets, we compute the mean offset per frequency band. It is important to note that within leads, the TFMRAs at any given threshold provided the same offset, except for the Ka-band at 80%, which was 0.02 m lower than the offsets at the other tested thresholds. For this study, we apply the calibration offsets of 0.34 m and 0.22 m for Ku- and Ka-band, respectively, neglecting the changes for Ka-band at 80% which are within the range resolution of the altimeter. For C/S-band, we used a-s and s-i interfaces identified with the PEAK re-tracker to compute the offsets, which resulted in 4.54 m for both interfaces. Using the CWT re-tracker, we obtained offsets of 4.74 m and 4.58 m for the identified a-s and s-i interfaces, respectively. This indicates a bias compared with PEAK, leading to thicker snow estimates when using the CWT re-tracker. Specifically, CWT would re-track 0.04 m below the s-i interface compared with PEAK and 0.2 m above the a-s interface, assuming that the re-trackers consistently re-track the same difference between interfaces. This results in a 0.24 m difference, equivalent to $\sim$0.19 m of snow depth, when using a snow density of 300 kg m$^{-3}$. However, it is likely that the interfaces are re-tracked differently depending on the surface conditions and the resulting waveform shape, thus, the bias may vary depending on the waveform. We note that PEAK is used as a reference since it identified the same offset over leads for both interfaces. The calibration offset for the snow radar does not impact derived snow depths using PEAK or CWT, since these are based on relative measurements between the identified interfaces and are not calibrated to align with the

ALS observations. We note that the alignment of the calibration is based on the assumption that the re-tracker successfully
and correctly re-tracks the lead at the same elevation. However, due to the increased range resolution of the airborne radar, the
different re-trackers may select different range bins as the time when the surface is best illuminated. Thus, we also provide the
offset to the maximum power in leads for each band. Here, the offset is computed as 0.31, 0.19 and 4.54 m for C/S-, Ku-, and
Ka-band, respectively, presenting biases of 0.03 m for both Ku- and Ka-band using TFMRA50 as reference (although within
the range resolution of the airborne system). There was zero bias for C/S-band when referenced to PEAK a-s. This suggests
that PEAK is successful in identifying the correct interface over leads, whereas CWT will re-track the a-s significantly before.

### 3.2.3 Discrimination between floes and leads in airborne radar observations

The necessity of discriminating observations from leads and floes is twofold. First, the radar and ALS observations perceive the
surface differently, with specular reflections from the water dominating the radar signal. Simultaneously, small floes within the
radar footprint might be present in the ALS swath and included in the observations used to generate the nadir laser profile. This
results in an increase in the elevation over leads compared with the radar-derived elevation, which would provide a snow depth
estimate when differencing laser and radar if not properly classified. Second, the aim of this study is to derive snow depth using
multi-frequency altimetry, and therefore, we are only interested in snow depth estimates over floes. To discriminate between
floes and leads, it is common in spaceborne radar altimetry to use a multitude of waveform parameters such as maximum power
(MAX) and PP, along with other waveform parameters specific to the delay-Doppler processing of SAR altimeters (e.g., Ricker
et al., 2014; Tilling et al., 2018, applied to Arctic studies). MAX is defined as the highest power within the range window, and
PP is computed as:

$$\sum_{i=1}^{N_{\text{WF}}} \frac{P_{\text{MAX}}}{P_{\text{i}}}, \tag{1}$$

where $N_{\text{WF}}$ denotes the number of range bins, $P_{\text{MAX}}$ is maximum power and $P_{\text{i}}$ is the power of the $i$th range bin. For easier vi-
sualisation, we convert to normalised values using maximum-minimum normalisation. Converting from normalised waveform
parameters and back, we use $y = y_{\text{NORM}} \cdot (\text{maximum}(x)\text{-minimum}(x)) + \text{minimum}(x)$, where $y$ denotes the original waveform
parameter value, $y_{\text{NORM}}$ denote the normalised waveform parameter and $x$ is the array of the waveform parameter. We note
that, while we use similar formulas to derive the waveform parameters as those used for spaceborne (and airborne e.g., Zyg-
muntowska et al., 2013) altimeter observations, we do not necessarily expect similar thresholds due to the different instruments
and platforms. Instead, we derive specific thresholds discriminating between floes and leads in the CReSIS radar. For PP, we
compute PP from $-100$ and $+156$ bins after MAX to align with the number of range bins for CryoSat-2, and to limit the impact
of thermal noise on the estimated PP values.

We find that PP provides the strongest discrimination between surface types, shown by strong separation in the cumulative
probability of the two distributions and minimal separation for MAX (Fig. 4c). From the cumulative distributions (Fig. 4c), we
use the 10% cumulative probability of PP, applied to the distribution of manually detected leads, as our lower bound for leads,
and 60% cumulative probability for the distribution of the entire flight (approximately where the distribution of leads starts) as
lower bound for mixed surfaces. Both values are rounded down to integers. Visual inspection of the waveform parameters and

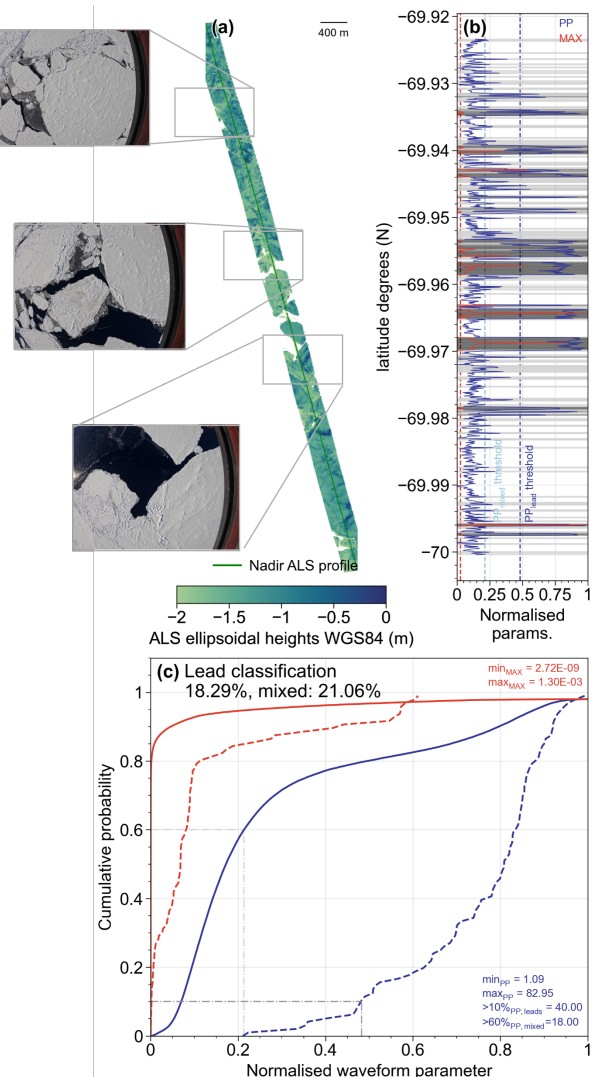

**Figure 4.** Discrimination of sea ice floes and leads in the airborne observations. (a) ALS swath along with optical images to visually confirm the surface classification. (b) Normalised waveform parameters (pulse peakiness (PP) and maximum power (MAX)) shown along the nadir track of panel (a) with the identified surface types "leads" and "mixed" (based on sub-panel c) highlighted as dark grey and light grey horizontal lines, respectively. The thresholds determined from panel (c) are also shown in panel (b) as dashed vertical lines. (c) Cumulative probability distribution of waveform parameters MAX (red) and PP (blue) for the entire flight (solid lines) and manually detected leads (dashed lines). Minimum and maximum values for the waveform parameters are provided, along with downwards rounded waveform parameter values at 10% ("leads") and 60% ("mixed"), which are used to identify surfaces based on comparisons with the cumulative distributions of each surface type. The statistics of observations classified as "leads" and "mixed" surfaces are provided for the entire under-flight. Grey dashed lines align the chosen cumulative probability thresholds with the location of the respective distribution used.

their lower bounds in Fig. 4b, compared with the ALS swath in Fig. 4a, supports the surface classification thresholds. Along the orbit, PP ranged between 1.09–82.95 with a 10% lower bound threshold of 40 using the distribution of manually detected leads and a 60% lower bound with PP of 18 using the full distribution, values which coincide well with the thresholds from former studies. Here, Ricker et al. (2014) classified leads through a combination of waveform parameters including $PP \geq 40$ and Tilling et al. (2018) denoted specular surfaces as $PP \geq 18$. In Tilling et al. (2018), mixed surface types had PP down to 9, from which one might argue that our classification is less robust and some mixed surfaces might still be included in the processing. Applying the requirement of leads and mixed surfaces having PP above thresholds of 40 and 18 (until 40), respectively, we identify 18.29% of the observations as leads and 21.06% as mixed (a total of almost 40% of observations), and remove these from further processing of snow depth estimates and evaluation of radar penetration.

### 3.3 Retrieval of snow depth from altimetry freeboards and/or elevations

Similar to Kwok et al. (2020), we calculate snow depth ($h_\mathrm{s}$) by differencing the height of interfaces at a-s ($h_\mathrm{a\text{-}s}$), equivalent to the total freeboard, and s-i ($h_\mathrm{s\text{-}i}$), equivalent to the sea ice freeboard, following:

$$h_\mathrm{s} = h_\mathrm{a\text{-}s} - h_\mathrm{s\text{-}i}. \tag{2}$$

The sea ice freeboard (or s-i interface, $h_\mathrm{s\text{-}i}$) is related to the radar freeboard, assuming full penetration to the s-i interface, which is often considered the case for Ku- and C-band radars, as follows:

$$h_\mathrm{s\text{-}i} = h_\mathrm{s\text{-}i,\,radar} + h_\mathrm{s}(\eta_\mathrm{s} - 1). \tag{3}$$

The second term in Equation (3) accounts for the reduced propagation speed of the radar wave ($c_s$) as it travels through the snowpack with a bulk density $\rho_s$ assuming a cold and dry snowpack. The refractive index ($\eta_\mathrm{s}$) at Ku-band, $\eta_\mathrm{s} = c/c_\mathrm{s} \cdot (\rho_\mathrm{s}) = (1 + 0.51\rho_\mathrm{s})^{1.5}$ (Ulaby et al., 1986), and $c$ is the speed of light in free space. Through combination of Equation (2) and (3), snow depth ($h_\mathrm{s}$) is given by:

$$h_\mathrm{s} = \frac{h_\mathrm{a\text{-}s} - h_\mathrm{s\text{-}i,\,radar}}{\eta_\mathrm{s}} \tag{4}$$

Now, snow depth ($h_\mathrm{s}$) is related to the differences between the a-s and s-i interfaces (extracted by a radar) with one free parameter, $\eta_\mathrm{s}$, which is dependent on the snow bulk density directly related to the snow conditions. Massom et al. (2001) presented an overview of Antarctic snow conditions from field observations acquired between 1992 and 1998 under a variety of seasonal and geographical conditions. The average snow density values from a number of studies ranged between 247 and 391 kg m$^{-3}$, while the overall range was from 99 to 817 kg m$^{-3}$. These results have guided some studies (e.g., Kwok and Kacimi, 2018; Kacimi and Kwok, 2020) to use 320 kg m$^{-3}$ as their reference bulk snow density. Kwok and Maksym (2014) used 300 kg m$^{-3}$, and noted that with a snow density uncertainty of 50 kg m$^{-3}$, the overall uncertainty of radar-derived snow depth ranged between 0.03 and 0.05 m for snow depths between 0.1 and 0.7 m. One could argue for using a bulk snow density of ~350–360 kg m$^{-3}$ in this study, as the observations were acquired during Antarctic summer (December to February), and this value is used for that time-period in the seasonal varying snow density scheme applied to CASSIS (Lawrence et al., 2024,

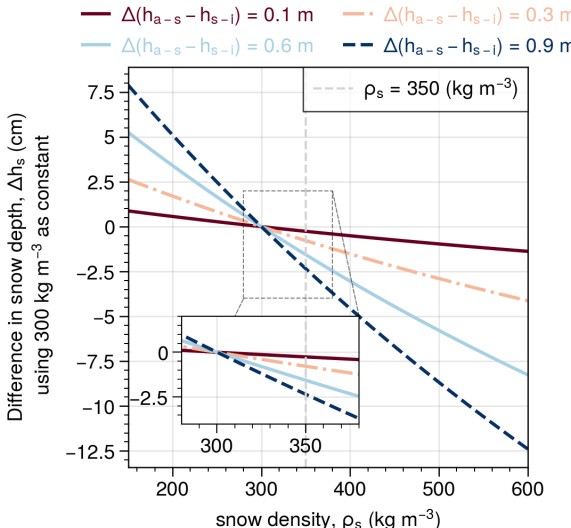

**Figure 5.** Difference in snow depth ($\Delta h_s$) from using a constant snow density of 300 kg m$^{-3}$ compared with varying snow densities between 150–600 kg m$^{-3}$ (based on field data presented in Massom et al., 2001) shown for four examples of height differences between interfaces ($\Delta(h_{a\text{-}s} - h_{s\text{-}i})$). An inset for the density range of 280–380 kg m$^{-3}$ is shown and discussed further in Section 3.3.

their Fig. 2a). Kacimi and Kwok (2020) also noted that while there is no accepted bulk density of Antarctic snow, Massom et al.
(2001) suggested 200–300 kg m$^{-3}$ under cold and dry conditions and higher density (320–500 kg m$^{-3}$) for warm and windy
conditions. Here, we present a sensitivity study on the impact of choosing different bulk snow densities. Figure 5 shows that
even for very thick snow depths, using either 300, 320 or 350 kg m$^{-3}$, the maximum difference was $\sim -0.03$ m. Therefore,
we do not expect a large impact when using slightly different snow densities, which is within the order of the resolution of the
airborne instruments themselves. Instead, the largest differences occur under extreme events at the edges of the density range.
For the densities reported by Massom et al. (2001), differences range from $-0.13$ to $0.08$ m in these cases (using a range of
150–600 m$^{-3}$). However, we note that in the case of extreme cases, the largest impact will likely be seen on the (in)availability
of detected s-i interfaces rather than on the derived snow depth. Based on this sensitivity analysis and the limited impact of
derived snow depth, we keep the bulk density of snow constant at 300 kg m$^{-3}$.

## 4    Results

### 4.1    Microwave penetration into the snow

For comparison of the backscatter horizons (interfaces, i.e. a-s or s-i interfaces) and evaluation of penetration capabilities, it is
assumed that the ALS observations, due to zero penetration into the snow cover, are used as a reference for the a-s interface.
Traditional hypotheses of the radars include Ka-band primarily scattering at the a-s interface, Ku-band at the s-i interface (over

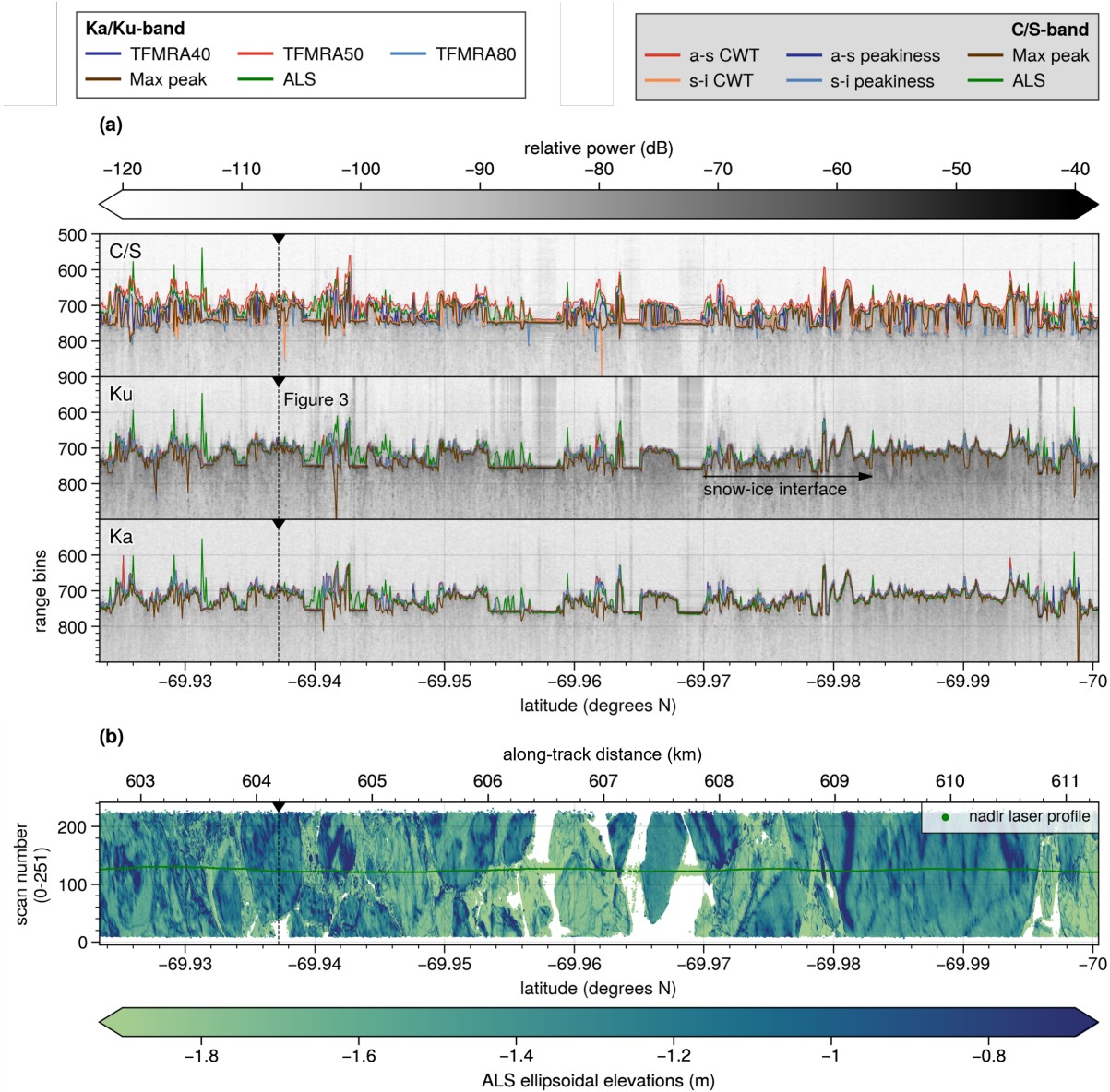

**Figure 6.** (a) Echograms at C/S-, Ku-, and Ka-band for the subset with overlaid re-tracked interfaces for all re-trackers. In the Ku-band, what appears to be a s-i interface is somewhat apparent based on a qualitative guess corresponding to the location of the s-i interface observed in the C/S-band; however, it does not represent the maximum scatter or somewhere on the leading edge of Ku-band. (b) Corresponding ALS swath and nadir profile shown as a function of scan number (not projected).

cold and dry snow), and that C/S-band tends to reflect at the s-i interface at maximum scattering (e.g., Kurtz et al., 2013; Kwok et al., 2017; Jutila et al., 2022b; **?**). Furthermore, airborne Ka- and Ku-band traditionally assume one surface to primarily contribute (thus, we only track one point of the waveform). In contrast, we assume the C/S-band to be influenced enough by

**Table 3.** Comparison between ALS and scattering horizons of C/S-band (snow radar), Ku-, and Ka-band over floes (observations with surface classification as leads or mixed removed). Here, $\mu$ denotes arithmetic mean, $\sigma$ is the standard deviation, and $N$ is the number of observations (in total, there were floe observations: $N = 56{,}020$). For the different versions of $N$, the percentages compared to the original data count ($N$) are provided in parentheses. We note here that the comparison between ALS and the radars has not accounted for slower wave propagation, thus we allow an uncertainty up to 0.1 m (i.e., double the range resolution in snow, which is about 0.04–0.05 m, see Appendix A2 and Table 1). Furthermore, we consider observations above 1.5 m as artefacts, since this would equate to a snow depth of 1.2 m using a density of 300 kg m$^{-3}$, which is considered towards extreme snow depths (Massom et al., 2001).

| Frequency-band | Re-tracker | $N_{<-0.1\,m}$ Above snow surface | $N_{>-0.1\,m\,\&\,<0.1\,m}$ Snow surface | $N_{>0.1m\,\&\,<1.5\,m}$ Within snow pack | $N_{>1.5\,m}$ Artefact | $\mu(\Delta_{>0.1\,m\,\&\,<1.5\,m}) \pm \sigma$ (m) |
|---|---|---|---|---|---|---|
| C/S-band | MAX | 2,826 (5.0%) | 9,478 (16.9%) | 41,637 (74%) | 2,079 (3.7%) | $0.57 \pm 0.33$ |
| | a-s$_{CWT}$ | 45,155 (80.6%) | 8,423 (15.0%) | 2,380 (4.2%) | 62 (0.1%) | $0.37 \pm 0.28$ |
| | s-i$_{CWT}$ | 5,095 (9.1%) | 14,701 (26.2%) | 34,292 (61.2%) | 1,932 (3.4%) | $0.59 \pm 0.33$ |
| | a-s$_{PEAK}$ | 16,188 (28.8%) | 22,120 (39.5%) | 17,301 (30.9%) | 481 (0.9%) | $0.5 \pm 0.32$ |
| | s-i$_{PEAK}$ | 9,058 (16.2%) | 7,242 (12.9%) | 37,740 (67.4%) | 1,980 (3.5%) | $0.62 \pm 0.34$ |
| Ku-band | MAX | 3,437 (6.1%) | 18,300 (32.7%) | 33,278 (59.4%) | 1,005 (1.8%) | $0.44 \pm 0.31$ |
| | TFMRA40 | 13,926 (24.9%) | 29,741 (53.1%) | 12,084 (21.6%) | 269 (0.5%) | $0.38 \pm 0.29$ |
| | TFMRA50 | 12,462 (22.2%) | 30,064 (53.7%) | 13,202 (23.6%) | 292 (0.5%) | $0.38 \pm 0.29$ |
| | TFMRA80 | 9,930 (17.7%) | 30,246 (54.0%) | 15,521 (27.7%) | 323 (0.6%) | $0.37 \pm 0.29$ |
| Ka-band | MAX | 5,313 (9.5%) | 24,024 (42.9%) | 25,866 (46.2%) | 817 (1.5%) | $0.42 \pm 0.3$ |
| | TFMRA40 | 17,287 (30.9%) | 29,234 (52.2%) | 9,343 (16.7%) | 156 (0.3%) | $0.36 \pm 0.28$ |
| | TFMRA50 | 15,429 (27.5%) | 29,995 (53.5%) | 10,419 (18.6%) | 177 (0.3%) | $0.35 \pm 0.28$ |
| | TFMRA80 | 12,284 (21.9%) | 30,740 (54.9%) | 12,787 (22.8%) | 209 (0.4%) | $0.35 \pm 0.27$ |

both interfaces to re-track both. We note that these conditions are not necessarily met during this campaign, with the relatively warm weather conditions. This is further discussed in Section 5. To evaluate the full extent of penetration, we here use MAX to present the strongest scatter.

The maximum power was reflected within 0.10 m from the a-s interface (using ALS) for 32.7% and 42.9% of the observations at Ku- and Ka-band, respectively (Table 3), from which 82–86% are coincident with those identified by TFMRA50 (see also Section 4.2). This is contrary to the assumption of total penetration at Ku-band scattering at the s-i interface. However, ~60% of Ku-band observations using MAX were scattered lower in the snowpack, between 0.1 m and 1.5 m, when compared to ALS. Since the a-s interface plays a significant role in the backscattering of the Ku-band radar signal, conventional re-trackers used for spaceborne estimates are unable to retrieve all the information available from airborne observations. Instead, the variability of re-tracked interfaces using MAX provides an indication of how much the contribution varies between interfaces. For comparison, Ka-band MAX is also reflected in the snow in 46% of the observations assuming ALS represents the a-s

interface. This suggests that either (1) the Ka-band is able to penetrate into the snow, or (2) the laser and the Ka-band radar are dominated by different scattering interfaces that might provide a deeper snow depth than actually present. Armitage and Ridout (2015) noted that the Ka-band reflects from mid-way in the snowpack and upwards, which appears to be consistent with several instances here (Fig. 6a Ka-band).

We observe that the Ku-band occasionally receives MAX reflections further into the snowpack, but rarely to the extent seen with the C/S-band. Additionally, there are instances where some s-i interface contribution is visible at Ku-band, but only after MAX has been reached (Fig. 6a, Ku-band). This suggests that a-s interfaces (or other internal layers or snow metamorphism) can be the dominating contributor to the waveform, but that the s-i interface may be extracted from a less dominant peak. Future work on revisiting retrieval methods where these instances occur under the assumption of multiple interfaces being present is encouraged. For all radar frequency bands, MAX is reflected more than 1.5 m lower than ALS between 1.5% and 3.7% of the time (Table 3), with the snow radar reflecting further than 1.5 m into the snowpack more frequently. In several instances, the Ka-band is reflected slightly below the Ku-band (see Fig. 3 or the larger variability in average values between TFMRA re-trackers in Fig. 7). This suggests a greater sensitivity to re-tracking thresholds at Ka-band and indicates that volume scattering may play a role, resulting in more scattering and delay in pulse travel time. These results open up a discussion on whether beyond footprint size, the frequency and the dominant backscatter mechanisms at airborne scales should be considered and not neglected.

MAX in C/S-band is only reflected above ALS in 5% of cases and has fewer observations at the same interface as ALS (16.9%), highlighting its ability to penetrate into the snowpack and primarily reflect at an interface below (Table 3). For the 74% of cases where MAX in C/S-band extends within the expected snow depth range, also considering the uncertainty of the observations (0.1 m to 1.5 m), the average depth below ALS is 0.57 m. In comparison, Ku- and Ka-band MAX within this range have average values of 0.44 and 0.42 m, respectively. If we consider all observations equally (and not disregard observations before or close to the a-s interface), Ku-band MAX reflects about 0.2 m above C/S MAX, and Ka-band MAX reflects an additional 0.1 m further up on average.

## 4.2 Air-snow and snow-ice: tracing the interfaces

Here, we evaluate the tracked interfaces using conventional assumptions and re-trackers applied. It is noticeable that, for the majority of the track, the retrieved elevations for both Ka- and Ku-band using TFMRA (40, 50 or 80%) align with the ALS observations (Fig. 6a, Fig. 7a–c, and Table 3). This indicates that the first significant peak at both Ka- and Ku-band is often reflected at the a-s interface. Specifically, between 52% and 55% of observations were within 0.1 m of the ALS interface at both Ka- and Ku-band, using TFMRA for all re-tracker thresholds (Table 3). In addition, several inconsistencies between elevations from the ALS nadir profile and the radars are observed where the ALS favours scattering from broken floes within open water areas, such as leads, whereas the radar is dominated by the specular reflection (see Fig. 6). Similarly, when highly heterogeneous surfaces occur within the radar footprint, ALS elevations from the same surface show inconsistencies. Thus, the different viewing geometries and sampling of the instruments lead to discrepancies in the surfaces explored (e.g., 9.5–30.9%

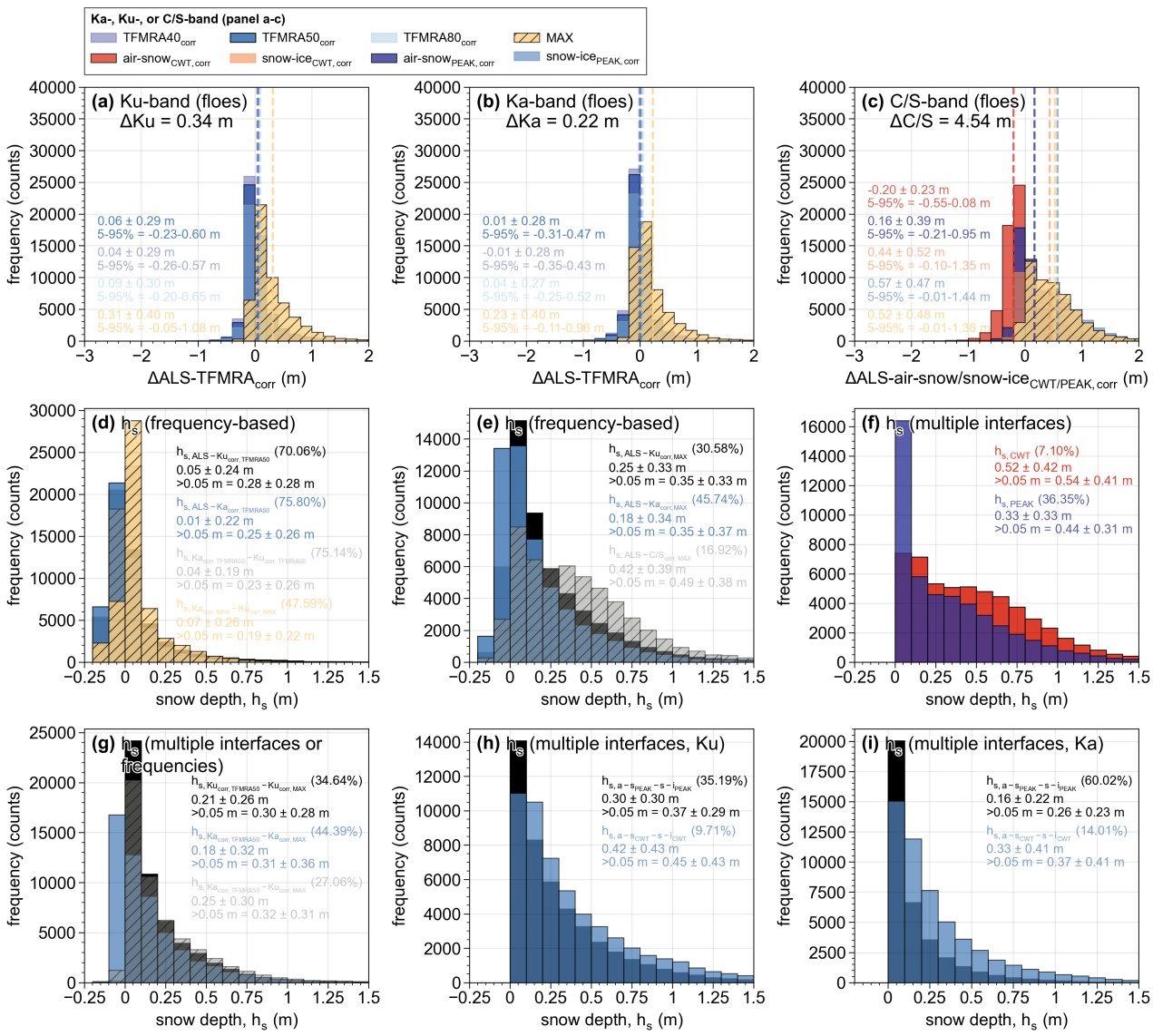

**Figure 7.** (a-c) Comparison between different re-trackers as ALS observations (nadir profile), where (a) shows ALS compared with TFMRA at 40, 50, and 80% and MAX for Ku-band, (b) shows the same as (a) but for Ka-band, and (c) presents the comparison of interfaces identified using PEAK and CWT for C/S-band as well as MAX. Statistics show average and standard deviation, along with 5–95% percentiles. (d-i) Statistics and distribution of snow depth estimations based on the combination of different frequencies, re-trackers and interfaces re-tracked. The percentage in parentheses denotes the number of observations below the threshold value used for statistics on the third line for each distribution. Bin width in sub-panels (a-c) is 0.2 m and 0.1 m for (d-i). The subscript "corr" refers to the applied calibration offset noted in (a-c) computed from the manually identified leads and presented in Table 2. Note the different y-axis scaling for sub-panels (d-i).

of the observations at Ka-band and 6.1–24.9% at Ku-band are reflected before the laser scanner, assuming an uncertainty up to 0.1 m).

The impact of a-s interface determination for the snow radar re-trackers shows that CWT re-tracks a surface before the ALS in 80.6% of the cases (Table 3), as opposed to PEAK, where this occurs in 28.8% of the cases (following the same tendency as TFMRA for both Ka- and Ku-band). For both Ku- and Ka-band, whenever TFMRA re-tracks a peak that is below the a-s

interface (by more than 0.1 m) and within the expected depth range, the average are ∼0.38 and ∼0.35 m for Ku- and Ka-band, respectively. However, this applies only for about 16.7–27.7% of cases, depending on re-tracking threshold used. Were we to consider all observations equally (Fig. 7a–c), TFMRA at all thresholds primarily re-tracks the a-s interface within the expected uncertainty of the radar system (maximum deviance from ALS is on average 0.09 m), with CWT a-s primarily re-tracking above ALS (on average by 0.2 m), and PEAK a-s is within 0.01 m from ALS. CWT's s-i interface is re-tracked ∼0.1 m above

the MAX power, whereas PEAK on average re-tracks s-i ∼0.05 m below MAX.

We note that PEAK has been applied with default parameters based on conditions during an Arctic spring campaign. Ideally, coincident in situ snow depth observations (either from ground-based radars, lidar scanning, or other snow depth estimates) are needed to align the thresholds with the conditions of the current campaign, since waveforms will differ in shape and magnitude depending on radar system and flight altitude. Significant work is required to understand how to tweak PEAK so that both

interfaces re-track the expected interfaces. One method would be to align the vertical laser profile with the PEAK a-s interfaces over homogeneous floe surfaces, assuming both instruments are dominated by the same surface conditions and represent the surface similarly. However, this still leaves out how well the s-i interface can be tweaked. Here, in situ observations are crucial. Furthermore, PEAK was originally developed for the 2–18 GHz snow radar system of AWI IceBird, and therefore, the applicability of this method over the 2–8 GHz system may be limited. Although comparing along the radar echogram in Fig.

6 show that when clear peaks from the s-i interface are visible, PEAK appear to re-track them, albeit the interface could be re-tracked some bins before/after MAX is reached, potentially limiting the total snow depth by some range bins. Additional work is required to understand how applicable PEAK is for the 2–8 GHz systems, which could be evaluated from former OIB campaigns whenever additional in situ snow observations are available. Such work is out of the scope of the current study. Nonetheless, using PEAK for 2–8 GHz systems looks promising.

## 4.3 Deriving airborne snow depths from combinations of re-trackers

In total, 16 different combinations of retrieved surfaces with different re-trackers and frequencies were investigated for snow depth estimations. Figure 7d presents snow depth computed between ALS and Ka/Ku-band (TFMRA50), Ka- and Ku-band (TFMRA50), or Ka- and Ku-band (MAX). Here, TFMRA50 generally reflects at or close to the a-s interface (∼70–76% of snow depth is below 0.05 m for Ku- and Ka-band, respectively). Removing these observations provides snow depth on average

around 0.25–0.27 m depending on frequency. Differencing Ka- and Ku-band to derive snow depth using TFMRA50 or MAX provides limited snow depth (75.14% and 47.59% are below 0.05 m). Again, TFMRA50 for both frequency-bands show to primarily (and clearly) reflect the a-s interface, whereas MAX diverges more. At times, Ku-band will have contributions from within the snowpack and/or closer to the s-i interface, resulting in some average snow depths of ∼0.2–0.3 m. However, this

average is significantly lower than expected based on the CASSIS model estimates presented in Fig. 1 and the snow radar re-
trackers (CWT and PEAK, Fig. 7f). Comparing TFMRA50 between the two frequency-bands also suggests that the interfaces
tracked in 3/4 of the times (likely the a-s interface) are retrieved more consistently, whereas the contribution to the MAX peak
at both frequencies appears to differ more often (average of 0.05 m difference when converting to snow depth, but in almost
50% of times). Investigating snow depths higher than 0.05 m (range resolution in snow) using Ka- and Ku-band MAX as the
a-s and s-i interfaces, respectively, the average snow depth retrieved is 0.19 m.

Evaluating how MAX compares with ALS provides an idea of how far, at the different frequency-bands, the penetration
occur. Here, snow depths using ALS and Ka-band MAX more often represent scattering from the same interface, within 0.05
m from the a-s interface (in almost 46% of times) compared to Ku-band ($\sim$31%) and C/S-band ($\sim$17%), see Fig. 7e. Average
snow depths (above 0.05 m) are 0.35 m, 0.35 m, and 0.49 m for C/S-, Ku-, and Ka-band, respectively. C/S-band extends
almost 0.15 m below on average, which supports the assumption and findings of non-complete penetration at Ku-band (and to
some extent Ka-band), but rather scattering within the snowpack (whenever it is not reflected at the a-s interface) or limited
penetration over thicker snow (Ricker et al., 2015). Interestingly, Ka- and Ku-band agree within less than a centimetre on
average when considering snow depths above 0.05 m, suggesting that either (a) they both penetrate into the snowpack and also
here, backscatter at same interface, or (b) ALS has for these instances identified the a-s interface further above, and that this
MAX is really reflected at the a-s interface, making these snow depths an artefact of the methods.

To evaluate whether snow depth can be retrieved from both radars individually, we use TFMRA50 as an estimate of the a-s
interface, and MAX as an estimate of the s-i interface, since this would limit the impact of viewing geometry, and provide
an idea of whether snow depth can be retrieved for both frequencies under these conditions (see Fig. 7g). Here, more often
Ka-band is reflected at the a-s interface with $\sim$44% having snow depths less than 0.05 m (MAX and TFMRA50 reflected more
or less at the same surface), whereas it is $\sim$34.5% for Ku-band. Considering only the instances where snow depth of more than
0.05 m is achieved, MAX at Ka- and Ku-band reflects at approximately the same location and provides $\sim$0.3 m of average
snow depth (with Ka-band having slightly higher snow depths, likely due to it being reflected slightly before Ku-band), see Fig.
7g. Using Ka-band at TFMRA50 as the a-s interface and Ku-band at MAX as the s-i interface, in $\sim$27% of the cases the snow
depth is less than 0.05 m. For the remaining cases, the average snow depth is once again close to 0.3 m. Thus, Ku-band appears
unable to extend fully to the s-i interface in many instances when compared with ALS-C/S$_{MAX}$, but is reflected on average
midway into the snowpack – as is Ka-band. Further work is necessary to evaluate this across other campaigns, in different
regions under varying conditions to understand fully its relevance. However, it may also explain why Garnier et al. (2021)
observed thinner snow depths from KaKu airborne derived estimates from the Arctic CryoVEx 2017 campaign compared to
ALS/Ka from the same campaign.

Applying CWT and PEAK to Ka- and Ku-band separately to evaluate the potential of extracting snow depth from multiple
interfaces by Ka- or Ku-band alone is presented in Fig. 7h–i. This is particularly interesting, since PEAK is able to re-track the
s-i interface at peaks extending beyond MAX. Here, in 35% of cases, PEAK at Ku-band retrieved snow depths lower than 0.05
m or provides no snow depth information at all, which is in the scale of TFMRA50-MAX at Ku-band. For the snow depths
above 0.05 m, averages are around 0.37 m when using PEAK, similar to those obtained by using ALS and MAX at Ku-band.

This reflects that, in the majority of cases, PEAK identified the s-i interface at MAX rather than at subsequent, but lower, peaks. Several instances where qualitatively the s-i interface contributes at Ku-band albeit is not the main scattering interface, PEAK was unable to extract this interface. This is likely due to either the power and right-hand peakiness thresholds applied being too strict for such interfaces to be re-tracked, or additional clutter after the peak would mean that the peak cannot be distinguished as a separate peak. Once more, tweaking of parameters would provide additional insights granted ground-based observations were available to validate the tweaking.

Using CWT for Ku-band results in ∼10% of snow depths below 0.05 m, and an average of 0.45 m (Fig. 7h). This is interesting, since CWT by definition is not able to re-track peaks below the MAX (it only re-tracks on the leading edge). However, what appears to drive these snow depths is that the a-s interface is re-tracked significantly before the a-s interface in PEAK, TFMRA in Ku-band at any threshold, and ALS (not shown). As such, while the range of snow depths computed using CWT are within expectations and on average follows ALS-C/$S_{MAX}$ well, the estimated location of the interfaces do not necessarily reflect the actual location. For Ka-band, PEAK is only able to extract a snow depth estimate in 40% of cases, reaching on average ∼0.26 m, which is about 0.1 m thinner than Ku-band. Here, it highlights the lower penetration at Ka-band. Applying CWT at Ka-band also presents higher snow depths (on average 0.37 m, considering only depths above 0.05 m) compared with PEAK which is again counter-intuitive. Along-track comparison over echogram (not shown) also highlights large variability in where the a-s interface is re-tracked at Ka-band, which could be caused by sidelobes that either have not been properly suppressed through deconvolution or larger leading edge at Ka-band caused by a potential higher sensitivity to scattering at this frequency.

Finally, we compare the snow depths retrieved using the PEAK and CWT re-trackers for the C/S-band snow radar. Here, clear differences in both shape of distribution and magnitudes of snow depths are evident. CWT rarely observes snow depth of less than 0.05 m (7.10% of cases) whereas PEAK observes it in ∼36% of cases (including times when snow depths are not available). On average, CWT retrieves 0.1–0.2 m thicker snow (depending on whether snow depths of less than 0.05 m are included or not). Here, it is clear that the snow radar algorithms show large variation and further work is necessary to fully understand the limitations of using snow radars over different surfaces. In all cases, the standard deviation of the derived snow depths are in the order of the average snow depths reflecting the high variability in retrieved snow depths.

## 5 Discussion and outlook

The assumptions of Ku-band penetrating the snow cover with a primary scattering horizon at the s-i interface appear not to hold for snow on Antarctic sea ice, especially not during the summer, considering the results herein. Precipitation events (snowfall, rainfall), melting, brine wicking leading to saline snow, and/or flooding due to increased snowfall precipitation and depression of floes, may cause the snow to become wet or snow metamorphism to occur (e.g., refreezing, snow-ice formation, ice lenses), which alter the dielectric properties and limits the Ku-band (and Ka-band) penetration. C/S-band appears less impacted by this, as seen by the deeper penetration occurring for C/S-band MAX and the detection of apparent s-i interfaces with both PEAK and CWT at, on average, ∼0.6 m below the a-s interface detected by ALS. For the area of interest (box in Fig. 1, 2b–

d), ERA5 simulates average 2 m air temperatures below freezing degrees until after the campaign (since 1 November 2022), although with maximum temperatures reaching thawing degrees for several instances throughout November (6–7 November, 12–22 November, 25 November–2 December, and 13 December onwards). It is also worth noting that the average temperature up until the campaign occurs around $-5$ °C, which could impact the snow and limit the radar penetration as shown by the presence of liquid water in snow at such temperatures (Barber et al., 1995; Kurtz and Farrell, 2011; Kurtz et al., 2013; Rösel et al., 2021). We have not removed data where the air temperature reaches $-5$ °C or above since the temperature data are based on reanalysis, however, we note that the presence of liquid water in the snowpack would significantly limit Ku-band and Ka-band. Figure 2a shows relatively warm temperatures in the latter part of November, which could have produced a complexly layered snow pack with multiple melt and freeze layers as well as large snow grains. This would limit penetration, especially at higher frequencies. We also note that while a thermal instrument was carried during the campaign and could potentially provide insights into surface temperatures, taking into consideration the field-of-view, the data processing of this data is still underway, and can therefore not be evaluated at this time.

Several precipitation events occurred before the campaign which are all identified as snowfall events (Fig. 2a). It is not until after the campaign (16–18 December 2022) that a precipitation event occurs, where the total precipitation is not equal to the snowfall (i.e., a precipitation event with rainfall occurred). Hence, we do not expect a rain-on-snow (ROS) event (e.g., Stroeve et al., 2022) to have occurred and to impact the radar penetration observed along the under-flight. However, there is potential for snowfall events to have depressed the floes and caused flooding, which will impact the scattering horizon (i.e., scattering occurring not at the actual s-i interface but rather above it at the flooded interface, where snow-ice formation could occur), although C/S-band would not be expected to penetrate flooded snow either. However, Marshall et al. (2004) showed for ground-based observations over terrestrial snow that C-band was able to reflect at ground reflections in wet snow, whereas Ku-band was quickly attenuated, which could explain to some extent the limited penetration occurring at both Ku- and Ka-band. C- and S-band generally penetrate deeper than Ka- and Ku-band (with overall dry and cold conditions) due partly to the frequencies of the radar, and also have potential for penetrating and scattering within the ice cover itself. However, the impact of this is assumed to be negligible for the airborne snow radars and the utilised retrieval methodology, as the maximum scattering at C/S-band snow radars is assumed to be at the s-i interface (e.g., Kwok et al., 2017; Kurtz et al., 2013; Jutila et al., 2022b; Jutila and Haas, 2025; Kurtz and Farrell, 2011; Farrell et al., 2012). Other factors include blow-off of snow from the Filchner-Ronne ice shelf or the Antarctic peninsula through katabatic winds onto the floes may also have increased the snow accumulation on the floes by redistribution of the snow. Additionally, melting of the snowpack when temperatures reach close to positive degrees could lead to scattering further up in the snowpack and result in an detected interface closer to actual a-s interface along with the potential for superimposed ice to form in which meltwater form the surface will percolate the snowpack towards the s-i interface, resulting in an increased presence of liquid water. It is noticeable that a significant contribution (30–42% for Ku- and Ka-band, respectively, compared to 17% at C/S-band) of the maximum power occurs within 0.10 m from the lidar a-s interface, and that Ku-band MAX on average was reflected more than 0.1 m earlier than MAX in C/S-band.

Of particular interest over first-year ice is the impact of brine wicking and a saline snow cover on the radar wave's penetration. Even with the western Weddell Sea sector having been measured to hold up to 80% multi-year ice (Worby et al., 2008), the

overall ice conditions of the Southern Ocean primarily consist of first-year ice. When first-year ice forms, a small amount of brine is expelled upward, yielding a thin layer of brine on the ice surface, and with subsequent snow accumulation, an upward wicking of brine via capillary action (Barber et al., 1998; Mallett et al., 2024) into the basal snow layer occurs (Nandan et al., 2017). While this wicking process produces brine-wetted snow which primarily occurs at the base of the snow cover (bottom 6–8 cm) (Nandan et al., 2017), for thicker snow, a strong salinity gradient is commonly observed in bottommost layers and with low brine volumes or brine-free conditions in the uppermost snow layer (Fuller et al., 2014). Importantly, from a remote sensing perspective, is the altering effect of brine on the dielectric and microwave scattering properties of snow (Geldsetzer et al., 2009), in particular, the strong microwave attenuation within snow volume (Nandan et al., 2016). Large snow grains, due to relatively warm temperatures and coated with brine due to brine wicking, will serve as significant scattering centres from Ku-band radar waves, which could explain some of the limited penetration observed at at Ku-band and Ka-band (even when using MAX as s-i interface, see Fig. 7). Sporadic warming events can also amplify the porosity of the ice and its permeability which can further enable upward brine movement (Tucker III et al., 1992), and hinder complete microwave penetration to the s-i interface at higher frequencies.

In some of the first studies of airborne CryoVEx Ku-band observations, Hendricks et al. (2010) presented observations from ASIRAS over late-spring western Arctic sea ice and found that little to no penetration into the snow cover occurred over the ice pack, which they concluded showed similarity to Antarctic results found in Willatt et al. (2010) with some regional dependency (comparing results from Greenland Sea and Lincoln Sea). Here, range was re-tracked using a threshold-spline-retracker-algorithm, where the spline curve models the leading edge and is used to find the half power point corresponding to the mean surface (analogous to applying TMRA with 50%). Similarly to Hendricks et al. (2010), a dominating signal from the a-s interface at both Ka- and Ku-band is evident for our observations using TFMRA (at any threshold) showing that scattering from the a-s interface plays a significant role (Fig. 7), and further analysis is encouraged for the variety of airborne observations available, considering the different radar systems and their resolutions, as well as seasonal, regional, and hemispheric dependency. In addition, the inconsistency between laser and radar over otherwise expected identical scattering surfaces noted by Hendricks et al. (2010) as also observed here (i.e., over leads, see also Fig. 6) highlights the differences in dominating scattering surfaces for laser and radars. This further complicates the snow depth retrieval when using lidar and radar for different interfaces.

Our results present similar discrepancies between CWT and PEAK as in Jutila et al. (2022b), and similarly to Kwok et al. (2017), a critical question emerges regarding where on the snow radar waveform the a-s and s-i interfaces should be re-tracked to represent the snow depth correctly? More work is necessary on the snow radars to evaluate when the methods apply, to which extent tweaking of the different thresholds is necessary, how to align methods for different radar systems and campaign conditions (e.g., the impact of low-altitude or high-altitude flights, and Antarctic or Arctic conditions), and to understand to what extent the snow radars can fully be used on sea ice and compared to satellites. Our results also show that the assumption of zero-freeboard may apply for this campaign along the airborne observations (Fig. 6 for C/S band show strong reflections at or even below the surrounding lead surfaces along the selected subset). However, using the snow depth from C/S when the s-i interface occurs below the surrounding lead surface will result in negative freeboards. In particular, negative freeboards can

result in flooding of seawater, which can lead to snow-ice formation and additional intrusion of seawater into the snowpack as well as gradual drainage of brine within newly formed snow ice. Such aspects further limit microwave penetration depending on frequency, where Ku-band, and Ka-band in particular, will have limited penetration.

Beyond the microscale impact of the complex snow stratigraphy of the microwave signal, we also need to consider the larger macroscale surface heterogeneity and roughness. Kwok and Maksym (2014) presented a comparison between snow radar (2–8 GHz) Antarctic observations from 2010 and 2011 with derived roughness from airborne lidar (airborne topographic mapper, ATM) observations. Here, roughness was derived as the standard deviation of detrended elevations along a transect of 4 km. They found correlations between average snow depth and surface roughness at 0.77 and 0.84 depending on the years (2010/2011) and locations (Weddell Sea/Bellinghausen Sea), which they conclude is consistent with the fact that deformed ice tends to trap deeper snow. However, while their roughness estimate primarily supported the observation that snow tends to accumulate around obstacles, such a large-scale roughness parameter provides little insight into how this, on a point-to-point scale, may impact the retrieved airborne observations. They do note that the snow radar is not able to detect several interfaces over pressure ridges, a similar observation qualitatively made from our observations (e.g., see Fig. 6a), where the snow radar is incapable of retrieving snow depth over ridges due to only one single peak being present (as also showed as a waveform example presented in Kurtz et al., 2013).

Hendricks et al. (2010) evaluated airborne Ku-band observations and observed that deformed ice, and thus surface roughness, plays an important role in the retrieved elevations. This has later also been observed for spaceborne observations (e.g., Landy et al., 2020), which also cover a significantly larger area. Hence, roughness at various scales (large-scale deformation or at the scale of wavelengths) impacts the backscatter distribution – and hence the waveform – to an extent that is not well known. In a recent study, De Rijke-Thomas et al. (2023) hypothesise the impact of surface roughness and slope on the backscattering at Ku-band from airborne observations acquired over first-year ice near Eureka during the Arctic spring. They propose the mechanism, known as probabilistic quasi-specular scattering from the s-i interface, to play a significant role in determining the contributions to radar backscatter distribution at various scales. In particular, an aspect of interest here is the ratio of scattering mechanisms dependent on frequency and, in turn, on the antenna design. Antenna beam-width is a function of frequency (Table 1), which impacts the shape of the retrieved waveforms, and the possibility of detecting individual scattering contributions from different interfaces. The beam-limited footprint is directly proportional to the antenna beamwidth (see Eq. A8), meaning that the C/S-band radar in combination with its antenna beamwidth has a significantly larger beam-limited footprint (∼160–550 m) compared to both Ku- and Ka-band (ranging from ∼116 m down to ∼80 m), and that potential strong scatters from off-nadir may impact more significantly at C/S-band than at Ka-band. More importantly, the antenna beamwidth along with the individual surface scatters drives the shape of the retrieved waveform, with off-nadir scattering reflected in the trailing edge of the waveform, which could be picked up as s-i interface using the PEAK re-tracker in case of strong off-nadir reflections. Or, for snow radars, lower spikes representative of the a-s interface observed before the main spike are, in fact, system artefacts known as sidelobes (Kwok and Haas, 2015), which could impact the snow depth retrieval. However, pre-processing steps have been taken to minimise the impact of sidelobes (see Appendix A). Nonetheless, it is worth considering what the impact of antenna beamwidth in combination with frequency has on which interface scattering dominates, and how this is reflected in the

waveform shape. However to our knowledge, this has not been evaluated in depth for the radars utilised, and is out of scope of the current study.

From the inconsistent results between radars and retrieval methods presented herein, it is evident that limited conclusions about the most appropriate methods can be derived without the presence of coincident in situ observations. In order to untangle the scattering complexities of snow cover on sea ice, ground-based observations sampled in a manner emulating the airborne systems sensing conditions should be collected to validate the methods and the sensors independently. There is an urgent need for detailed field research and collection, hence, we urge future field campaigns to consider in situ components as a necessity to fully understand the capabilities and limitations of the instruments and their sensing abilities.

## 6    Conclusions

In this study, we examined airborne observations from a CRYO2ICE (CryoSat-2 and ICESat-2) under-flight carried out over the Weddell Sea on 13 December 2022 during the CRYO2ICEANT22 campaign. The airborne campaign carried an airborne laser scanner (ALS) and Ka-, Ku-, and C/S-band radars from which snow penetration and potentially snow depth could be derived. This is achieved by utilising four different re-trackers (TFMRA at 40%, 50%, and 80%; MAX; CWT; and, PEAK), with different re-tracking thresholds used and by questioning conventional assumptions, in particular regarding whether methodologies applicable to satellites are justified for airborne observations, and whether traditionally used re-trackers are retrieving what we expect.

At airborne scales, the a-s interface makes a significant contribution at both Ka- and Ku-band, and renders re-trackers traditionally applied on satellite scales unequipped for deriving the same surface on airborne scales. Furthermore, Ku-band showed incapability to reach the full extent of the snow radar (and therefore the s-i interface) in the majority of cases. The primary scattering horizon (MAX) at Ku- and Ka-band was reflected inside the snowpack or at the s-i interface in 60% and 46% of times, respectively, whereas at C/S-band MAX was reflected further down into the snowpack (expected at the s-i interface) in 74% of times. C/S MAX compared with ALS showed penetration down to 0.57 m on average (without accounting for slower-wave propagation speed). Ku-band MAX re-tracked on average 0.2 m before than C/S-band MAX, and Ka-band re-tracked even 0.1 m before Ku-band. In contrast, using TMFRA showed scattering at or before the a-s interface in approximately 40–70% of times at Ka- and Ku-band. Based on these results, future work on evaluating additional Ka- and Ku-band observations from different campaigns is encouraged to understand under which conditions this applies and to which extent we can track different interfaces. Several instances showed primary backscatter at or close to the a-s interface for Ku-band with distinguishable, but lower, scattering from the s-i interface, which could allow for retrieval of snow depth from Ku-band alone. However, a novel re-tracker to identify these peaks would be necessary, as it would need to not make assumptions on which interface contributes the most at the different frequencies. PEAK was applied as an example of such re-tracker, however several lower peaks were not identified with PEAK likely due to too strict thresholds. Additional tweaking of PEAK would be necessary to understand to what extend it can capture such multiple scattering in snow packs. Using TFMRA50 (as the a-s interface) and MAX (as the s-i interface) at Ku-band gave similar snow depth results as applying PEAK, supporting conclusions that (a) the a-s interface

contributes significantly and is distinguishable as separate peaks, and (b) if the main scattering at Ku-band represents the s-i interface, snow depth could be retrieved from such frequency alone on airborne scale.

The snow radar re-tracking algorithms (CWT and PEAK) show inconsistencies and discrepancies at C/S-band. CWT re-tracked the a-s interface before ALS in 80% of times, whereas PEAK was in the order of TFMRA for Ka- and Ku-band (around 30% of observations being reflected before or simply discarded due to no snow observations). PEAK s-i interface was on average scattered 0.05 m above MAX, whereas CWT s-i interface scattered 0.2 m before MAX on average. This is expected since PEAK can identify peaks before and after MAX as the s-i interface provided the peak fulfils the requirements. In comparison, CWT only tracks the location on the leading edge of a waveform as the s-i interface once it has exceeded a noise+clutter threshold.

Several inconsistencies are observed for the a-s interface determined by ALS and the radars and are likely caused by viewing geometry and different surface scattering mechanisms that dominate the signals. This was particularly evident over leads with broken floes. 5–10% of observations at all frequency bands were reflected before the a-s interface identified by ALS. This further complicates the use of lidar as the a-s interface and any re-tracked interface as s-i interface from radars. Care should be taken considering highly heterogeneous sea ice cover.

Additional work should be invested to evaluate the snow radar processing methods further and to consider at which point on the snow radar waveforms the a-s and s-i interface should be re-tracked to not only represent the snow depth but also align the location of the derived interfaces with the actual interfaces. The different methods available show large inter-variability in where the interfaces are re-tracked depending on waveform shape, and snow and ice conditions. For snow depth estimates, CWT on average measured 0.1–0.2 m thicker snow than PEAK. PEAK also shows a more strict requirement to successfully identifying snow depth, with more than 36% of observations discarded or within 0.05 m (identifying the a-s and s-i interfaces at the same peak resulting in 0 m of snow or discarding waveforms due to too much clutter/too many peaks). Using TFMRA provided limited snow depth information due to primarily scattering at the a-s interface. Using MAX at Ka- and Ku-band showed higher variability, although still limited penetration. Here, ALS and MAX at Ku-band (and Ka-band) was on average underestimated by at least 0.15 m compared to ALS and C/S MAX, and for several instances, Ka- and Ku-band reflected either within the snowpack or contributions from different interfaces and attenuation of the signal limits the full extent of penetration. Based on this, it appears that Ka- and Ku-band's maximum is retrieved from mid-way in the snow and upwards in the majority of times.

A significant shortcoming of this analysis is the missing validation of the methods and retrievals here, since there are no direct in situ observations available to compare against. Hence, with the significant discrepancies observed between the different methods and combinations of measurements, it is still not clear which combination or method provides observations most consistent with a ground truth. While we acknowledge the significant efforts required to ensure a successful field-based in situ component, and that often, such efforts are planned but due to unforeseen circumstances, cannot be successfully completed, we note that without in situ observations, we cannot independently validate the aerial observations, nor the satellites. This is especially true for under-represented locations such as the Southern Ocean and has a seasonal dependency that cannot

be overlooked. Thus, we urge the continuation of strategically planned field-based efforts necessary to validate the airborne
systems, consistently and throughout the season, to ensure that we can properly validate the satellite observations.

*Code and data availability.* CRYO2ICEANT22 campaign data are currently in final stages before approval from ESA and will, once approved, be published on the ESA campaign site and on www.cs2eo.org. The processed CRYO2ICEANT22 data presented here are available from DTU DATA at under DOI: 10.11583/DTU.26732227. The code for processing the provided CRYO2ICEANT22 data and the satellite/model data are available on the following GitHub repository (Fredensborg Hansen, 2024): https://github.com/reneefredensborg/cryo2iceant22-airborne-cryo2ice-weddell-sea-ice (last accessed: 11 September 2024).

ERA5 reanalysis data (Hersbach et al., 2023) are available from the Copernicus Climate Change Service (C3S) at DOI: 10.24381/cds.adbb2d47 (last accessed: 08 August 2024).

To generate Figure 1, the following data has been extracted: The daily output on 13 December 2022 of CASSIS in was provided by IRL via Andy Ridout (University College London, UCL). Currently, CASSIS model output from 1981–2021 (Lawrence et al., 2024) is available from http://www.cpom.ucl.ac.uk/cassis/ (last accessed: 09 July 2024, no DOI provided). AMSR-E/AMSR2 Unified L3 Daily 12.5 km Brightness Temperatures, Sea Ice Concentration, Motion & Snow Depth Polar Grids, Version 1 (Meier et al., 2018) is available under DOI: 10.5067/RA1MIJOYPK3P (last accessed 08 August 2024) on NSIDC.

## Appendix A: Pre-processing of CReSIS radar observations

Here, the various pre-processing steps of the radar observations are presented. This includes the signal processing applied to generate the radar waveforms used in the study (Appendix A1), the expected radar footprints and resolution (Appendix A2), and the collection of the radar waveforms to provide one track from the segmented data provided by CReSIS for one full flight (Appendix A3).

### A1 Signal processing of radar waveforms

Here, we briefly summarise the signal processing applied by CReSIS, also present in different README documents available on the CReSIS FTP server and their website at https://data.cresis.ku.edu/ (last accessed 2024/09/11). In addition, for a more in-depth description, the reader is referred to Panzer et al. (2013) and Yan et al. (2017). First, digital errors are set to zero (to minimise digital error effects on processing) and the GPS data is synchronised with the radar observations using the UTC time stored in the radar data files. Next, the analogue-to-digital-converter (ADC) input is used to convert from quantisation to voltage, and the direct-current-bias is removed by subtracting the mean of the amplitude. Next, quick-look waveforms are generated using unfocused SAR processing for a total of 16 coherent average including hardware and software averages. Next, a fast-Fourier transformation (FFT) is applied with a Hanning window to convert the raw data into the range domain (analogous to pulse compression), and the data are flipped around based on the Nyquist zone. A high-pass filter is applied in the along-track direction to remove coherent noise. Then, a 1 range-bin by 5 along-track-range-line boxcar filter is applied to the power detected data, and then decimated in along-track by 5. The quick-look output is used to find the ice surface location

(fully automated and available in the product, but not used for scientific data). Finally, the output is elevation compensated (with radar range bin accuracy) and truncated in fast time based on the data posting settings (here, truncation information is provided, see further processing with this in Section A3). Information about the radars are provided in Table 1, and the formulas relevant for deriving footprints and range resolution are presented in Section A2.

## A2 Resolutions and expected radar footprints

For a flat surface, the range resolution is:

$$\delta_R = \frac{k_t c}{2B\eta}, \tag{A1}$$

where $B$ is the bandwidth, $\eta$ is the index of refraction for the medium (for air, set to 1), $c$ is the speed of light in vacuum, and $k_t$ = 1.5 due to the application of a Hanning time-domain window to reduce the range sidelobes of the chirped transmit waveform. $B$ can be computed from the parameters $f_0$, $f_{mult}$, and $f_1$ in the radar observations by the following:

$$B = (|f_1 - f_0|) \cdot f_{mult}, \tag{A2}$$

and is 6 GHz for the current CReSIS radars.

Estimate of the along-track resolution before any coherent averaging can be derived similarly to the across-track resolution. However, a processing known as stacking, coherent averaging, or unfocused SAR processing is applied. The data is coherently averaged 16 times, and is decimated by the same amount. Thus, the along-track resolution ($\sigma_{\text{along-track}}$) is given by:

$$\sigma_{\text{along-track, SAR}} = H \tan\left(\sin^{-1}\left(\frac{\lambda_c}{2L}\right)\right), \tag{A3}$$

where $H$ is the nominal altitude, $\lambda_c$ is the center frequency, and $L$ is the SAR aperture length can be given as:

$$L = \sqrt{\frac{H\lambda_c}{2}}, \tag{A4}$$

Assuming all effects are accounted for when coherently averaging the data. However, since the data is only averaged coherently 16 times (sum of hardware and software averaging), we instead compute the SAR aperture length as:

$$L = \frac{nv}{\text{PRF}}, \tag{A5}$$

where $n$ is the number of averages, $v$ is the velocity of the aircraft, and PRF is the pulse-repetition-frequency. A factor of 5, equivalent to the number of incoherent integrations after unfocused SAR processing, must be accounted for when deriving the effective SAR aperture length ($L$) by dividing by 5.

For smooth or quasi-specular targets (such as e.g., interfaces), the primary response is from the first Fresnel zone. Thus, the
750 directivity of specular targets effectively creates the appearance of a cross-track resolution equal to this first Fresnel zone. The first Fresnel zone is a circle with a diameter given by:

$$\sigma_{\text{Fresnel-limited}} = \sqrt{2\left(H + T/\sqrt{3.15}\right)\lambda_c}, \tag{A6}$$

where $T$ is the depth in ice of the target (here, assumed to 0), and $H$ is the height above the air-snow interface (or here, the altitude). However, for rough surfaces with no clear layers, the cross-track resolution is constrained by the pulse-limited footprint, approximately given by:

$$\sigma_{\text{pulse-limited}} = 2\sqrt{\frac{\left(H + T(\sqrt{3.15})\right) ck_t}{B}} = 2\sqrt{\frac{k_t cH}{B}}.$$  (A7)

Finally, considering the beam widths and impact of range, the cross-track resolution be computed as beam-limited considering potential off-nadir reflections that could impact the signal:

$$\sigma_{\text{beam-limited}} = 2H\tan\left(\frac{\beta}{2}\right),$$  (A8)

where $\beta$ is the beam width in radians. Here, we use the beam width given for the lowest frequency of the radar since beam-width changes with frequency, and is the largest at lowest frequencies.

## A3   Processing of CReSIS radar segments into one under-flight track

The CReSIS data are available as .mat files grouped into segments, where one segment is a continuous data set where the radar settings do not change. One day can be divided into segments if the radar settings were changed, hard drives were switched, or other operational constraints required the radar recording to be turned off and on. In addition, each segment is broken into frames (analogous to satellite SAR scenes) to make analysis of the data easier. Frames span 33 seconds covering approximately 4–5 km dependent on the aircraft speed. As such, one under-flight will have several frames spanning the segments and the entire under-flight. The CReSIS parameters of interest to this study include: radar waveforms (*Data* in the CReSIS product), flight altitude (*Elevation*), GPS time (*GPS_time*), heading (*Heading*), pitch (*Pitch*), and roll angle (*Roll*) of the aircraft, latitude (*Latitude*), longitude (*Longitude*), two-way-travel-time for a full waveform (*Time*), and various parameter records (such as *f1*, *f0*, and *fmult* which are necessary to run a re-tracking algorithm for the snow radar). It may be that the data are truncated (focused only around a clear surface signal to limit data size) where the waveform parameter can be truncated both before and after the identified surface signal. If that is the case, the parameter *Truncate_Bins* is activated and shows where in the *Time* parameter the waveform is relative to. This is important to incorporate since if truncation has been activated, the size of the waveform (number of range bins) will differ depending on frames. This plays a role when creating one file for the entire flight as the frames of one flight will change in size (of the waveforms) depending on the truncation occurred and the surface encountered, and therefore, the data must be padded somehow to ensure the same parameter size. Furthermore, this allows us to properly estimate the range to a surface depending on the re-tracked waveform. Whenever the roll angle is more than $|2°|$, the range observation is set to not-a-number.

**Table B1.** Lever arms (*x*, *y*, *z*) in metres for the different instruments. *x* is pointing to nose, *y* is pointing to the right wing and *z* is pointing downwards. For corrections, the arithmetic averages of the transmit and receive offsets were used as the offsets of radar phase centres with respect to the GPS. Only $\Delta$ in *z*-axis is used to correct when aligning the ranges for visualisation purposes in Fig. 2a, the leverarms have otherwise been corrected for in the actual data.

|  | C/S-band | Ku-band | Ka-band |
|---|---|---|---|
| Transmit (TX) | (-2.62, 0.38, 2.69) | (-2.95, 0.75, 2.71) | (-2.86, 0.69, 2.77) |
| Receiver (RX) | (-3.20, 0.73, 2.69) | (-2.88, 0.35, 2.71) | (-2.95, 0.41, 2.77) |
| $\mu$(TX, RX) | (-2.91, 0.55, 2.69) | (-2.91, 0.55, 2.71) | (-2.91, 0.55, 2.77) |
| $\Delta\mu_{Ku} - \mu_{Ka\ or\ C/S}$ | (0.00, 0.00, 0.02) | — | (-0.01, 0.00, -0.05) |

## Appendix B:  Lever arms

For position estimation, failing to compensate for the lever arm from the IMU to the GNSS antennas yields an attitude-dependent error (Stovner and Johansen, 2019). Lever arms, which are known locations of the GNSS receivers relative to the IMU, must therefore be used to correct the observations, and are provided in Table B1.

*Author contributions.*  Conceptualisation: RMFH, HSK. Data curation: RMFH, JL, FRM, HSK, IRL. Formal Analysis: RMFH, HSK. Funding acquisition: HSK, KVH, RF, JW. Investigation: RMFH, HSK, AJ, IRL, SBS, GV. Methodology: RMFH, HSK, AJ, IRL, AS, KVH, ER. Project administration: HSK, RF, TGDC, JW. Software: RMFH. Supervision: HSK, ER, KVH, RF, JW, TGDC. Validation: RMFH. Visualisation: RMFH. Writing – original draft: RMFH. Writing – review & editing: All authors.

*Competing interests.*  The authors declare no competing interests.

*Acknowledgements.*  This work was supported by ESA with DTU Space CRYO2ICEANT 2022 – Technical Support for the Antarctica CRYO2ICE 2022 Experiment (#4000141420/23/NL/IB/ab) and the UK Natural Environment Research DEFIANT project (#NE/W004747/1) RMFH and HSK acknowledge the great discussions of the International Space Science Insitute (ISSI) Team 501 - "Multi-Sensor Observations of Antarctic Sea Ice and its Snow Cover". RMFH was supported by the Nordic5Tech joint PhD-alliance research project between DTU and NTNU to characterise extreme sea ice features with a combination of remote sensing, in-situ data, and physical modelling. AJ was supported by the Research Council of Finland grant #341550.

We acknowledge the use of data and/or data products from CReSIS generated with support from the University of Kansas, NASA Operation IceBridge grant NNX16AH54G, NSF grants ACI-1443054, OPP-1739003, and IIS-1838230, Lilly Endowment Incorporated, and Indiana METACyt Initiative.

We thank the crew of the Rothera research station and the pilots who made the CRYO2ICEANT2022 campaign possible.

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

# Multi-frequency altimetry snow depth estimates over heterogeneous snow-covered Antarctic summer sea ice – Part II: Comparing airborne estimates with near-coincident CryoSat-2 and ICESat-2 (CRYO2ICE)

Renée M. Fredensborg Hansen[1,2,3], Henriette Skourup[1], Eero Rinne[3], Arttu Jutila[4], Isobel R. Lawrence[5,6], Andrew Shepherd[6], Knut V. Høyland[2], Jilu Li[7], Fernando Rodriguez-Morales[7], Sebastian B. Simonsen[1], Jeremy Wilkinson[8], Gaelle Veyssiere[8], Donghui Yi[9], René Forsberg[1], and Taniâ G. D. Casal[10]

[1]Department of Geodesy and Earth Observation, The National Space Institute (DTU Space), The Technical University of Denmark (DTU), Kgs. Lyngby, Denmark
[2]Department of Civil and Environmental Engineering, Norwegian University of Science and Technology (NTNU), Trondheim, Norway
[3]Department of Arctic Geophysics, The University Centre in Svalbard (UNIS), Longyearbyen, Norway
[4]Finnish Meteorological Institute (FMI), Helsinki, Finland
[5]ESA – ESRIN, European Space Agency, Frascati, Rome, Italy
[6]Centre for Polar Observation and Modelling, Department of Geography and Environmental Science, Northumbria University, Newcastle, UK
[7]University of Kansas, Center for Remote Sensing and Integrated Systems (CReSIS), USA
[8]British Antarctic Survey, UK
[9]GST Inc., Laboratory for Satellite Altimetry, Center for Satellite Applications and Research, NOAA, College Park, MD, USA
[10]ESA – ESTEC, European Space Agency, Noordwijk, Netherlands

**Correspondence:** Renée M. Fredensborg Hansen (rmfha@dtu.dk)

**Abstract.** For the first time, a comparison of altimetry-derived snow depth estimates between dual-frequency spaceborne and near-coincident multi-frequency airborne estimates is conducted using data from the recent under-flight of a CryoSat-2 and ICESat-2 (CRYO2ICE) orbit by a simultaneous airborne campaign over the Weddell Sea in December 2022 carrying Ka-, Ku-, C/S-band radars and a scanning near-infrared lidar. From this unique combination of airborne sensors, the accuracy of snow depth captured by the near-coincident CRYO2ICE orbits can be evaluated. The CRYO2ICE snow depth achieved along the orbit was, on average, 0.34 m, which is within 0.01 m from passive-microwave-derived observations and 0.12 m from a model-based estimate. The retrieval methodology appears to play a significant role, which we suspect is highly dependent on the classification and filtration schemes applied to remove potentially ambiguous altimetry observations. Comparison with airborne snow depths at 25-km segments showed correlations of 0.51–0.53, a bias of 0.03 m and root-mean-square-deviation of 0.08 m when using the airborne lidar scanner as air–snow interface and C/S-band at maximum amplitude at the snow-ice interface. To understand how comparisons across ground, air and space shall be conducted, especially in preparation for the upcoming dual-frequency radar altimeter mission Copernicus Polar Ice and Snow Topography Altimeter (CRISTAL), it is critical that we

investigate the impact of different scattering mechanisms at varying frequencies, for diverging viewing geometries considering dissimilar spatial and range resolutions.

## 1 Introduction

With its insulating properties and its regulating role in sea ice growth and melt, snow on sea ice plays a pivotal role in the climate system (Webster et al., 2018; Sturm et al., 2002). Snow is considered a key component in the sea ice thickness retrieval methodology using satellite altimetry under the assumption of hydrostatic equilibrium, thus it is critical that accurate and timely large-scale estimates of snow depth are available. For Antarctic sea ice, this is especially critical considering the multi-layered, complex, and heterogenous snow cover present due to the relatively warm air temperature coupled with strong winds, heavy precipitation and relatively thin ice (Massom et al., 2001). Such exceptional conditions result in significant snow metamorphism occurring, limiting microwave radar penetration into snow and further complicating the radar signals observed. The relatively warm temperatures can enlarge snow grain sizes, create internal ice layers, and ensure the presence of liquid water within the snowpack (Webster et al., 2018). Considering the predominantly seasonal and relatively thin ice cover (Worby et al., 2008), capillary brine wicking into the basal snow layer and increasing salinity of the snow cover can further limit microwave penetration (Nandan et al., 2017). These conditions permit the survival of a year-round, highly complex, and diversified snow cover (Arndt and Paul, 2018).

Recent studies (e.g., Garnier et al., 2021; Lawrence et al., 2018; Kwok and Maksym, 2014; Kacimi and Kwok, 2022) have utilised dual-frequency or multiple-interface-tracing approaches to derive snow depth from altimetry by using a measure of the air-snow (a-s) and snow-ice (s-i) interfaces. Frequently, the a-s interface has been observed using laser observations that reflect at (or very close to) the snow surface (Kacimi and Kwok, 2022, 2020), where freeboard (elevation of ice above local water level) observations computed from lasers are referred to as the total freeboard (snow + ice). Since its launch in 2014, observations from a high-frequency satellite radar altimeter operating at Ka-band have also been used as a measure of a-s interface based on the assumption of limited penetration into snow (Garnier et al., 2021; Guerreiro et al., 2016). The s-i interface has usually been observed by use of Ku-band radars, where laboratory experiments have shown that Ku-band signals can penetrate to the s-i interface under cold and dry conditions (Beaven et al., 1995). Thus, Ku-band observations are generally assumed to penetrate the snow cover, hence providing radar freeboards which, when corrected for slower wave propagation speed, are converted to sea ice freeboards (e.g., Hendricks, 2022; Rinne and Hendricks, 2023; Mallett et al., 2020). However, this assumption has been disputed in several studies (e.g., Nab et al., 2023; Willatt et al., 2023, 2010, 2011; De Rijke-Thomas et al., 2023; Armitage and Ridout, 2015; Rösel et al., 2021; King et al., 2018; Nandan et al., 2017, 2020, 2023) based on data from ground-based, airborne, and spaceborne observations. Here, they argued that phenomena such as snow metamorphism, redistribution, brine wicking, or flooding, can significantly limit the penetration of Ku-band waves. However, even based on observations showing that Ku-band penetration can be limited, the dual-frequency approach remains one of the few Earth observation methods expected to observe snow depth over sea ice under optimal conditions. These assumptions are pivotal for the future dual-frequency Ka/Ku-band polar radar altimetry mission, Copernicus Polar Ice and Snow Topography Altimeter

(CRISTAL), planned for launch in 2027/2028 (Kern et al., 2020), where the dual-frequency approach is one of the driving factors of the satellite design. To investigate the simultaneous dual-frequency approach further, ESA altered the orbit of their polar Ku-band radar altimetry mission, CryoSat-2, in July 2020 to align periodically with NASA's polar laser altimetry mission, the Ice, Cloud and land Elevation Satellite-2 (ICESat-2) for the Northern Hemisphere, known as the CRYO2ICE Resonance Campaign. This alignment was further adjusted in July 2022 to maximise orbits in the Southern Hemisphere.

Currently, spaceborne dual-frequency altimetry for snow depth estimation has been relatively widely used in the Arctic (e.g., Lawrence et al., 2018; Guerreiro et al., 2016; Garnier et al., 2021; Kacimi and Kwok, 2020; Fredensborg Hansen et al., 2024a), with few attempts to apply it in the Antarctic (e.g., Kacimi and Kwok, 2022; Garnier et al., 2021) due to the challenging snow conditions. Furthermore, most studies on satellite altimetry dual-frequency approaches rely on monthly-based, gridded snow-depth-composites rather than orbit-wise estimates. Fredensborg Hansen et al. (2024a) presented the first estimates of satellite-derived dual-frequency snow depth along orbits, however a direct validation of these observations were limited to few in situ buoys available. Fortunately, in December 2022, the CryoSat Validation Experiment (CryoVEx) programme with ESA in collaboration with the Natural Environment Research Council (NERC) Drivers and Effects of Fluctuations in sea Ice in the ANTarctic (DEFIANT) project completed an airborne campaign over Antarctic land and sea ice. Here, they under-flew a CRYO2ICE orbit on 13 December 2022 carrying a full suite of instruments relevant for estimating snow depth over sea ice and evaluating microwave penetration with the use of different sensors, including C/S-("snow radar"), Ka- and Ku-band radar altimeters. This campaign presents the first opportunity to compare near-coincident dual-frequency satellite-derived snow depth estimates with simultaneous airborne snow depth estimates derived from multi-frequency approaches.

Here, we present the second part of a two-part study, where we compare airborne snow depth estimates derived based on the traditional hypothesis of microwave penetration into snow (Part I, see Fredensborg Hansen et al., 2024b) with near-coincident spaceborne radar (CryoSat-2) and laser (ICESat-2) orbits along a dedicated CRYO2ICE orbit. This is currently the only CRYO2ICE (or any dual-frequency satellite orbit) validation under-flight carried out with a full suite of instruments able to evaluate penetration into the snow. In particular, this study aims to evaluate the capabilities of CRYO2ICE (emulating the future CRISTAL dual-frequency polar satellite altimetry mission) to estimate snow depth along-track similar to what the airborne sensors observe. Of particular interest is the unique combination of airborne sensors (at C/S-, Ku-, and Ka-band) from which different information on the snow conditions might be leveraged.

## 2 Data

### 2.1 ICESat-2 and CryoSat-2 (CRYO2ICE)

#### 2.1.1 ICESat-2

ICESat-2 carries a photon-counting Advanced Topographic Laser Altimeter System (ATLAS) that transmits green laser pulses (532 nm) split into a six-beam measurement configuration separated into three beam pairs (gt1–3). Each beam pair carries a strong and a weak beam, and they are denoted as left (gtl) or right (gtr) beams relative to the satellite orientation (Neumann

et al., 2019). For this particular track, the ICESat-2 spacecraft was in forward orientation, which resulted in gtl beams being designated as strong beams and gtr as weak. Each laser pulse is emitted with a pulse-repetition-frequency of 10 kHz, leading to a 0.7 m along-track separation that, with a footprint of 11–17 m at nominal altitude (Magruder et al., 2020), allows for oversampling and unprecedented surface sampling. The operational Sea Ice Heights product (ATL07, Kwok et al., 2023a) aggregates 150 photons into surface segments and identifies the segments as floes or leads based on a radiometric classification. These 150-photon aggregates result in inconsistent segment-lengths based on surface reflectivity and specularity. From ATL07, along-track freeboards are calculated in the ATL10 product based on a reference sea ice surface derived from the available lead (or sea surface height) segments. For ATL10 (Kwok et al., 2023b), a sea surface height reference is provided for consecutive 10-km along-track sections where at least one sea surface sample (lead) is available for each beam independently. Hereafter, the sea surface height segment elevation is subtracted from the floe heights to provide total freeboards. Negative total freeboards are set to 0. For this study, ATL07 and ATL10 release 006 (r006) are used. ATL07 is used only for visualisation purposes, whereas ATL10 is used for processing. ICESat-2 is sensitive to the presence of clouds and fog, altering the location of the reflection interface due to limited penetration, resulting in gaps along the track (e.g., Fredensborg Hansen et al., 2021; Petty et al., 2021).

### 2.1.2 CryoSat-2

CryoSat-2 carries the interferometric Ku-band SAR altimeter (SIRAL), which primarily acquires observations in SAR mode over sea ice. In SAR mode, the radar waveform is computed through delay-Doppler SAR processing of 64 individual bursts, which are multi-looked to provide a waveform for each Doppler beam with a footprint of $\sim$300 x 1,650 m sampled at 20 kHz (Scagliola, 2013). Due to the various processing chains used to derive radar freeboards from CryoSat-2, we introduce three different products used to evaluate the sensitivity of different retrieval methods. While different geophysical corrections might be applied across the various freeboard products, we do not assume these to play a significant role, since freeboards are relative observations and the impact is in the order of centimetres, primarily related to lead-sparse areas (Ricker et al., 2016). The processing chains and products are briefly described below. For all products, the radar freeboard is derived by differencing an interpolated sea level anomaly, using observations classified as leads as tie-points, from the observations classified as originating from ice floes.

- *ESA Baseline-E (ESA-E):* The operational ESA-E observations in SAR mode provide radar freeboard by a combined waveform re-tracker: for diffuse waveforms, which are expected to originate from floes, a 70% threshold-first-maximum-retracker-algorithm (TFMRA) is applied, and for specular echoes (i.e., leads), a peak-finder based on the model-fitting method described in Giles et al. (2007) is implemented. Leads and floes are discriminated based on a combination of parameters including sea ice concentration, waveform peakiness, standard deviation of the stack of waveforms, and peakiness of the stack (Meloni et al., 2020), following methods presented in Passaro et al. (2018). ESA-E radar freeboards used here are pre-processed following the methodology presented in Fredensborg Hansen et al. (2024a, their Supporting Information S1) to remove erroneous observations.

– *ESA CryoSat ThEMatic PrOducts (CryoTEMPO) over sea ice:* The CryoTEMPO products aim to deliver simplified, harmonised, and agile CryoSat-2 products that are easily accessible and usable by non-altimeter experts and end-users with the possibility of evolving the processing chain depending on the current state-of-the-art. The CryoTEMPO sea ice thematic product (Hendricks, 2022) contains geolocated and time-associated radar freeboards and sea ice freeboards with associated uncertainties. The surface elevations are derived using Sar Altimetry MOde Studies and Applications over ocean (SAMOSA+) algorithm with an unrestricted waveform classification allowing retrieval of freeboards over mixed surfaces and thinner floes (personal communication, Stefan Hendricks, 2024). The radar freeboard is converted to sea ice freeboard (not used in this study) with an outlier restriction of $-0.25$ to $2.25$ m applied, where identified outliers are removed from both parameters for consistency. The reader is referred to Hendricks (2022) for further information.

– *Fully-focused SAR (FF-SAR):* The FF-SAR technique performs the coherent processing of the radar echoes during the whole illumination time of a surface scatterer (Egido and Smith, 2017, 2019). This processing enhances surface resolution, enabling the discrimination of smaller features on the sea-ice surface, potentially leading to improved freeboard estimation. Surface elevation information is retrieved from multi-looked FF-SAR waveforms with a physical sea-ice retracker (Egido et al., 2022). Waveforms are classified as lead, floe, or ambiguous ice based on the specular versus diffuse power ratio (often denoted as SDR).

## 2.2 CRYO2ICEANT22 airborne campaign and sensors

With the continuation of the ESA validation programme CryoVEx under the Cryo2IceEx and NERC defiant projects in December 2022, the first under-flight of a CRYO2ICE orbit carrying radars possible to evaluate microwave penetration was completed. This airborne campaign (dubbed CRYO2ICEANT22) captured a dedicated CRYO2ICE orbit over sea ice in the Weddell Sea on 13 December 2022. The full survey flight took place from 15:52 UTC to 00:29 UTC (following day), with the satellite orbit under-flight occurring at ∼18:48–21:46 UTC, ICESat-2 (orbit number 23676, reference ground track 1283) passing at ∼17:36 UTC and CryoSat-2 (absolute orbit number 67222) at ∼20:16 UTC. The airborne campaign was aligned with the CryoSat-2 predicted orbit, and all observations were within 60 m from the CryoSat-2 observations with ∼80% within 20 m (not shown).

The surveys were carried out using a British Antarctic Survey (BAS) DASH-7 aircraft from Rothera Research Station with CReSIS Ka- (32–38 GHz), Ku- (12–18 Ghz) and C/S-band (2–8 GHz) radar altimeters, and an airborne laser scanner (ALS) of the type Riegl LMS Q-240i-80. An in-depth description of the airborne sensors and associated data processing can be found in Fredensborg Hansen et al. (2024b, Part I). Briefly summarised, the Ku- and Ka-band radar altimeters are assumed to have a primary scattering from a single surface (assumed to be the snow-ice and air-snow interface, respectively), which are assumed to be trackable using the threshold-first-maximum-retracker-algorithm (TFMRA). Here, different re-tracking thresholds (40, 50 or 80%) are used to track different locations of the leading edge of the radar waveform which have commonly been used to best represent the "average" surface elevation within a footprint. The C/S-band radar (snow radar) is assumed to have scattering from both the air-snow and snow-ice interface trackable from single waveforms. These interfaces are retrieved using two algorithms: continuous wavelet (CWT; Newman et al., 2014) or peakiness (PEAK; Jutila et al., 2022). For all three radars,

the location of the maximum power (MAX) on the waveform is extracted to represent the dominant scattering interface. The swath available from ALS is used to extract a vertical nadir profile emulating the a-s interface at the location of the radars. This is achieved by averaging all ALS observations within a 5 m diameter footprint at each radar observation.

There was no in situ component carried out over sea ice, hence ground-truth data describing the snow conditions are unavailable. We have also not been able to identify other reference observations (e.g., buoys) available to provide input on the conditions.

## 2.3 Auxiliary data

Ideally, for the validation of snow depth derived from the air- and spaceborne Ka/Ku-combinations, one would use the snow radar onboard the same platform. However, the methodologies currently used to derive snow depth from snow radars have been validated primarily for the Arctic (Kwok et al., 2017a; Jutila et al., 2022; Kurtz and Farrell, 2011); or, applied during winter conditions (October) in the Weddell and Bellingshausen Seas (Kwok and Maksym, 2014; Kwok and Kacimi, 2018) for campaigns flown in 2010–2016, and may not represent the sea ice conditions during this campaign. Therefore, we also compare the CRYO2ICE results with additional snow depth estimates (passive-microwave-derived or modelled).

### 2.3.1 AMSR2 passive microwave radiometer-derived snow depth

We use the passive microwave-derived snow depth product by (Meier et al., 2018). The snow depth estimate is derived from AMSR2 observations using the methodology of Markus and Cavalieri (1998) and is provided in a polar stereographic projection at a 12.5 km resolution as a daily estimate of a 5-day average. The same method is applied to both hemispheres but excludes perennial areas of the Arctic (Meier et al., 2018). The algorithm is only applicable to dry snow. Large temporal fluctuations can occur due to thaw-freeze events, where wet snow during the day refreezes during the night resulting in large grain sizes. This, in turn, leads to reduced emissivity at higher frequencies, which can result in an overestimation of snow depth (Meier et al., 2018). For this study, we extract the nearest neighbouring AMSR2 observations along the CRYO2ICE observations for comparison.

### 2.3.2 CASSIS snow depth

Centre for Polar Observation and Modelling (CPOM) Antarctic Snow on Sea Ice Simulation (CASSIS) simulates the daily creation and drift of floes, where the floes accumulate snow from the atmosphere and Antarctic ice sheet and lose snow to the ocean and snow-ice formation (Lawrence et al., 2024). Here, they utilise a Lagrangian framework to accumulate snow on top of floes while taking into account katabatic snowfall from ice shelves and snow-ice formation by assuming 55% of the daily snow accumulation submerges and refreezes. The CASSIS model is provided at a daily resolution in a polar stereographic grid with a tangential plane at 70°S at 10 km-grid resolution and the Lagrangian points are initiated on the 21 February every year with ship-based observations collected within the Antarctic Sea-Ice Processes and Climate (ASPeCt) program. We extract the nearest-neighbouring CASSIS observation along the CRYO2ICE observations for comparison.

 **3   Methodology**

### 3.1   Collocation of CRYO2ICE observations

For the collocation of CryoSat-2 and ICESat-2 freeboard observations along a CRYO2ICE orbit, we follow the methodology presented in Fredensborg Hansen et al. (2024a). In summary, one uses the observed CryoSat-2 radar freeboards as the baseline for collocating ICESat-2 observations. Here, we compute this individually for all three processing chains of CryoSat-2 data
(ESA-E, FF-SAR, and CryoTEMPO). All ICESat-2 total freeboard observations (from all three beam pairs including both strong and weak beams) within a search radius of 3.5 km from the observed CryoSat-2 radar freeboard are selected. An average inverse-distancing-weighted ICESat-2 total freeboard observation per CryoSat-2 radar freeboard is then computed. At each CryoSat-2 observation, smoothing is applied using the same search radius (analogous to a smoothing along-track window of 7 km) to reduce noise.

An example of the spatial scales to consider is presented in Fig. 1. Here, it is evident that ICESat-2 and CryoSat-2 are not fully aligned along the track i.e., for this part, the best alignment between ICESat-2 and CryoSat-2 occurs with the left beam pair (gt1r/l), which accounts for 87% of CRYO2ICE observations along the entire track (Fig. 1e). Towards the pole, the remaining 13% aligns with the centre beam pair. This is also illustrated in Fig. 1a, particularly in the northern part of the transect where the beams are positioned on the easternmost side of the ALS swath, whereas the opposite is the case at the southernmost
part of the transect. In addition, it is clear that applying a 3.5 km search radius incorporates observations from neighbouring beam pairs when they are in the vicinity. Figure 1d shows the average distance between the closest ICESat-2 beam pair and CryoSat-2 varies between ~2,000 and 3,000 m with a minimum distance of less than 1 meter, where highest coincidence with one of the beams occur. Since a smoothing radius is applied, each CRYO2ICE ICESat-2 weighted- average-freeboard includes observations before and after the CryoSat-2 observation along the track (see also Fig. 1 for the spatial scales). The reader is
referred to Fredensborg Hansen et al. (2024a) for further details and comparison over Arctic sea ice.

### 3.2   Snow depth retrieval from dual-frequency approaches

Similarly to former dual-frequency snow depth studies (e.g., Kwok et al., 2020; Lawrence et al., 2018; Garnier et al., 2021; Fredensborg Hansen et al., 2024a), snow depth is retrieved by differencing the height of the air-snow interface (total freeboard from laser) and the snow-ice interface (represented by Ku-band radar freeboard assuming full penetration) while accounting
for delay in radar wave propagation speed by use of the refractive index of snow considering its density. Here, we utilise a bulk density of snow constant at 300 kg m$^{-3}$. The methodology is described in detail in Part I.

### 3.3   Accounting for the spatial scales through smoothing and segmentation

For comparison with the CRYO2ICE spaceborne data, we first average the airborne observations into 1 km segments following the method presented in Jutila et al. (2022). These 1 km average segments are used to smooth the high-resolution spatial
snow depth variability that cannot be obtained from the Doppler-beam footprints of CryoSat-2. From these 1 km segments, we

further derive CRYO2ICE-comparable snow depths using the same along-track smoothing search radius of 3.5 km. In addition, we segment both the CRYO2ICE snow depths and the airborne counterparts into 25 km segments, following the approach of Garnier et al. (2021) and aligning with the CRISTAL requirements (Kern et al., 2020).

## 4  Results

### 4.1  Evaluation of radar and total freeboard along CRYO2ICE orbit

First, it is important to note the amount of observations removed from processing when computing freeboards. This processing includes removal of ambiguous data through surface classification, errors in surface retrieval, filter by set thresholds, and distance to closest leads. For example, ESA-E provides 3,269 sea surface height (or ellipsoidal height) observations along the full track (where the surface has been identified as sea ice). In comparison, FF-SAR provides instead 10,785 observations. However, the individual processing chains applied to ESA-E, FF-SAR and CryoTEMPO resulted in only 440, 286 and 2,212 radar freeboard observations, respectively, considering only radar freeboards within 3 m of the water level. Of these, 29.77%, 24.83%, and 32.10% of the observations, respectively, had radar freeboards below $-0.1$ m. Here, we remove freeboards below $-0.1$ m (as opposed to Fredensborg Hansen et al. (2024a) who kept all negative freeboards over Arctic sea ice), since they present a significant contribution to a smoothed CRYO2ICE CryoSat-2 freeboard (compared with their $\sim$5% negative freeboards). This results in 309, 215, and 1,502 radar freeboard observations for ESA-E, FF-SAR, and CryoTEMPO, respectively, to apply the smoothing window on ($\sim$7 km, or search radius of 3.5 km following Fredensborg Hansen et al., 2024a). For further data processing, we consider only the filtered and smoothed CryoSat-2 products.

Comparing the freeboards along-track (Fig. 2), it is evident that both ESA-E and FF-SAR achieve similar results at the few points available along track (average of $0.31 \pm 0.19$ m and $0.29 \pm 0.15$ m, respectively), whereas CryoTEMPO has significantly lower average radar freeboard (at $0.07 \pm 0.1$ m). Furthermore, CryoTEMPO provides significantly more observations than any of the other processing chains (>4 times). FF-SAR and ESA-E generally provide observations where mostly highly diffuse waveforms are present (see Fig. 2a), whereas CryoTEMPO is able to generate freeboards over less diffuse waveforms likely caused by thinner sea ice and/or a heterogeneous ice cover with significant dominance of specularity. This is probably due to the different surface classification schemes applied, where ESA-E and FF-SAR apply more strict requirements to their waveforms. This also explains the ability to extract much smaller freeboards (of, on average, 0.07 m, and where 931 of the 1,502 (or 62%) filtered radar freeboard observations were below the average value, with 453 of these 931 observations being positive). CryoTEMPO's average radar freeboard ranges close to a zero-freeboard assumption (that the sea ice surface, where Ku-band under dry and cold conditions is reflected from, is at the water level), although were we to apply such assumption to the ICESat-2 total freeboards for snow depth retrieval, the average snow depth would increase by 0.05 m on average (from $0.34 \pm 0.16$ m to $0.39 \pm 0.16$ m). Thus, this suggests that (a) the floes could have been flooded and that the first 0.05 m (on average) represents wet snow, where snow-ice formation could occur under freezing conditions, (b) other snow metamorphism prohibiting the full penetration at Ku-band is occurring, or (c) the zero-freeboard assumption does not hold true for this track – or at least part of the track. Applying such an assumption basin-wide can have significant impacts on retrieved sea ice thickness (Kwok and Kacimi,

2018). However, along the track (Fig. 2b), the snow depth of ICESat-2 with zero-freeboard assumption strongly follows the CryoTEMPO snow depths and has a correlation of 0.85. In comparison, the correlations between snow depths derived using CRYO2ICE ESA-E and FF-SAR with ICESat-2 at zero-freeboard assumption are 0.55 and 0.69, respectively. Here, since ESA-E and FF-SAR have primarily re-tracked thicker ice (higher radar freeboards, i.e. non-zero freeboard assumption valid), the snow depth is driven in part by the variability of the CryoSat-2 radar freeboards.

## 4.2 Comparison of CRYO2ICE snow depth with other composites

Comparing the derived CRYO2ICE snow depths (here, we use only CryoTEMPO for further evaluation due to the limited data coverage of ESA-E and FF-SAR) with other snow depth composites (CASSIS and ASMR2, see Fig. 3, see Fig. 1 in Part I for a zoom-in), it is evident that CRYO2ICE radar observations vary more (standard deviation of 0.16 m compared to 0.03 and 0.08 m for CASSIS and AMSR2, respectively), see Fig. 2b. In comparison, both CASSIS and AMSR2 are smoother along the transect, indicating that the spatial variability captured by ICESat-2 along the track is smoothed when using all the observations within the $10 \times 10$ km (or $12.5 \times 12.5$ km for ASMR2) resolution of the gridded products. On average, CASSIS snow depth is higher than CryoTEMPO CRYO2ICE snow depth by 0.12 m along the full extent of the CRYO2ICE track (763 valid observations). In contrast, AMSR2 agrees within a centimetre of the CRYO2ICE observations on average. However, it is noteworthy that between $-70°$ N and $-71°$ N, AMSR2 follows a similar drop in snow depth as observed by CRYO2ICE. This suggests that both Earth-observation methods are sensitive to the snow conditions here, which is not well captured by the model. Here, one might argue that either the dielectric conditions of the surface limit full retrieval of the snow depth, that AMSR2 is not fully tuned to the snow conditions of the Antarctic and thus not able to capture this information (which CRYO2ICE is unable to as well), or that the model has not fully captured the different processes that may alter the snow depth (such as snow metamorphism or re-distribution). Another drop is observed between $-72°$ N and $-73.5°$ N by CRYO2ICE, this is however not captured to the same extent by AMSR2. In addition, the AMSR2 product is intermittent, especially toward the ice edge in the north (see Fig. 1 in Part I). This can be due to either liquid water in the snowpack or snow metamorphism, which might explain the drop in snow depth caused by a change in dielectric conditions of the surface, or it can be caused by the presence of multi-year ice (Rostosky et al., 2018). A visual comparison with operationally produced maps of multi-year ice concentration (Melsheimer et al., 2019) from before the austral summer and directly after (when available) supports a presence of multi-year ice in this region. However, the method notes caveats with misclassification over rough first-year and young ice, or when snow wetness or metamorphism occurs. Due to such caveats, the product is not available during the austral summer and, therefore, also not during the airborne campaign.

In addition, one may also question the freeboards obtained by CryoTEMPO and whether these are representative of the ice conditions within one CryoSat-2 footprint, or whether the ice conditions are too heterogeneous to derive a meaningful freeboard. While contributions of such ice conditions may not be distinguished in the CryoSat-2 waveforms due to the range resolution ($\sim$0.47 m), a more restrictive classification scheme could remove such waveforms if confidence in the estimated freeboards is low and they are considered ambiguous. Additional work is needed to identify if such a classification scheme is necessary for the CryoTEMPO product. However, the recent study by Müller et al. (2023) has shown the possibility of

**Table 1.** Percentage of snow depth ($h_s$) observations, identified as floes, either at not-a-number (NaN), above 1.5 m or below $-0.05$ m for the different combinations of airborne snow depth retrievals used in the comparison against satellite observations.

|  | NaN | >1.5 m | < −0.05 m |
|---|---|---|---|
| CWT | 3.33% | 2.22% | 0% |
| PEAK | 14.77% | 0% | 0% |
| ALS-Ku$_{MAX}$ | 0.87%[a] | 0.82% | 6.97% |
| ALS-Ka$_{MAX}$ | 0.87%[a] | 0.73% | 14.88% |
| ALS-C/S$_{MAX}$ | 0.87%[a] | 1.77% | 4.87% |
| ALS-Ku$_{TFMRA50}$ | 4.11%[b] | 0.19% | 28.04% |
| ALS-Ka$_{TFMRA50}$ | 4.11%[b] | 0.12% | 33.78% |

[a]Since maximum power of waveforms is always available, this percentage denotes the times where there was not any ALS observations available within 5 m search radius of the radar to derive an along-track surface profile. [b]This percentage is a mix of the times when ALS was unavailable, or when the TFMRA50 re-tracker failed due to not being able to compute a threshold based on the restrictions set in Part I.

separating waveforms over thin ice from the ambiguous class, and providing freeboards over such ice would improve the ice conditions represented by the CryoSat-2 observations without favouring thicker ice. Currently, CryoTEMPO employs the physical re-tracker SAMOSA+, which is able to fit the waveforms and extract the tracking point based on assumptions on the distribution of surface scatters, and classify the waveforms independent of a restrictive classification scheme.

### 4.3 Comparison against airborne data

For the final comparison between the derived CRYO2ICE snow depths and the airborne snow depths derived along the under-flight, we compare the following snow depth combinations: ALS-Ku$_{TFMRA50}$, ALS-Ka$_{TFMRA50}$, ALS-Ku$_{MAX}$, ALS-Ka$_{MAX}$, ALS-C/S$_{MAX}$, CWT, and PEAK, as these combinations are based on conventional assumptions. They are compared using airborne observations identified as sea ice floes (identified using the surface classification with pulse peakiness), and with snow depths above $-0.05$ m to limit the impact of viewing geometry accounting for measurement uncertainties. Furthermore, we also limit the upper threshold of snow depths to 1.5 m assuming that higher snow depths are artefacts of the waveforms, viewing geometry, or caused by limitations to the retrieval method (see Table 1 for data size constraints).

First, we evaluate the comparison between the airborne observations and the CRYO2ICE CryoTEMPO observations by smoothing to CRYO2ICE resolution from 1 km binned airborne snow depths (Fig. 4). Here, the highest correlations are observed for PEAK and CWT (0.35 and 0.37, respectively), although all seven airborne snow depth combinations show similar low to moderate correlations (0.31–0.37). The lowest bias is observed when using ALS-C/S$_{MAX}$ of 0.03 m and with the lowest root-mean-square-deviation (RMSD) of 0.12 m, followed by CWT with a bias of negative 0.06 m and RMSD of 0.13 m.

PEAK presents a bias of 0.11 m and RMSD of 0.15 m. The other four snow depth combinations show significantly larger biases (0.16–0.27 m) and higher RMSD (0.19–0.29 m). Here, the spatial variability and resolution of the surface elevations clearly have an impact, and the fact that the airborne and satellites are observing in different ways seems to play a significant role.

We further evaluate the airborne and CRYO2ICE observations binned to 25 km segments (Fig. 5). While the bias remains the same for CRYO2ICE CryoTEMPO compared with both CWT and ALS-C/S$_{MAX}$, the correlations have greatly increased to a moderate correlation of 0.51–0.53 for ALS-C/S$_{MAX}$ and CWT, respectively. RMSD has also decreased to 0.08 m and 0.1 m, respectively. PEAK has increased by 0.01 m in bias but shares the highest correlation with CWT at 0.53 and has a decreased RMSD of 0.14 m. The other airborne estimates still have a lower moderate correlation in the order of 0.3–0.37 (with Ku$_{MAX}$ achieving the highest), and with larger RMSD (0.17–0.27 m) and high biases (0.15–0.26 m). Overall, using a combination of ALS and C/S$_{MAX}$ achieves the best result, which further supports the conclusion that the snow radar re-trackers have discrepancies that need additional investigation.

We note here that there has been no ice drift correction applied neither between CryoSat-2 and ICESat-2 (where the time difference was approximately 2 hrs and 40 minutes) nor between the airborne observations and the CRYO2ICE track (where the under-flight started approximately 1 hr and 10 minutes after ICESat-2 overpass and ended approximately 1 hr and 30 minutes after CryoSat-2 overpass, see Section 2.1). While the airborne estimates themselves do not require ice drift correction since they were acquired on the same platform, there is, of course, a potential for misalignment when comparing the airborne estimates along both CryoSat-2 and ICESat-2, if used individually, or the combined CRYO2ICE (where ideally the observations would also be drift corrected). Such correction would require high spatial and temporal resolution drift observations in the order of hours and at less than 7 km, which currently is not provided through publicly available products to the best of our knowledge. Fredensborg Hansen et al. (2024a) evaluated sea ice drift from daily medium-resolution products across one year's worth of CRYO2ICE observations and noted that, on average, the modal drift was in the order of a few kilometres between orbits Arctic-wide, hence would be minimised when smoothing with a 7 km along-track filter. Kwok et al. (2017b) presented average monthly-estimated drifts in the Weddell Sea of less than 10 km d$^{-1}$, hence hourly drifts of less than a kilometre. Thus, we assume such drifts to be minimised with the CRYO2ICE smoothing, and expect a further reduction of any drift impact when considering the 25 km segments.

## 5   Discussion and outlook

Arguably the most important aspect is considering how to collocate the radar observations acquired from ground-based as well as air- and spaceborne instruments over sea ice. Willatt et al. (2023) evaluated the ground-based KuKa radar system over Arctic spring-time sea ice and found that in the horizontal-horizontal (HH) polarisations (which are the current polarisations used on most airborne and spaceborne instruments, only the AWI IceBird carries a quad-polarised system), the strongest scattering in the majority of cases came from the a-s interface at both frequencies. Studies on their results over Antarctic sea ice from the Weddell Sea campaign in March/April 2022 are, at the time of writing, still pending, however, one might ponder whether

similar results would occur considering the more complex snowpack of Antarctic sea ice and expected scattering further up in the snowpack. Importantly, Willatt et al. (2023) also argues that while the strongest scattering came from the a-s interface at Ku-band, scattering also did occur at the s-i interface, which could be leveraged. For example, they propose a *waveform shape* technique that assumes the trailing edge of the waveform includes information about the scattering further in the snowpack, which we also applied to the airborne observations (not shown). Unfortunately, it did not yield comparable results to the snow radar nor provided much physical meaning when applied to Ka- and Ku-band for this campaign. The derived snow depths using the waveform-shape technique were approximately the same for the Ka- and Ku-band, and the centroid used to denote the s-i interface rarely coincided with strong peaks on the trailing edge or corresponding snow-ice interfaces identified in the C/S-band, which we hypothesize is due to the specifications and resolutions of the instruments and the snow conditions observed.

There are several examples along our track, where it appears that the strongest scattering occurs at the a-s interface (also in agreement with findings from e.g., Fons and Kurtz, 2019; Willatt et al., 2010, 2011) but that significant, distinguishable scattering occurs further down supposedly at the s-i interface within the same radar waveform, suggesting that both interfaces contributes to the waveform and that both interfaces could be traceable even at Ku-band. However, tracking such interfaces would require a new, novel re-tracker able to detect smaller peaks below the main scattering horizon. Applying PEAK, which is able to detect up to 5 separate peaks regardless of whether they originate on the leading or trailing edge of the waveform, with the default parameters, showed that, for some cases, it was able to track an interface below MAX. Tweaking the PEAK thresholds would allow for smaller peaks, which could be the s-i interface, to be retrieved, however, further work is required to understand how to fully apply this and which threshold values to use. Nonetheless, these findings suggest that both interfaces could – under the right circumstances and tweaked correctly – be retrieved from airborne Ku-band (and potentially Ka-band) alone, a suggestion also supported by De Rijke-Thomas et al. (2023) over Arctic spring-time sea ice. Additional work is necessary to evaluate under which conditions this applies both geographically (Arctic/Antarctic along with regionally/basin-specific) and seasonally. Similarly to the airborne data (Part I, Section 5), the snow conditions also matter to the satellite observations, and the retracked scattering horizons are directly related to the instrument specifications of the spaceborne altimeters, as with the airborne. Hence, any snow conditions (e.g., saline snow, icy layers, snow grains, etc.) limiting penetration at airborne scales are also expected to be impactful on the satellite scales, although at different magnitudes, which one may assume when the footprints are larger and the resolution lower.

Furthermore, current methods used to validate satellite observations with airborne observations apply similar re-tracking procedures at both scales for consistency without fully considering the fact that they are of different resolution and viewing geometry and may hold different information. Recent studies have aimed to evaluate which processes may play a significant role at different scales, and to what extent. For example, De Rijke-Thomas et al. (2023) presented some of the processes that may present differently from ground-based, air- and spaceborne scales dubbed as quasi-specularity. Hendricks et al. (2010) also noted that the airborne laser and radar altimeters could be statistically biased by the presence of smaller patches of open water (i.e., leads) or heavy ice deformation zones, observations also made in our study where the preferential sampling of the altimeters (radar or lidar) impacts the surface observed at the footprint scales (leads vs. broken floes). A more in-depth

dedicated analysis of the airborne data is necessary, and linking different sensors of different viewing geometries (laser vs radar) and different penetration and backscatter signatures (using multi-frequency radars) is urged as a critical focus point to

understand the full extent of where satellite multi-frequency altimetry is applicable.

The findings of this study suggest that the scattering mechanisms dominating at the Ka-band and Ku-band differ (with the Ka-band being more sensitive to different scatter interactions within the snow), resulting in several cases where the Ka-band's MAX occurred further down in the snowpack than the Ku-band. Evaluation of whether that is caused by the difference in dominating scattering mechanisms alone (e.g., volume scattering increasing biquadrate with frequency following the improved

Born approximation Mätzler, 1998) or different ice observed by the radars still needs further work. However, future work on modelling the radar waveforms and understanding the backscatter contributions from different interfaces at the various frequencies with either 1D models such as multi-layer backscattering models as presented in Tonboe et al. (2021) or the Snow Microwave Radiative Transfer (SMRT) model (e.g., Meloche et al., 2024, in review) is highly encouraged, provided viewing geometry, roughness, and resolutions are thoroughly considered and to the best of abilities accounted for. This is critical to

investigate further for the upcoming CRISTAL mission, but also for understanding the impacts of slant-looking Ka-band radar altimetry when using NASA's recently launched interferometric Surface Water and Ocean Topography (SWOT) mission, and for consolidating radar altimetry time-series when considering different operation modes (from low-resolution-mode (LRM) and pseudo-LRM to FF-SAR, SAR and interferometric SAR) which also impact the retrieved waveform and the derived range.

Considering different viewing geometries, one may also consider whether the current CRYO2ICE methodology is appli-

cable to Antarctic sea ice. One limitation for in-depth comparison is the limited number of CryoSat-2 radar freeboard observations, where both ESA-E and FF-SAR only provided observations for diffuse, non-ambiguous surfaces. However, were observations available (e.g., CryoTEMPO), comparison between airborne-derived snow depth estimates and CRYO2ICE CryoTEMPO showed similar statistics as for an along-track comparison with a monthly snow depth composite (Garnier et al., 2021). Discrepancies between snow radar and CRYO2ICE snow depths (using CWT and PEAK) present a current challenge

associated with evaluating airborne and spaceborne-derived geophysical variables. Using lidar and $C/S_{MAX}$ showed the highest agreement amongst all snow depth estimates derived at the 25 km scale. Additional work is encouraged for the evaluation of the CRYO2ICE orbits across the full Antarctic season to evaluate the information retrievable from near-coincident spaceborne laser and radar altimetry. This is especially relevant in preparation for CRISTAL, where understanding how to align observations – acquired even on the same platform (even if not at the frequencies expected for CRISTAL) – at different footprints

and with potentially different scattering mechanisms should be approached. Here, the future airborne simulator for CRISTAL, known as CRISTALair (Garcia-Mondejar et al., 2023), will provide further insights and allow for continued dual- or even multi-frequency airborne observations to support the research on consolidating different spatial scales.

For our study, we have not evaluated the relationship between roughness and the different re-tracked interfaces, however it cannot be neglected and is therefore discussed here. Various roughness estimates and their relationships would need to

be evaluated to understand how they impact the airborne radar observations, and how this is potentially translated into the spaceborne estimates. As such, one could compute a roughness estimate from the standard deviation of the lidar elevations within 5 m from the radar to evaluate the impact on the airborne observations further, similarly to De Rijke-Thomas et al.

(2023). To provide a measure of roughness applicable towards the satellite scales, one must use the full swath. Here, roughness estimates could be computed per scan line following the methodology of Beckers et al. (2015) and Hutter et al. (2023), which could supply some information on the large-scale variability in roughness. Yet, such a roughness estimate only partially covers the CryoSat-2 footprint (400 m versus the across-track Doppler-beam footprint of 1,600 m) and provided at approximately every metre along-track (compared to the 300 m along-track footprint of CryoSat-2). Thus, significant work should be invested into understanding roughness at the different scales of the instruments, and how this may play a role. This is out of the scope of the current study but considered critical future work for the sea ice altimetry community. In addition, one must consider whether the roughness derived from the lidar (essentially, a measure of snow roughness) might be different from the sea ice roughness. Here, effects such as surfaces occurring smoother whenever snow redistributes around the rougher ice surfaces or rougher when snow features like sastrugi are created by wind redistribution likely have an effect. How and to what extent roughness from different interfaces impacts the radar pulse are also questions for future work.

It is critical that we continue and enhance the collection of consistent, detailed observations from a variety of spatial scales (ground-based to different areal components); however, without a strategically planned field-based campaign emulating what is observed from air to space, the evaluation of such observations is limited. However, without in situ components, validation of the airborne systems from different locations (regionally as well as dependent on the hemisphere, and considering the seasonal impact) cannot be independently ensured. Hence, the evaluation of the satellite observations, when compared to airborne observations, can only discuss similarities, but not whether they are independently able to obtain the variables we expect given their observational capabilities.

## 6 Conclusions

In this study, we compared airborne snow depth estimates along a CRYO2ICE (CryoSat-2 and ICESat-2) under-flight carried out over the Weddell Sea on 13 December 2022 during the CRYO2ICEANT22 campaign. The derived airborne snow depth estimates (Part I Fredensborg Hansen et al., 2024b) were compared with snow depth estimates derived along the CRYO2ICE orbit. The CRYO2ICE snow depths were also compared with two other Antarctic snow depth estimates (CASSIS and AMSR2) for further evaluation.

Comparison between three different CryoSat-2 processing chains showed large discrepancies likely caused by the waveform classification scheme utilised, and the ability to re-track waveforms from mixed-to-specular surfaces. Due to ESA-E and FF-SAR primarily re-tracking diffuse waveforms, their average radar freeboards along the CRYO2ICE track, whenever observations were available, was around ∼0.30 m. All three products had about 30% of negative freeboards. Comparing the derived CRYO2ICE snow depths showed, on average, CASSIS having 0.12 m thicker snow, and AMSR2 and CRYO2ICE agreeing within 0.01 m (using CryoTEMPO as reference). Evaluating additional CRYO2ICE orbit across the Antarctic sea ice cover would be valuable to understand how well the snow depth variability can be retrieved along orbits using the established CRYO2ICE methodology.

We argue that future work should evaluate how we bridge different spatial scales (ground, air, and space) with the modelling of radar backscatter taking into consideration viewing geometry, frequency, and footprints. Through comparison with the airborne-derived snow depth estimates (using ALS, Ka-, Ku-, and C/S-band "snow radar" observations), and binned to 25 km average, correlations of 0.51–0.53 were achieved with bias down to 0.03 m and RMSD of 0.08 m. The snow radar retrieval algorithms showed discrepancies, but still a better agreement than with the snow depth estimates using Ka- and Ku-band (in any combination with ALS, and different re-trackers), suggesting that the different processing methods from air and space, considering the different footprints, need further work to be made comparable. We strongly urge the community to further evaluate collocated past ground-based, air- and/or spaceborne observations from which new insights can hopefully be identified. Finally, we note, that without an in situ component, we cannot state with certainty whether our satellite missions are able to retrieve what we believe they are retrieving. The same goes for the airborne data. Here, it is critical that we bridge the scales with both ground, aerial, and satellite observations – ideally simultaneously – while accounting for the differences in spatial scales by sampling in patterns emulating what the satellite/airborne systems will observe. While it is dependent on logistics, weather and sea ice conditions whether the ground-based in situ campaigns are successful, they are nonetheless a necessity to fully understand the capabilities of our remotely sensed observations, and must stay on the agenda when planning validation efforts.

*Code and data availability.* CRYO2ICEANT22 campaign data are currently in the final stages before approval from ESA and will, once approved, be published on the ESA campaign site and on www.cs2eo.org. The processed CRYO2ICEANT22 data presented here are available from DTU DATA at under DOI: 10.11583/DTU.26732227. The code for processing the provided CRYO2ICEANT22 data and the satellite/model data are available on the following GitHub repository (Fredensborg Hansen, 2024): https://github.com/reneefredensborg/cryo2iceant22-airborne-cryo2ice-weddell-sea-ice (last accessed: 11 September 2024).

CryoSat-2 ESA-E L1B and L2, and ESA CryoTEMPO sea ice thematic data product are available from their FTP server: ftp://science-pds.cryosat.esa.int/ (last accessed: 08 August 2024). The CryoSat-2 FF-SAR data product was provided by DY. ICESat-2 data ATL07 (Kwok et al., 2023a) and ATL10 (Kwok et al., 2023b) are available at National Snow and Ice Data Center (NSIDC) under DOI: 10.5067/ATLAS/ATL07.006 and DOI: 10.5067/ATLAS/ATL10.006, respectively (last accessed: 08 August 2024).

The daily output on 13 December 2022 of CASSIS in was provided by IRL via Andy Ridout (University College London, UCL). Currently, CASSIS model output from 1981–2021 (Lawrence et al., 2024) is available from http://www.cpom.ucl.ac.uk/cassis/ (last accessed: 09 July 2024, no DOI provided). AMSR-E/AMSR2 Unified L3 Daily 12.5 km Brightness Temperatures, Sea Ice Concentration, Motion & Snow Depth Polar Grids, Version 1 (Meier et al., 2018) is available under DOI: 10.5067/RA1MIJOYPK3P (last accessed 08 August 2024) on NSIDC.

*Author contributions.* Conceptualisation: RMFH, HSK. Data curation: RMFH, JL, FRM, HSK, IRL. Formal Analysis: RMFH, HSK. Funding acquisition: HSK, KVH, RF, JW. Investigation: RMFH, HSK, AJ, IRL, SBS, GV. Methodology: RMFH, HSK, AJ, IRL, AS, KVH,

ER. Project administration: HSK, RF, TGDC, JW. Software: RMFH. Supervision: HSK, ER, KVH, RF, JW, TGDC. Validation: RMFH. Visualisation: RMFH. Writing – original draft: RMFH. Writing – review & editing: All authors.

*Competing interests.* The authors declare no competing interests.

*Acknowledgements.* This work was supported by ESA with DTU Space CRYO2ICEANT 2022 – Technical Support for the Antarctica
460   CRYO2ICE 2022 Experiment (#4000141420/23/NL/IB/ab) and the UK Natural Environment Research DEFIANT project (#NE/W004747/1) RMFH and HSK acknowledge the great discussions of the International Space Science Insitute (ISSI) Team 501 - "Multi-Sensor Observations of Antarctic Sea Ice and its Snow Cover". RMFH was supported by the Nordic5Tech joint PhD-alliance research project between DTU and NTNU to characterise extreme sea ice features with a combination of remote sensing, in-situ data, and physical modelling. AJ was supported by the Research Council of Finland grant #341550.
465   We acknowledge the use of data and/or data products from CReSIS generated with support from the University of Kansas, NASA Operation IceBridge grant NNX16AH54G, NSF grants ACI-1443054, OPP-1739003, and IIS-1838230, Lilly Endowment Incorporated, and Indiana METACyt Initiative.

We thank the crew of the Rothera research station and the pilots who made the CRYO2ICEANT2022 campaign possible. Finally, thanks to Nathan Kurtz for providing access to the quick-look ATL07 and ATL10 files for initial processing until release 006 was publicly available.

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

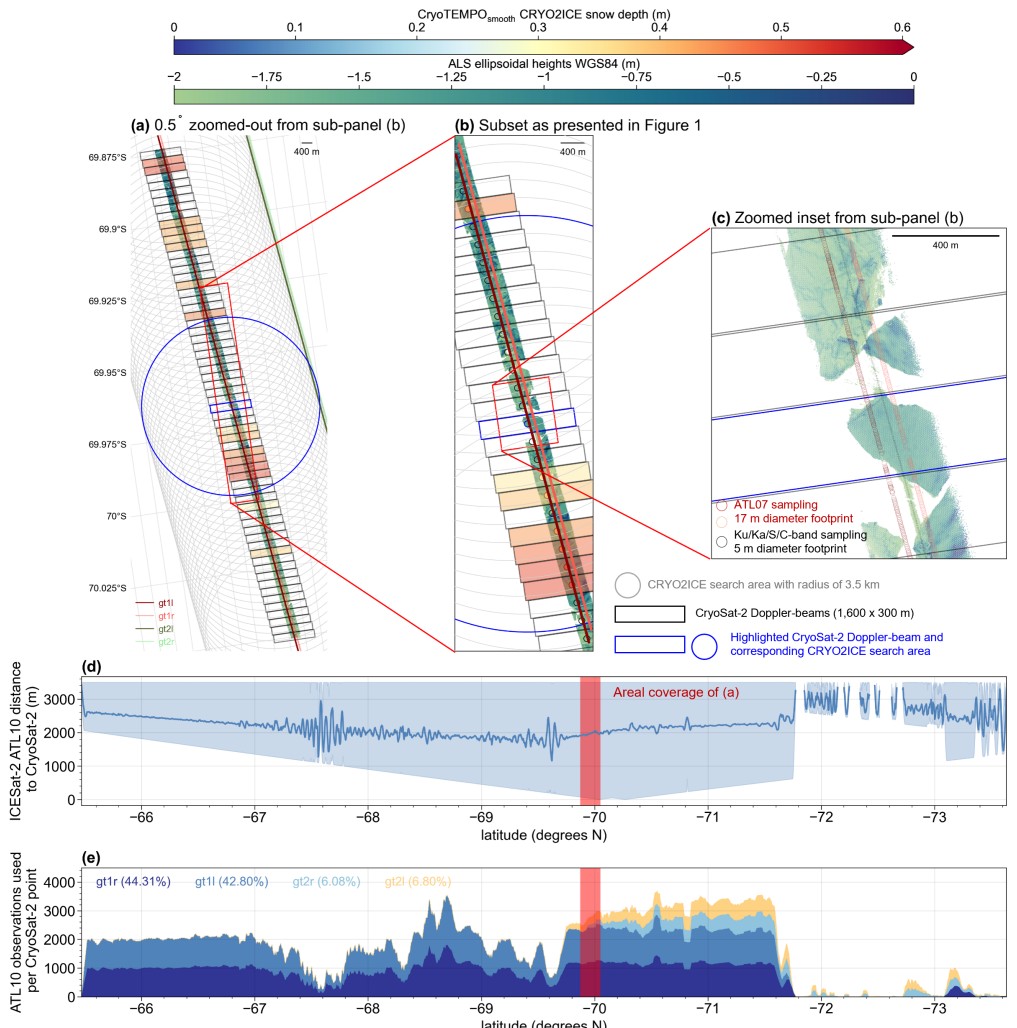

**Figure 1.** Comparison of data coverage between CryoSat-2, ICESat-2 (coinciding beams), and airborne data, and statistics of CRYO2ICE collocation. The CryoSat-2 Doppler beams are coloured according to the CRYO2ICE-derived snow depth when available. (a) Transect along the CRYO2ICEANT22 and CRYO2ICE under-flight (0.5° zoom-out from the subset shown in sub-panel b) highlighting the full coverage of the 3.5 km search radius applied to CryoSat-2 and ICESat-2 observations for collocating CRYO2ICE observations, including ICESat-2 leftmost and central beam-pairs. (b) A subset of the CRYO2ICEANT22 transect (for location, see Part I), where, in addition to the CryoSat-2 Doppler-beam footprints, the central location of each Doppler-beam is shown by a small, not to scale, point-coloured according to the derived CRYO2ICE snow depth. (c) Further zoom into the subset highlights the footprint of ICESat-2, CryoSat-2 Doppler-beams, and the airborne observations. (d) Average distance to ICESat-2 ATL10 freeboard observations from each CryoSat-2 observation, where the shaded area denoted maximum and minimum distance. (e) The Number of ATL10 observations used per CryoSat-2 point to derive the CRYO2ICE ICESat-2 collocated total freeboard observations, separated per beam. The red vertical span on (d–e) presents the coverage presented in sub-panel a.

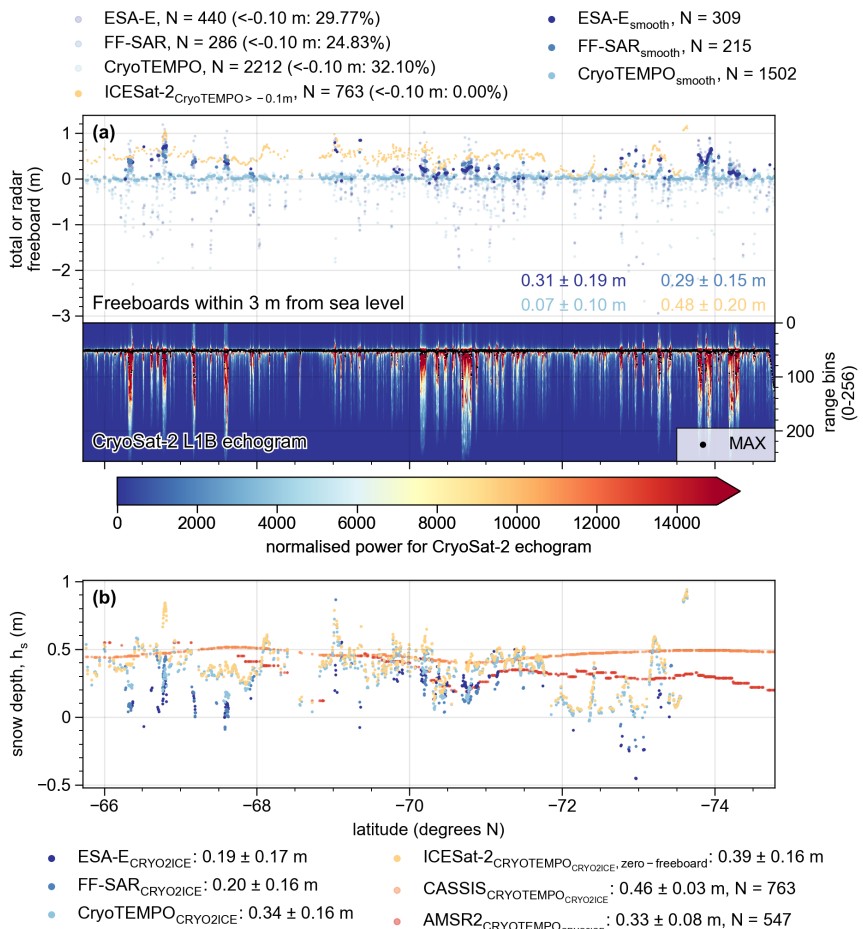

**Figure 2.** Estimation of snow depth along CRYO2ICE orbit using CryoSat-2 radar freeboard observations from Baseline-E (ESA-E), fully-focused SAR (FF-SAR), or the CryoTEMPO processing chains, and ICESat-2 ATL10 observations. (a) CryoSat-2 radar freeboards within 3 m from water level at native resolution, smoothed CryoSat-2 radar freeboards (using radar freeboards above −0.1 m), and total freeboard from ICESat-2 derived from smoothed CryoTEMPO observations following the methodology of Fredensborg Hansen et al. (2024a). In addition, the CryoSat-2 L1B echogram (radar waveforms) is provided to identify outliers in radar freeboards, where the colorbar shows the normalised power of the waveforms as available in the CryoSat-2 ESA-E product (normalised to 0–65,535 with the colorbar adjusted for visualisation purposes). (b) Derived snow depth for each CryoSat-2 processing chain (smoothed) using derived ICESat-2 CRYO2ICE total freeboards, and compared with nearest-neighbouring CASSIS and AMSR2 snow depths. Average and standard deviation are given for radar/total freeboards (for all available data points) and snow depth (whenever CRYO2ICE observations are available, and for CASSIS and AMSR2 snow depths, the statistics are provided along CryoTEMPO$_{CRYO2ICE}$). Note, the latitudinal extent of the CRYO2ICE only partly overlaps with the under-flight as shown here. In particular, AMSR2 observations are mostly available south of the airborne under-flight, see Part I.

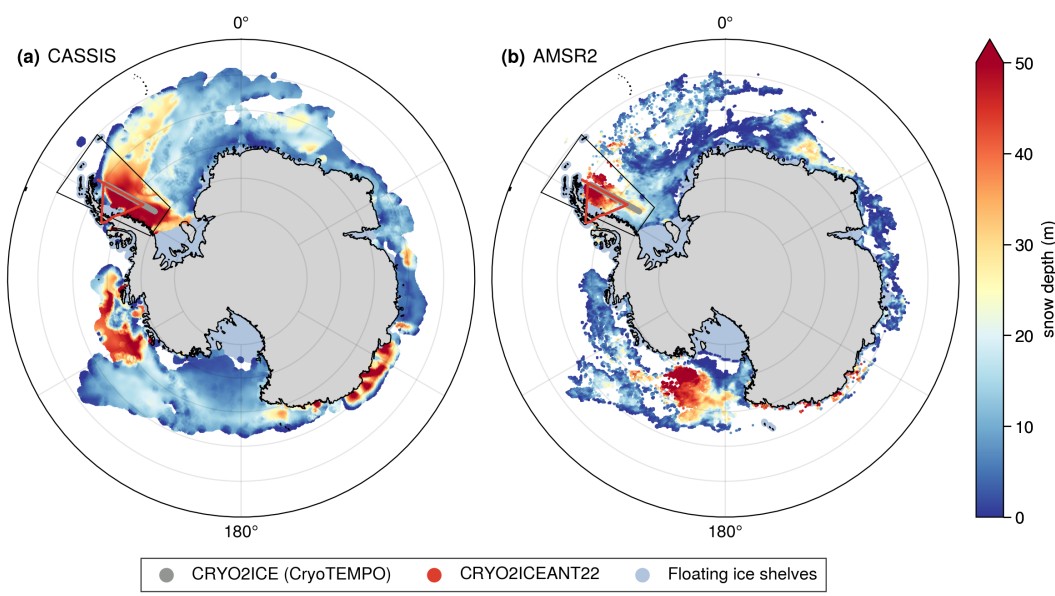

**Figure 3.** Pan-Antarctic snow depth distribution of 13 December 2022 from (a) CASSIS and (b) AMSR2. The location of the airborne CRYO2ICEANT22 track and collocated CRYO2ICE track is shown, and the black outlined box refers to the area showcased in Fig. 1 of Part I. Floating ice shelves are provided in the NSIDC-0780 Antarctic regional mask data product (Meier and Stewart, 2023) at 6.25 km (using the NASA classification).

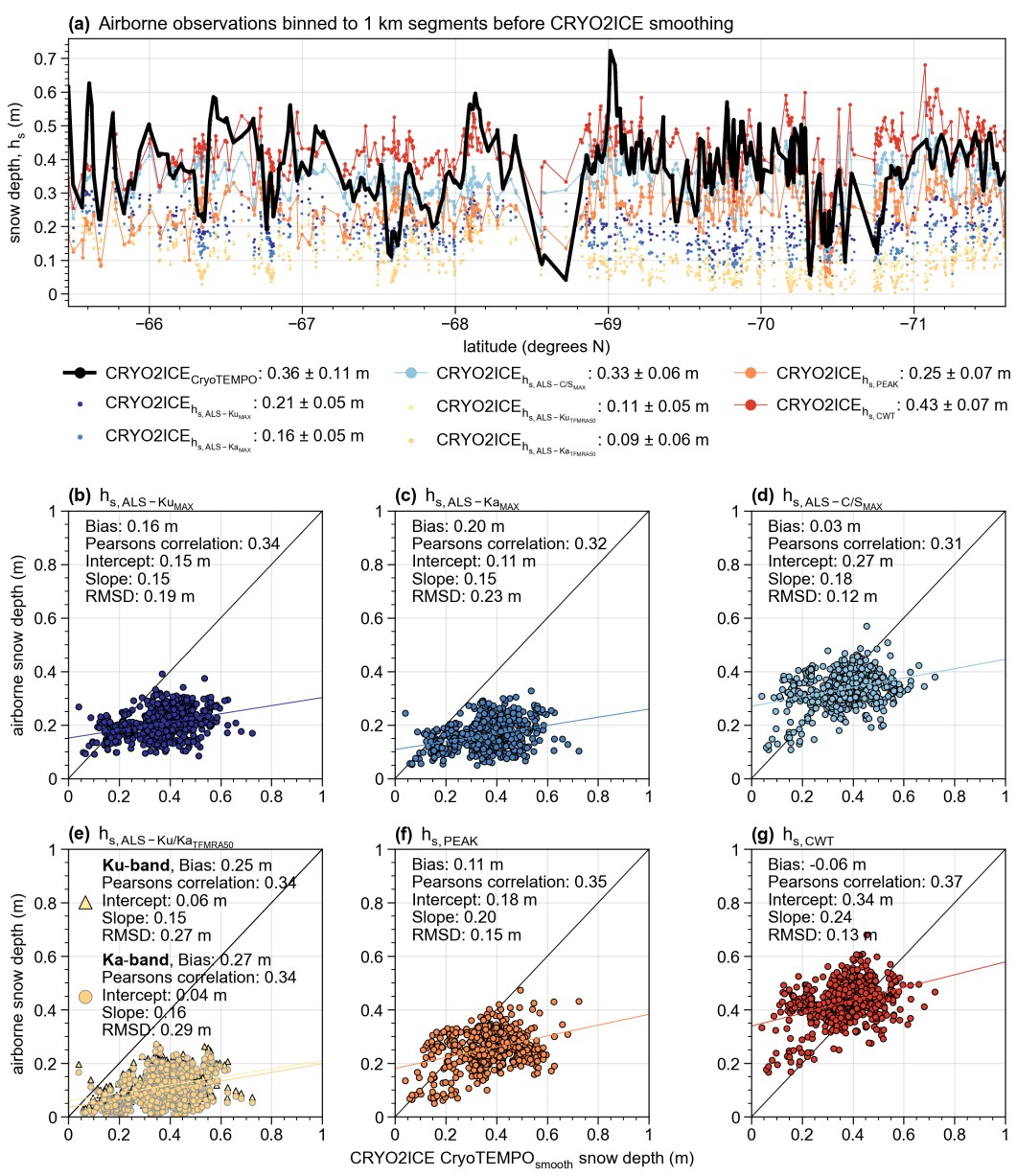

**Figure 4.** Comparison between airborne snow depth estimates and CRYO2ICE CryoTEMPO snow depths. Statistics computed for $N = 576$ (coincidence of CRYO2ICE CryoTEMPO and airborne data). Bias computed as $CRYO2ICE_{CryoTEMPO} -$ airborne snow depth. (a) Along-track comparison where airborne snow depths with most similar statistics are shown with both a marker and line. (b-g) Comparison between CRYO2ICE CryoTEMPO and each airborne product.

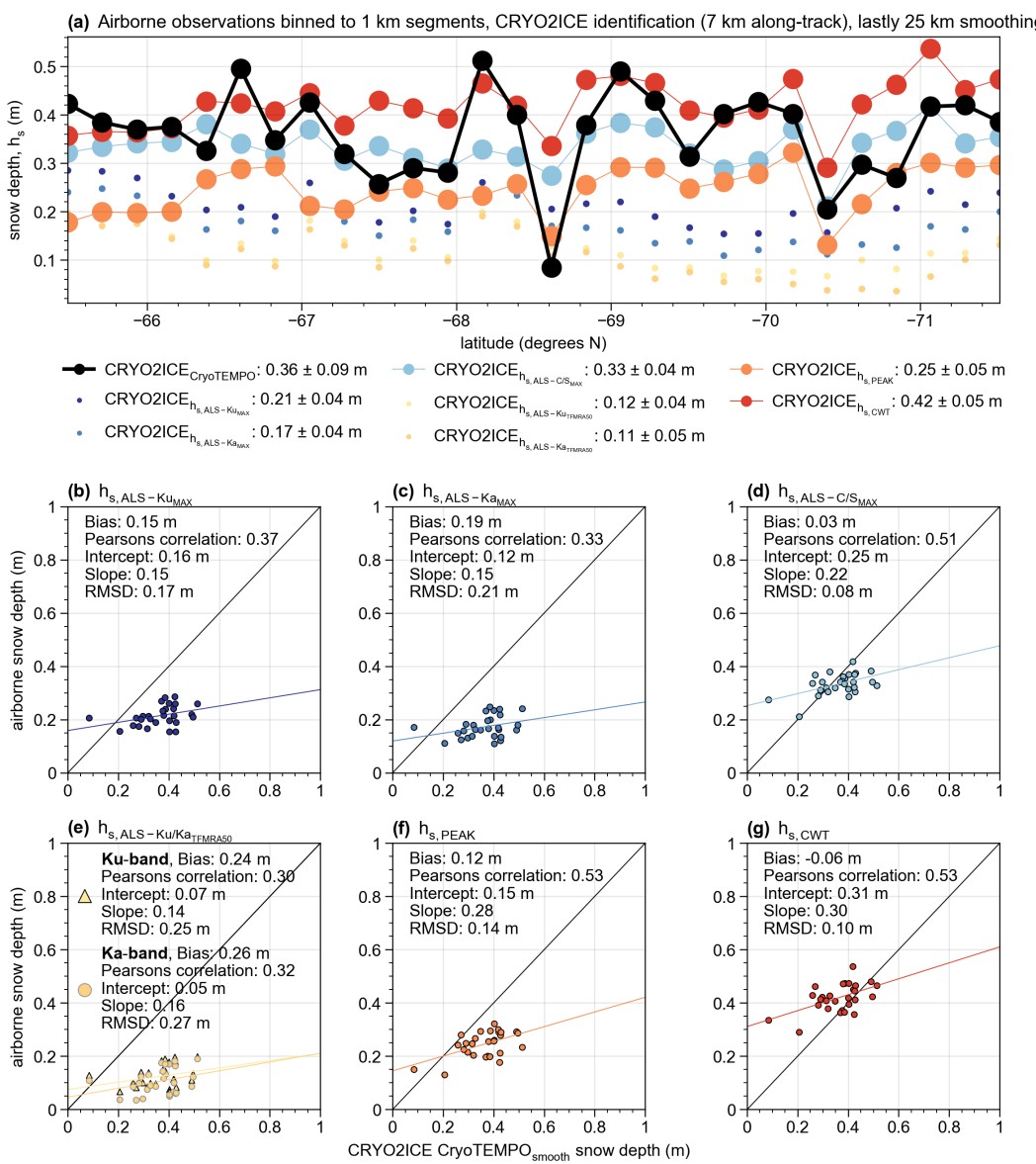

**Figure 5.** Same as Fig. 4 with 25 km binning applied after CRYO2ICE identification. Statistics computed for *N* = 28.