# Peer review of "Multi-frequency altimetry snow depth estimates over heterogeneous snow-covered Antarctic summer sea ice – Part I: C/S-, Ku-, and Ka-band airborne observations"

_EGUsphere, 2024_

## Author Comment (AC4)

*Final response - author comments*

*The final response is provided in bold, italic, and blue font. We thank the reviewers for the constructive feedback and positive comments, and present the final response based on the discussion.*

- *The manuscript has been split into two submissions (noted Part I and Part II) as suggested by both reviewers, and based on suggestion by the editor.*
    - *Based on this, the title has changed on both submissions, inspired by the reviewer comments, to:*
        - *"Multi-frequency altimetry snow depth estimates over heterogeneous snow-covered Antarctic summer sea ice -- Part I: C/S-, Ku-, and Ka-band airborne observations"*
        - *"Multi-frequency altimetry snow depth estimates over heterogeneous snow-covered Antarctic summer sea ice -- Part II: Comparing airborne estimates with near-coincident CryoSat-2 and ICESat-2 (CRYO2ICE)"*
    - *Please see specific comments to reviewer comments (as final response) on why the title has been adapted from their suggestions.*
- *Both manuscripts now include a "call for in situ planning efforts" as suggested by reviewers in their discussion and conclusion sections.*
- *Part I includes an extended discussion on impact of brine-wetted snow and antenna beamwidth.*
- *Figure 3a and 7a (Part I) have been updated to change the order of frequency-bands. All tables in Part I have also been edited to change the order of frequency-bands.*
- *Figure 5 has not been updated to include more models, as we had not received input from reviewers regarding this (although there was limited time). If the reviewer has other models of refractive index considering brine-wetted snow, please do provide and we will be happy to incorporate.*

*If there are no additional "final response" comments to the specific reviewer comments, please consider the first response as still valid.*

**Author comments**

**We thank the reviewers for the constructive feedback and positive comments. We have provided our response below, in bold text prefixed by "Response: (...)". Please also find a short summary of proposed edits (of major edits), along with minor expected edits to be made for the final version - pending editor/reviewer approval.**

- **While we believe that the manuscript benefits from being one manuscript due to the choices made for different analyses and processing are interleaved, we agree with the reviewers that it is a longer paper. If the editor and reviewers all agree that separation into smaller manuscripts is required, we have given the following suggestion:**

- - Instead of submitting completely separate papers, since the studies are linked, we have suggested separating the manuscript into "Part I and Part II" pending editor/reviewer approval. If editor/reviewers agree, we suggest reorganizing the current manuscript as Part I and will submit a new manuscript, linked to this one, as Part II. Here, the objectives will be split, so Part I will focus on the extensive airborne processing, and Part II on the collocation and comparison with satellite observations.
  - However, any other major or minor comments directly related to text or figures in the paper are addressed before any separation into such manuscripts. This might also help decide whether such separation is necessary.
- Both reviewers have noted that a lack of in situ data limits the evaluation of how well these combinations compare with actual conditions . Indeed, we have tried several combinations to mimic what could represent "truth" and compare these with satellite data, where it is still clear that there is work to be done. To ensure that this is clear, we have (based on suggestions) edited the title of the paper, see proposed titles in the response to Reviewer #1 and Reviewer #2 (depending on whether the manuscript is split into two parts, or kept as one). In addition, based on reviewer recommendation, we will also ensure that the manuscript calls for future in situ data validation in combination with the airborne data, since it is also clear from this study, that the snow radar (which is usually used as ground-truth) still shows discrepancies and that additional insights would help understand the limitations of the frequencies currently in play.
- Sub-section 5.1 (in case the manuscript is kept as one manuscript) will be expanded to include a more in-depth description of the impact of flooded or brine-wetted snow in relation to negative freeboards and the snow loading over thinner ice.
- Sub-section 5.2 (in case the manuscript is kept as one manuscript) will be expanded to include a description of the impact of differences in scattering ratios (specular/diffuse) when working with air- and spaceborne data.
- Figure 3a; here we will change the order of subplots to follow the range of frequencies. Same goes for Figure 7a.
- Figure 5 might be updated, pending the exploration of relevant models for the refraction index in snow, to show the impact of e.g., brine-wetting. We have asked the reviewer to specify the particular model(s) they have in mind.

*Reviewer #1*

Review of "Exploring microwave penetration into snow on Antarctic summer sea ice along CryoSat-2 and ICESat-2 (CRYO2ICE) orbit from multi-frequency air- and spaceborne altimetry" by Fredensborg Hansen et al.

The topic is very welcome and the MS deals with a very large and difficult dataset. No doubt a lot of work went into this.

The MS is well written and it contains a lot of references to relevant publications, to the extent that it reads more like a literature review than a research MS. The figures in the MS are illustrative, with a few minor comments (see specific comments). However, we get to page 21 before we start the results section and the MS is 42 pages without the references and appendices. The intro and the data sections need to be much more focused.

There are two objectives with the study stated at the end of the introduction: "With this study, we attempt to bridge several gaps: (a) evaluate microwave penetration into Antarctic snow cover at Ka- and Ku-band and the applicability of snow-radar, (b) compare airborne snow depth estimates derived based on traditional hypothesis of penetration with near-coincident spaceborne radar (CryoSat-2) and laser (ICESat-2)." These are really interesting questions to answer but I would suggest to split this MS into two separate submissions and focusing each of the 2 MS's on each of the objectives (a) and (b).

**Final response: The manuscript has now been split into two following each of the objectives as proposed by the reviewer.**

**Response: We thank the reviewer for the positive comments and the feedback! Indeed, it is a long paper, and it was discussed in detail between the co-authors before submission whether it should be represented as two separate manuscripts or as one. We originally kept it as a single study, since much of the processing of the airborne observations was shaped by the original research question of comparing them with satellite data. Additionally, the processing choices of both airborne and satellite observations are interlinked. We are keen to keep it as one study, pending editor approval.**

**Should the editor decide that two separate papers would be more beneficial, we suggest separating the papers as "Part I" and "Part II" to be submitted to The Cryosphere. We believe it would be more beneficial to keep the papers somehow linked since the studies themselves are linked. We await editor - and reviewers - feedback regarding this. In this case, the current manuscript will be altered to fit Part I and primarily focus on the extensive airborne data.**

**Suggested, altered, titles to each part (including title suggestion from Reviewer #2, although slightly adapted for this purpose) could be:**

- **An intercomparison of multi-frequency altimetry over heterogeneous snow-covered Antarctic summer sea ice - Part I: Snow depth retrievals from airborne sensors for satellite altimetry validation**
- **An intercomparison of multi-frequency altimetry over heterogeneous snow-covered Antarctic summer sea ice - Part II: Comparison between airborne and near-coincident CryoSat-2 and ICESat-2 (CRYO2ICE) snow depths**

In these separate, but linked, manuscripts, the introduction and data sections as well as the results, discussion and outlook sections, can be much more focused. Overall, we believe this approach is optimal for limiting the length in case a single manuscript is no longer an option, allowing each manuscript to be more focused while maintaining a cohesive thread through the processing, analysis, and objectives.

This is also based on Reviewer #2's "Less Major Concern", which also concerned the length of the manuscript. Here, they noted, that a dramatic edit would be to split the paper into one more focused on the processing of the data including most of the results, and another to be more of a literature review. While we believe "a literature review type paper on multi-frequency and multi-scale radar altimetry of snow-covered summer sea ice which would include all of Section 5 (Discussion and outlook) and other pertinent material" would be great and beneficial, especially in preparation for the CRISTAL mission. In addition, this also relates to the concern the Reviewer states regarding how the manuscript might be read as a literature review.

It was important for us to highlight the many different studies already conducted in this field - particularly how most of them have not ventured into the Ku- and Ka-band observations from airborne scales, but primarily focusing on snow radar observations in the Arctic. Additionally, due to the lack of in situ data necessary for understanding the physical properties that might impact our altimeter observations, we referenced the studies that have previously shown these challenges. However, we acknowledge that the manuscript could be more focused to reduce the length. Therefore, we hope that the suggestion of separating into two parts (since the studies are linked) meets the reviewers' approval, in the case that a single (albeit more focused) manuscript is not an option.

There are no in situ measurements at the surface which makes any conclusion based on these datasets pretty vague and speculative. The discussion of the results has references to other peoples studies with possible explanations to the observed variations. For example, in section 5.3 which is about 1 page long, the only observation from own data is: L877: "…based on visual comparison along a subset (Fig. 7a), that the snow-radar is incapable of retrieving snow depth over ridges due to only one single peak being present." That there are no snow and ice in situ data in CRYO2ICEANT22 (and so many CryoSat validation campaigns before that) is a waisted opportunity and something to think about for ESA (part of this study) for future campaigns.

*Final response: More direct in-text examples (reference to Figures) have been made for the discussion (Part I). Furthermore, the importance of in situ observations to support the validation of both airborne and satellite observations have been included in both the discussion and conclusion of both parts.*

Response: Indeed, having in situ data would have been ideal for a more precise evaluation of why the methods are incapable of retrieving the same magnitudes of snow depth, as they otherwise should (under the different assumptions). We will include more direct examples in-text (with reference to Figures) of where what has been observed in the previous studies may apply, and will (also based on the recommendation of Reviewer #2) include the importance of in situ observations for such campaigns. This will either be included as a separate sub-section (in case of the two-piece linked papers) or be integrated into the discussions sub-sections whenever relevant.

It is worth noting, however, that many airborne campaigns (including CryoVEx, Operation IceBridge and AWI IceBird campaigns) have been completed without in situ campaigns simply because of the challenges with respect to logistics, funding and making comparable collection strategies. Also, several in situ components have been planned and conducted for airborne campaigns. Some of these were successful in terms of collection of data, but it is always a challenge to collocate in time and space, whereas others were unable to be conducted in the field, usually due to restricting weather or sea ice conditions. There are almost always some challenges when comparing the in situ and airborne data and often we are challenged by the temperature being questioned, as in particular the airborne campaigns are restricted to take place in a very limited time-window due to the polar darkness and melt-season, but also resources to actually analyze and collocate the data is also limited. There are several campaigns where the data simply have not been fully utilized, which should have higher priority in the future. We still need to develop a strategy for the "perfect" or at least, ideal, set-up. Efforts have been initiated within e.g., the St3TART/St3TART-FO and CRISTAL IN-PROVA projects regarding evaluating such set-ups.

Furthermore, it is essential to recognize the value of airborne observations as critical data sources. They provide representations closely aligned with satellite observations, albeit with varying spatial and range resolutions. These observations offer insights that may not always be captured in field measurements due to limitations in field coverage and differences in resolution and spatial extent, together with the challenges in aligning such observations over drifting ice in the polar regions. Landings on the drifting sea ice are becoming more and more challenging with higher risks due to the thinning and more dynamic sea ice cover. Once this is said we emphasize that in situ measurements are crucial in order to better understand and validate airborne estimates. We further encourage collaboration and coordination of airborne and in situ components.

Specific comments:

L120: Ku and Ka band is mentioned in the objectives, what about C/S-band?

**Response: It was already mentioned in the sentence "and the applicability of the snow-radar", but will be further highlighted. Ka/Ku-band were primarily mentioned, since these sensors are directly relevant for CRISTAL (and what CRYO2ICE is partly emulating). In our study, C/S-band snow-radar is used as our "ground-truth" following assumptions made on Arctic sea ice. However, we also show that the ground-truth from the snow-radar for Antarctic summer sea ice is still very much work in progress.**

**The sentence now reads: "With this study, we attempt to bridge several gaps: (a) evaluate microwave penetration into Antarctic snow cover at Ka- and Ku-band and the applicability of C/S-band snow-radar, with comparison to lidar airborne observations (...)".**

*Final response: In Part I, the sentence now reads as: "In Part I, we evaluate microwave penetration into Antarctic snow cover at Ka- and Ku-band and the applicability of a C/S-band snow-radar, with comparison to lidar airborne observations"*

L126 and onwards: I think that it would be useful in the data section (2) to structure some of the information on instrument frequencies and bandwidth, antenna size, flight altitude etc. in tables for example by moving TA1 up into the text. Actually, I think that TA1 contains the relevant technical data for the discussion.

**Response: We thank the reviewer for the comment, and have included the TA1 in the data section. In case of a two-manuscript linked study (Part I and II), the table will be included in Part I.**

Figure 3: Please order the frequencies from high to low or low to high, here it is Ku-, Ka-, C/S-band.

**Response: We thank the reviewer for the comments, and note that this will be changed in the final version.**

*Final response: Has been updated.*

L420: While the spatial scales vary from airborne to spaceborne radar observations, so does the waveform and the ratio between specular and diffuse scattering. The antenna beam width (in TA1) is a function of frequency and this affects the waveform shape and the detecting individual scattering contribution from the snow surface and the snow ice interface. Please, include a discussion of that and how it displays in the waveform data. This is also relevant when discussing the satellite and airborne data overlap in section 3.4 and the re-trackers in section 4.1.

**Response: Indeed, it does. We thank the reviewer for mentioning this important aspect! This is also highlighted in the study of de-Rijke Thomas et al. (2023), when discussing the impact of quasi-specular scattering, and what we aimed at discussing in detail in Section 5.2 "Viewing geometry and sampling differences between sensors at different spatial scales". We will expand on these aspects in the relevant sections. We encourage the reviewer to note specific studies/work to be referenced for this discussion.**

*Final response: A discussion on this has been included in Part I, discussion with the following text:*

*"In particular, an aspect of interest here is the ratio of scattering mechanism dependent of frequency - and in turn, on the antenna design. Antenna beam-width is a function of frequency (Table 1) which impacts the shape of the retrieved waveforms, and the possibility of detecting individual scattering contributions from different interfaces. The beam-limited footprint is directly proportional to the antenna beamwidth, meaning that the C/S-band radar in combination with its antenna beamwidth has a significantly larger beam-limited footprint (~160--550 m) compared to both Ku- and Ka-band (ranging at ~116 m down to ~80 m), and that potential strong scatters from off-nadir may impact more significantly at C/S-band than at Ka-band. More importantly, the antenna beamwidth along with the individual surface scatters drives the shape of the retrieved waveform with off-nadir scattering reflected in the trailing edge of the waveform which could be picked up as s-i interface using the PEAK re-tracker in case of strong off-nadir reflections. Or for snow-radars, lower spikes representative of the a-s interface observed before the main spike are in fact system artifacts known as sidelobes (Kwok and Haas, 2015), which could impact the snow depth retrieval. However, pre-processing steps have been taken to minimise the impact of sidelobes (see Appendix A). Nonetheless, it is worth considering what the impact of antenna beamwidth in combination with frequency has on which interface scattering dominates, and how this is reflected in the waveform shape. However to our knowledge, this has not been evaluated in depth for the radars utilised, and is out of scope of the current study."*

*We hope this meets the reviewers approval, but are open for further discussion and feedback from the reviewer.*

L429: "… assuming a cold and dry snowpack." Even though the snow pack may be cold or at least "not melting", then it is most likely not dry. Snow on first-year ice contains brine and that has a significant impact on the refractive index and the light propagation speed (and microwave absorption). This is not included in the equation from Ulaby and, in addition, the Ulaby equation can be quite different from other models for computing the refractive index of snow. Frequency also plays a role small role when the snow is brine wetted. This is most relevant at C/S-band which is penetrating to the snow-ice interface. A discussion of that would be relevant and please include the results from different models in figure 5.

**Response: We thank the reviewer for the comment. Indeed, a wetted - or brine-wetted - snow pack impacts the microwave penetration capabilities, an aspect not neglected in the discussion in Section 5.1. Nonetheless, we understand the importance of further showing the impact that brine-wetted snow will have on multi-frequency observations. We will expand on this discussion, and if possible, include the different relevant models in Figure 5. However, this mostly makes sense in case of a split manuscript. Otherwise, the manuscript becomes even longer, and since there currently are no studies that use a different assumption than cold/dry snow packs for the snow depth retrieval of air/space-borne altimetry, this also ventures into a different study, outside the scope of this study, which is worthy of its own analysis and publication.**

**If possible, we do urge the reviewer to detail which models could be of interest to include in Figure 5. However, we also think it is important to note that this methodology - and specifically this equation of refractive index - is used in the majority of dual-frequency studies, and thus, for the sake of comparing across studies, use this as the primary refractive index equation. However, we will expand on the discussion related to brine-wetted and saline snowpacks even if the model with a refractive index reflecting a saline snowpack is not presented in Figure 5.**

*Final response: We have included a longer discussion on the impact of brine-wetted snow in Part I in the discussion.*

*We have not updated Figure 5 to include a model that takes into account brine-wetted snow (as we are not currently aware of an example that presents the refractive index considering this aspect), but await the reviewers comments on this.*

Figure 7. What are ellipsoidal observations? The term is used in several places in the MS (e.g. fig. 4). Please define.

**Response: Thank you for this keen observations, we will define this at first mentioning, where it is defined as:**

**"Ellipsoidal heights are elevations relative to a reference ellipsoid in this case WGS-84. This is computed as the range obtained from re-trackers (Section 3.1) subtracted the altitude of the air- or space-craft."**

L927: "Based on these, it appears that Ka- and Ku-band both scattered from mid-way in the snow and upwards in the majority of times." I think there is a confusion here between the track-point, which is derived with the re-tracker and the waveform and the actual scattering surfaces. These two things are not the same and here they are combined into a confusing statement that "…Ka- and Ku-band both scattered from mid-way in the snow…". Please reformulate to make this clear.

**Response: Thank you for the comment. We will re-formulate this paragraph for clarity, taking note of specifying what is meant by "scattering interfaces" and "tracking horizons", and the assumptions linking them.**

*Final response: The sentence now reads:*

*"Here, ALS and MAX at Ku-band (and Ka-band) was on average underestimated by at least 0.15 m compared to ALS and C/S MAX, and for several instances, Ka- and Ku-band reflected either within the snowpack or contributions from different interfaces and attenuation of the signal limits the full extent of penetration. Based on this, it appears that Ka- and Ku-band's maximum is retrieved from mid-way in the snow and upwards in the majority of times."*

General comments

*Exploring microwave penetration into snow on Antarctic summer sea ice along CryoSat-2 and ICESat-2 (CRYO2ICE) orbit from multi-frequency air- and spaceborne altimetry* by *Renée M. Fredensborg Hansen, Henriette Skourup, Eero Rinne, Arttu Jutila, Isobel R. Lawrence, Andrew Shepherd, Knut V. Høyland, Jilu Li, Fernando Rodriguez-Morales, Sebastian B. Simonsen, Jeremy Wilkinson, Gaelle Veyssiere, Donghui Yi, René Forsberg, and Taniâ G. D. Casal* presents a multi-scale examination of microwave penetration into the snow cover on Antarctic sea ice considering different viewing geometries from airborne and satellite platforms; namely CRYO2ICE and an airborne Ka-, Ku- and C/S-band radar and complimentary NIR lidar at 904 nm, the latter of which flew as an under-flight to the near coincident IS2 and CS2 overpasses. The paper is very well written and organized, albeit quite lengthy which is no doubt attributable to the many different sensors used. The authors have done a remarkable job at processing the various datasets and integrating them together and for this they should be commended. From a remote sensing of sea ice perspective, it is truly a globally unique dataset and for this reason alone I recommend that the manuscript be published, primarily for the intercomparison aspect of datasets and 16 different combinations of retrieved surfaces alone, subject to addressing my major and minor concerns and comments.

**Response: We thank the reviewer for the positive feedback!**

Major Concern

Not surprisingly, my major concern centers of the lack of in-situ snow and sea ice physical property validation data to support the altimetric interpretations. As the manuscript alludes to, there are several salient environmental conditions that need to be considered when interpreting airborne and satellite multi-frequency microwave data of snow (which is likely to be complex in nature) covered seasonal sea ice. Fig's 2a and b suggest that the Dec 13th 2m air temperatures (from ERA5) ranged between +1 and -3oC (mean ~ -2.5oC) at the time of CRYO2ICE overpasses and CRYO2ICCEANT22 airborne flight. I note that UTC times listed (L186-189) are 3 hours difference from Rothera meaning that the CRYO2ICE overpasses and CRYO2ICCEANT22 airborne flight occurred during the warmer part of the mid-afternoon. As such, we might infer surface air temperatures were on the warmer side of the +0.5 to -3oC range despite the known warm air temperature bias in ERA5 often reported in the literature (~2-3oC) e.g., https://doi.org/10.3390/rs13142813. Notwithstanding, it is likely safe to assume that the snow volume and snow-ice interface temperatures will be slightly warmer than the estimated surface air temperature (range), given the expected thin sea ice resulting from this warm Antarctica winter. I used data from https://www.timeanddate.com/weather/antarctica/rothera-research-station/historic?month=11&year=2022 from March 1 to Dec 13, including precip/snowfall data to crudely calculate sea ice thickness to be no greater than 80-100 cm (albeit at Rothera which I acknowledge is some distance away from your study site) using a simple 1-D

thermodynamic model. For thicker snow cover regions in the northern two-thirds of your study region it is likely that sea ice thickness is no greater than 80 cm. Such warm snow temperatures are associated with elevated snow brine volumes owing to the brine wicking mechanism that you mention several times (Massom et al., 2001; Barber and Nghiem, 1999; Barber et al., 1995; Geldsetzer et al., 2009). Furthermore, there is no information on the expected mean ice thickness on Dec 13th for your study region. A crude calculation for locations with snow depths > 30 cm (assuming a bulk snow density of 300 kg/m3 (as you suggest on L449) and 80 cm thick sea ice would produce negative freeboards at many? most?? locations along the coincident CRYO2ICE/airborne line. As such, your thick snow-covered areas likely include significant brine wetted snow and/or flooded sea ice in the basal layer with or without snow-ice and your thin snow-covered sea ice likely has melt-freeze layers, including surface melt-freeze crusts, and likely enlarged snow grains in the basal layer and greater snow roughness interfaces (at the surface and within the volume).

**Response: We thank the reviewer for the in-depth comment and for the initiation of this important discussion regarding flooded and/or brine-wetted snow, and the impact it has on our altimetric studies.**

**Indeed, there is a significant possibility of negative freeboards along the entire transect, affecting penetration at different frequencies due to various aspects as you mention including enlarged grain sizes, wetness, crusts, layers, snow-ice formation, and flooded layers, which has been brought up several times in the paper already, and which is not news when it comes to Antarctic sea ice. This is also evident in Figure 7a (S/C-band subplot), where the tracked snow-ice interfaces are either very close to or even below surrounding leads. However, it appears we were not sufficiently clear about the various aspects that could limit the penetration. We will elaborate more on these effects on radar penetration in the discussion Section 5.1.**

**However, we prefer not to speculate  too much on the ice thickness in the region, which is why it was not mentioned in detail in the paper. Additionally, calculating an estimate using a thermodynamic model and surface air temperatures from Rothera would likely be inaccurate, as these air temperatures do not reliably represent the surface air temperatures over the sea ice in our study region (see also Figure 2). Ideally, the thermal instrument onboard the aircraft could provide some insights, however these data are still being processed.**

Therefore, my concern and likely that of yourself and co-authors, is that your evaluation of microwave penetration depth estimates are not all that meaningful when they cannot be assessed using any in situ snow physical property data and corresponding estimates of dielectrics, surface roughness, etc. … especially during relatively warm conditions, and particularly for thick snow producing sea ice which is not likely to be thick enough to continuously produce positive freeboards along the entire transect. L666-669 and L747 suggest as much. While your ALS swath and optical images allow you to 'geophysically' associate what the altimetry is *sensing* (i.e., leads versus floes including ridges), you

have no information on the thermodynamic, snow structure, and dielectric states. As a result, I feel like nearly half of these 16 different combinations of retrieved surfaces is just as likely to represent the truth as any other. In other words, which of Fig 10a (or Fig 11a) is closer to the truth?

**Response: Yes, we agree that it is difficult to say exactly what is going on and what "combination" is the truth, since we do not know the physical properties of the snow with the lack of in situ data. But, as you state in the end of the this paragraph, nearly half of these combinations would likely represent the truth as any other which leads into a discussion of what is the true snow depth retrievable when using altimetry due to the impact that the dielectric properties of the snow might have, that can limit penetration and alter scattering surfaces? And, one of the main questions of this study concerns "to what extent can we extract the same snow depth - under these study site conditions - using the various combinations relevant for the future dual-freq missions and airborne campaigns".**

**Now, as we state, the plan is to use the snow radar as the ground-truth, as this is what has been done in multiple satellite studies, however what we also show is that the airborne combinations and re-trackers can provide vastly different results. Hence, more work is required to ensure consistency between methods and instruments. One way to ensure that we then track what we think we are tracking, is of course to have in situ data provided these are collected in a way that correlates with what is observed by the airborne sensors and the satellite sensors.**

As such, the analysis presented in this manuscript is likely better reflected in a title change (by removing the notion that you are able to 'explore' microwave penetration depth) to one which emphasizes the intercomparisons of CRYO2ICE and CRYO2ICCEANT22 airborne altimetry data on an unknown but likely heterogenous and complex surface with unknown thermophysical snow structure. My suggestion would be 'An intercomparison of CryoSat-2 and ICESat-2 (CRYO2ICE) and airborne multi-frequency altimetry over heterogeneous snow-covered summer sea ice without surface validation data'.

**Response: We thank the reviewer for the detailed comment, and agree that all the aspects mentioned indeed means that we cannot - with certainty - say which combination is "true", hence also the wording "Explore" used in the title. However, we are happy to include the aspects mentioned by the Reviewer and adapt the title accordingly. We have changed the title slightly from the suggested to be:**

**"An intercomparison of CryoSat-2 and ICESat-2 (CRYO2ICE) and airborne multi-frequency altimetry over heterogeneous snow-covered Antarctic summer sea ice"**

**We have however opted to not include the "without surface validation data" since it is reflected in the title already, that airborne and satellite data will be used. Also,**

it is critical to not take away from the importance of having these airborne observations themselves even without in situ data, which we feel the negative phrasing of "without surface validation data" alludes to.

We also note that this suggested title could be altered to highlight the "Part I" and "Part II" suggested as an alternative solution to deal with the otherwise lengthy manuscript, as noted in the response to Reviewer #1.

*Final response: The manuscript has now been split into two parts, representing the two objectives of the first version of the manuscript, and the titles have been edited accordingly. We have adapted the titles slightly from the proposed version, but kept the majority of the proposed title with the inclusion of "over heterogeneous snow-covered Antarctic summer sea ice", but rather specify for each part what the focus is.*

Less Major Concern

The length of the manuscript is somewhat of an issue. In order for comparison of the airborne sensors with the CRYO2ICE data, no less than seven instruments and their individual processing chains dealing with calibration, pre- and post-processing, including inter-calibration of sensors between floes and leads, etc. As you say on L552, 16 different combinations of retrieved surfaces are presented in this paper, and it is challenging to trace all the different considerations. A somewhat dramatic suggestion would be to split the paper into two manuscripts. One focused on the more technical aspects of the processing of the 16 different combinations of retrieved surfaces and much of the Results sections (to a Remote Sensing journal) and the other as a literature review type paper on multi-frequency and multi-scale radar altimetry of snow-covered summer sea ice which would include all of Section 5 (Discussion and outlook) and other pertinent material.

**Response: We thank the reviewer for the comment. While we are keen to keep the manuscript combined, we acknowledge that it is lengthy. We have suggested to focus the manuscript more, however most of the information included is pertinent to understanding the data and the processing steps (as also noted by the Reviewer). Another solution has been suggested, in response to Reviewer #1, which is to separate the manuscript into two parts, but still keeping them linked. We await reviewers and editor comments regarding this.**

*Final response: The manuscript has been separated into two.*

Detailed minor comments

Line 35 – multiplex? … did you mean to say 'complex'?

**Response: We thank the reviewer for the comment. Indeed, complex was the intended word.**

Line 48 – I feel you are missing a key ref here wrt to microwave propagation speed in snow on sea ice (Mallett et al., 2020 TC) which has been cited 47 times at last count.

**Response: Indeed, an oversight on our part. Has now been included as reference in the following:**

**"Based on these experiments, Ku-band signals is often assumed to penetrate the snow cover, providing radar freeboards which are directly converted to sea ice freeboard after accounting for the slower wave propagation speed (e.g., Hendricks, 2022; Rinne and Hendricks, 2023; Mallett et al., 2020)".**

Line 71 – '… Rösel et al., 2021), on Ku-band radars combined … ' I think it should be 'or'.

**Response: We thank the reviewer for the comment. However, we do not agree, since this question is noting three "types" of analyses or comparisons made. To clarify this, we have changed the sentence to:**

**"Most airborne studies have so far focused on (a) the Arctic snow depth using snow-radars or combinations of snow-radar and lidar (e.g., Kurtz and Farrell, 2011; Newman et al., 2014; Rösel et al., 2021); (b) Ku-band radars combined with either ground-based reference observations or airborne lidars(e.g., Willatt et al., 2011; De Rijke-Thomas et al., 2023; King et al., 2018); or, (c) deriving snow depth from snow-radar, Ka- and/or Ku-band for the purpose of validating satellite observations (e.g., Garnier et al., 2021; Armitage and Ridout, 2015)."**

Line 115-116 – This sentence reads awkwardly. Please revise.

**Response: We thank the reviewer for the comment. The sentence initially read:**

**"Currently, methods for this snow-radar have been developed and tested for the Arctic (Jutila et al., 2022b) and one Antarctic campaign with a sea-ice component has been conducted."**

**It now reads:**

**"Currently, a method to derive snow depth from this particular snow-radar has been developed, however only tested for the Arctic (Jutila et al., 2022b). One Antarctic campaign with a sea-ice component has been conducted, but further analysis of the applicability of the methodology is pending."**

Line 169-170 – While 904 nm waves can generally be assumed to reflect from surfaces, they are likely to penetrate several cm under optimal cold, low density surface snow (Deems et al., 2013 S1.4). Most certainly a 532 nm laser reflects almost entirely at the a-s interface but that's not the case here. I think a sentence is warranted to indicate that under such conditions, 904 nm waves are likely to penetrate and/or attenuate and/or scatter within the first few centimeters, unless you know definitively otherwise that none of your surface along the C2I and airborne flight lines did not have this condition (which I

suspect is > 95% of the time for this study). I realize that in the grand scheme of things 1 or 2 cm is not going to significantly influence your results here.

**Response: We thank the reviewer for the comment. Indeed, some centimeter-level penetration can occur, and we will include a sentence to indicate this. However, as you pointed out, we do not assume that this will be the most impactful effect in the broader context.**

*Final response: the following sentence has been added in Section 2.1.2 (ALS data).*

*"While NIR (904 nm) waves are generally assumed to reflect from first-encountered surfaces, they are likely to penetrate several centimetres under optimal cold, low density surface snow \citep{deems2013}. However, we assume under these conditions such potential penetration to be negligible. "*

Line 265 – …" This, in turn, leads to reduced emissivity at higher frequencies, …"

**Response: Agreed, has been edited accordingly.**

Section 3.1.2 and Figure 3 (C/S-band plot). So, after reading all of the detail in 3.1.2 on multiple scattering horizons, as it relates to Figure 3 C/S-band example waveform plot, I am still confused as to why the C/S CWT a-s interface is detected ~ 12 cm higher up than the a-s interface from the ALS (Fig 3 – right panel)? I do note that in Fig 3 caption Deltaz,C/S is potentially offset by 2 of these 12 cm. Is it because Jutila et al., 2002b developed PEAK on Arctic snow or it just an artifact (interpolation) of the C/S band processing? I suspect I just shouldn't be that concerned about it given the quite large calibration offsets needed (3.2.2). Perhaps the example waveform shown in Fig 3 consists of leads and/or open water thus your explanation L386-390 is useful. Furthermore, the large 'grey shading' of local waveforms to the example waveform shown here (Figure 3) suggests vastly different snow conditions and surface/volume scattering mechanisms over short distances further supporting my Major Concern comment of highly variable and/or complex snow, which can't be commented on due to the lack of in-situ data.

**Response: Indeed, a potential small offset by the lever arms is present (of those 2 cm as noted by the Reviewer), but the reason that the air-snow interface is tracked that much higher in CWT compared to PEAK relates to how they define the air-snow interface to be tracked.**

**PEAK re-tracks the air-snow interface once it is above the user-defined power threshold for noise, and the left-hand peakiness threshold (in a logarithmic scale). That means that PEAK re-tracks the air-snow interface on the leading edge of the maximum value whenever the thresholds are met. However, the snow-ice interface on PEAK can be on either the leading edge or any subsequent peaks provided they fulfill the threshold requirements. In contrast, CWT uses the first range bin where the power exceeds a noise threshold, which is essentially when the radar**

beam starts to illuminate the snowpack (which could include e.g., snow drifts, sastrugi etc.). Thus, PEAK is more strict in selecting on which part of the leading edge the air-snow interface should occur assuming that it has to be of a significant magnitude to represent scattering from an interface, where there is potential for CWT to re-track lower power spikes or power in general on the waveform as its air-snow interface. Furthermore, CWT always picks the snow-ice interface on the leading edge. All of these differences in "interface" tracking can be seen on Figure 3 (C/S-band), where PEAK tracks primarily on the leading edge of a significant peak, whereas CWT re-tracks primarily once the waveform exceeds some "thermal noise" or where it has some power, which is visible. ALS is the average surface value within a 5 m circular footprint at the radar observation locations, and as noted, could have some minor penetration which is however within uncertainties of the radars and their range resolution.

It is important to note, that the specific example waveform shown in Figure 3 is not of leads (you can see in Figure 7b, that the example waveform is from a relatively large floe, and from Figure 6 you can deduce the scale of what is included in the waveform of airborne scales).

However, yes, the shading shows the heterogeneity of the snow and ice conditions along a subset. Here, it is only +/- 5 nearest waveforms (which essentially covers, approximately, 50 m of distance).

L486-488 – Its worth reiterating again here that you are most certainly not dealing with a cold and dry snowpack (re: Major Concern). Data from your ERA5 air temperature map (Figure 2) and the Rothera station data https://www.timeanddate.com/weather/antarctica/rothera-research-station/historic?month=11&year=2022 suggest considerable diurnal snow temperature cycling in your study area during the latter half of November which has likely produced a complexly layered snow pack with multiple melt/freeze layers, including large poly-aggregate snow grains, which when brine coated from brine wicking, serve as significant scattering centers for Ku-band waves.

Response: Indeed, we will reiterate this, and link to Section 5.1 where this is further discussed.

*Final response: Has been reiterated by inclusion of the following:*

*"We note, that these conditions are not necessarily met during this campaign with the relatively warm weather conditions. This is further discussed in Section 5."*

L616 – I might suggest you use (or at least add) the word 'differences' instead of 'variations'.

Response: We thank the reviewer for the comment, and have changed the word "variations" to "differences.

L634 – on_space(

**Response: Thank you, adapted accordingly.**

L710 – I suggest using 'best' instead of 'greatest'

**Response: Thank you, adapted accordingly.**

L730 – I would prefer that the sentence be adjusted to state what the 'brine wicking' process does, that is, leads to saline snow in the basal layer. Suggestion would be …" brine wicking *leading to saline snow*, and/or flooding due to …"

**Response: We agree, this has been adapted accordingly.**

L728 – "… appear not to hold true …'

**Response: Thank you, adapted accordingly.**

L735-738– while liquid water may in fact be present in the snow along the joint CRYO2ICE overpasses and CRYO2ICCEANT22 airborne flight lines during the diurnal maximum on Dec 13th, the likely more important factor (related to Major Concern above) is the appreciable brine volume and/or flooded snow-ice interface from negative freeboards. To conclude on the air temperature of ~ -5oC (L736) is not sufficient. You need to translate what this air temperature forcing would produce in terms of snow volume and snow-ice interface temperature given the expected ice thickness and assuming the densities of snow, sea ice, and ocean water. It will/would be quite interesting to see what the surface temperature might have been like from the thermal instrument (not yet completely processed or included in this paper).

**Response: Indeed, it would have been interesting to see that the surface temperature would have been. While we are not keen on providing exact estimates of expected thickness when we see such discrepancies in derived snow depths from these different combinations, we will expand on the discussion that relatively warm temperatures will have on the dielectric and physical properties of the snowpack. Provided the thermal data has been processed, this could be included to support the discussion.**

*Final response: The thermal data has not yet been processed. However, we have included a longer discussion on impact of brine-wetted snow and impact of warmer temperatures in Section 5 with following examples:*

*"Figure 2a shows relatively warm temperatures in the latter part of November, which can have produced a complexly layered snow pack with multiple melt and freeze layers as well as large snow grains. This could further limit penetration, especially at higher frequencies."*

*Furthermore, a longer paragraph has been included in the discussion section on impact of brine-wetted snow.*

L749-752 – I am surprised that the authors have chosen to reference a multi-frequency snow wetness study over land (Marshall et al., 2004) when there are several multi-frequency microwave snow on sea ice papers available that discuss microwave penetration issues into saline snow when snow temperatures are warm enough to contain appreciable amounts of brine in the basal layer and nearly warm enough to possess small quantities of liquid water in the snow (e.g., Willatt et al., 2010; 2011 and https://doi.org/10.1016/j.rse.2017.06.029).

**Response: We thank the reviewer for the comment. This aspect was mainly included to discuss to what extent C-band can have dominant scattering at the snow-ice interface even with a relatively wet snowpack - a focus that Willatt et al. 2010, 2011 primarily address for Ku-band. Nandan et al. (2017, J.RSE - linked by Reviewer) evaluates both Ku-, X-, and C-band, however, only for scatterometers. We will include Nandan et al. (2017) as reference if relevant once we have read it in detail. We do note that we have already included several other altimeter studies over sea at different frequencies discussing partial penetration (including Willatt et al 2010, 2011) and the various factors contributing to it. However, as noted before, we will expand on the discussion of brine-wetted snow and include the relevant sea ice altimeter studies.**

*Final response: The manuscript has been included in the discussion on brine-wetted snow. Furthermore, the mentioned paragraph for Marshall et al. (2004) has been updated to the following to specify relevance to this study:*

*"However, Marshall et al (20049) showed for ground-based observations over terrestrial snow that C-band was able to reflect at ground reflections in wet snow whereas Ku-band was quickly attenuated"*

L774 '…. occurs …'

**Response: Thank you, adapted accordingly.**

L943-944 – I'm surprised that the authors don't recommend that such investigations (and science planning efforts) make surface validation data a priority. I get that it would have been nearly impossible and/or costs hundreds of thousands of $ to acquire even a handful of surface validation points but its important to emphasize this point for funding agencies when such good analytical work, like is the case for this paper, is completed.

**Response: We thank the reviewer for the comment, which Reviewer #1 also noted. We will make sure to note such investigations and efforts as priority, however, there is still work needed to understand how such observations should be completed to simulate what is observed by air- and spaceborne platforms. Simply doing transects may not be sufficient. That alone has been discussed when comparing freeboards from air- and spaceborne systems in e.g., the CryoVal-SI project. We will include this aspect in the discussion section and urge for continuation of such science planning efforts.**

*Final response: The importance of coincident in situ observations and ground-based field campaign planning efforts has been noted in both discussion and conclusion for both Part I and Part II.*

---

## Referee Report (RR1)

I think the authors have made a wise decision splitting the original manuscript into two parts. The two-part manuscript now reads succinctly and fluidly. The authors have done an exceptional job at reorganizing this work. It makes good sense that Part 2 is a slightly shorter paper than Part 1 because the authors can reference Part 1 frequently, particularly wrt to some introduction, study site and some methods, towards reducing redundancy. I have only a few technical issues and comments for consideration.

Part 1

A really small technical thing. Figure 5 caption and legend have slightly different terms. In the legend which appears above the figure, the delta (ha - i) term is different than the delta (ha-s) term. I believe the 'i' in the legend for this expression should be an 's'. Also, I don't see the need to extend the x-axis range out to 800 kg m-3. Why not just extend to ~ 550 or 600 and perhaps start at 150 instead of 100. Then your inset plot of the 280-380 range would expand in size a bit, which would be nice.

Figure 6b (axis above figure). Should the along-track distance be in km? (not meters). 0.07 degrees latitude corresponds to 7.77km so going from slightly less than 603 to slightly larger than 611 km would make sense.

L521 - ....'we note the presence of liquid **water** in the snowpack would likely ..."

L522 - ....'which **could** have produced ...'

L541-555. I appreciate the addition of this section to explain the potential dielectric and scattering complexities which can arise at these temperatures. However, I suggest replacing Barber et al., 2003 reference on L545 with Barber et al., 1998 The role of snow on microwave emission and scattering over first-year sea ice | IEEE Journals & Magazine | IEEE Xplore as the better place to find the description of how brine is expelled upward into the basal snow layer. Furthermore, a recent assessment of the process is described here https://doi.org/10.1017/aog.2024.27 and would be worth citing behind Barber et al., 1998.

L548 – "Importantly, **from a** remote sensing perspective ..."

L650 – 'This is expected caused ...." reads awkwardly. Please revise.

L654 – I suggest revising the sentence 'Several inconsistencies are observed between the a-s interface determined by ALS and the radars likely caused by ... " to "Several inconsistencies are observed **for** the a-s interface determined by ALS and the radars **and are** likely caused by ..."

Part 2

Here, I have two comments for consideration.

1) In Part 2 you make the comparison of your CRYO2ICE derived snow depth to AMSR2, but you have left the AMSR2 background image in Figure 1 of Part 1 (because this was one big manuscript initially and you have decided to not alter the figure ... I wouldn't either ... it's a really nice figure). So, for all the AMSR2 discussion in Part 2, we are left to viewing a few AMSR2 points on Figure 2 Part 2. As a result, your CRYO2ICE tracks in Figure 1 of Part 2 lack a bit of spatial context to the broader AMSR2 estimates, especially if you are not going to refer to Figure 1 Part 1 in all of the CRYO2ICE to AMSR2 comparison discussion in section 4.2 of Part 2. So, I guess what I'm trying to say is, can you figure out a way of presenting Figure 1 from Part 1 to show the AMRS2 background image needed for your discussion of AMSR2 in Part 2 and for the larger spatial context of your CRYO2ICE tracks shown in Figure 1 of Part 2. I get that you can't show Figure 1 from Part 1 in both papers, but can you possibly integrate a sub-image of the AMSR2 background image shown in Figure 1 Part 1 into Figure 2 Part 2

so that AMSR2 image data presented in Figure 1 Part 1 doesn't go to waste, especially if you are not going to refer to it in section 4.2 Part 2?

2) So, brine wicking and its potential for altering dielectrics and scattering was thoroughly discussed in Part 1 (L541-555) and it is introduced in the Introduction of Part 2, but it is not mentioned again in the rest of the manuscript (Part 2). In additional to ice surface roughness, the brine wicking process and basal snow layer brine volume affects the snow dielectrics, and it affects Ka- and Ku-band scattering and attenuation as you rightfully acknowledge in Part 1. As such, I would have expected some additional discussion of its potential affect on CRYO2ICE penetration depth, scattering and attenuation. The fact that the Ka- and Ku-band sensors are on satellites (Part 2) as opposed to aircraft (Part 1) doesn't change the implications for dielectrics and microwave penetration, scattering and attenuation. I strongly suggest you add some additional commentary in either section 4.3 or 5. Or, make stronger reference in Part 2 back to your description in Part 1 (L541-55)

Finally, as an overall comment on both Parts, I agree that there appear to be MAUP (Modifiable Areal Unit Problem) issues at play (L419 of Part 2).
Openshaw https://www.uio.no/studier/emner/sv/iss/SGO9010/openshaw1983.pdf is a good reference for trying to understand the problem wrt measuring snow covered sea ice from various microwave sensor characteristics and ground resolutions/footprints as a function of height above the surface. These scale issues are no doubt a result of different altimeter processing techniques as a function of frequency but some which are likely as result of MAUP as a function of the length scales of snow thickness distributions and ice surface properties particularly wrt to ice surface roughness resulting from heterogeneous ice freezing/consolidation processes and dynamics+deformation, etc. I would be curious to hear the authors thoughts on how MAUP can be overcome (or at least minimized) as the sea ice community continues to use both airborne and surface-based multi-frequency microwave measurements towards either calibrating and validating satellite-based estimates.

---

## Author Response (AR2)

**Editor comment**

Your paper has received two reviews based on your revisions. The second reviewer has raised some concerns on how the discussion section has been structured in a speculative manner owing to the lack of in situ data for validating snow depths on Antarctic sea ice. While I understand your limitations and challenges to justify your discussions, I feel the second reviewer has their rationale related to how the different frequencies behave in terms of its penetration on a diverse range of snow covers especially during the summer season where snow undergoes geophysical and thermodynamic changes drastically causing snow covers to become even more complex and complicated to be tracked. I suggest you go through the reviewer comments and think about how the speculatory discussion can be reduced (I understand it cannot be avoided in this context). For that reason, I recommend major revision for this round of review.

**Response to editor and reviewers**

Reviewer comments are provided in bold, and responses to reviewer comments are in *italic blue*.

*We thank the editor and reviewer for their valuable comments and feedback. Indeed, the summer season noticeably changes the physical and thermodynamic conditions, however, the summer impact is less noticeable for Antarctic snow cover compared to the Arctic, where we can have snow cover surviving the melt season, although with a complex structure, and studies of the Antarctic snow cover using satellites have been applied for the entire year (for passive microwave radiometry, but also from altimetry using the same methodology). Thus, while we acknowledge that the impact of the summer season cannot be neglected, and discuss this more in the updated version, the results themselves speak for the possibility of extracting snow depth (or at least that some microwave penetration occur and appears to reflect at some clear inter-faces) from some of the radars during this period. And overall, this manuscript presents a unique collection of data sources inter-compared at different scales, which has not been published before.*

*We have taken reviewer and editor comments into consideration, and have made the following amendments to the papers (Part I and Part II):*
- *Updated figures (updated Fig. 5 in Part I and included a new figure in Part II) and text based on Reviewer #1 comments.*
- *Inclusion of new references either suggested by Reviewer #1 (Barber et al. 1998; Mallett et al. 2024), or to support new additions (e.g., Jutila et al., 2025; Farrell et al., 2012).*
- *Included a short paragraph on the impact of snow conditions and instrument specifications for satellite observations in Part II, which was missing (Reviewer #1 suggestion), with a strong reference to the snow conditions discussion already included in Part I.*
- *Updated the discussion (in response to Reviewer #3) to be less speculative where we saw it needed.*

*We hope that these amendments are to your satisfaction. We look forward to your response.*

**Reviewer #2: John Yackel**

I think the authors have made a wise decision splitting the original manuscript into two parts. The two-part manuscript now reads succinctly and fluidly. The authors have done an exceptional job at reorganizing this work. It makes good sense that Part 2 is a slightly shorter paper than Part 1 because the authors can reference Part 1 frequently, particularly wrt to some introduction, study site and some methods, towards reducing redundancy. I have only a few technical issues and comments for consideration.

*We thank the reviewer for their efforts in reviewing our manuscript, the positive feedback, and the great suggestions. Please find a detailed response to each point below.*

Part 1
A really small technical thing. Figure 5 caption and legend have slightly different terms. In the legend which appears above the figure, the delta (ha - i) term is different than the delta (ha-s) term. I believe the 'i' in the legend for this expression should be an 's'. Also, I don't see the need to extend the x-axis range out to 800 kg m-3. Why not just extend to ~ 550 or 600 and perhaps start at 150 instead of 100. Then your inset plot of the 280-380 range would expand in size a bit, which would be nice.

*We thank the reviewer for the comment and the keen eye. We will correct the Figure 5 caption and legend. We have originally expanded to 100-800 based on papers which showed examples of this variety of densities. Based on the reviewer's comment, we will change the x-axis from 150-600 kg m-3 instead and expand the inset plot, as shown below.*

[Figure]

Figure 6b (axis above figure). Should the along-track distance be in km? (not meters). 0.07 degrees latitude corresponds to 7.77km so going from slightly less than 603 to slightly larger than 611 km would make sense.

*Corrected, thanks.*

L521 - ….'we note the presence of liquid **water** in the snowpack would likely …"

*Corrected, thanks.*

L522 - ….'which **could** have produced …'

*Corrected, thanks.*

L541-555. I appreciate the addition of this section to explain the potential dielectric and scattering complexities which can arise at these temperatures. However, I suggest replacing Barber et al., 2003 reference on L545 with Barber et al., 1998 The role of snow on microwave emission and scattering over first-year sea ice | IEEE Journals & Magazine | IEEE Xplore as the better place to find the description of how brine is expelled upward into the basal snow layer. Furthermore, a recent assessment of the process is described here https://doi.org/10.1017/aog.2024.27 and would be worth citing behind Barber et al., 1998.

*We thank the reviewer for suggesting these references, which have now been included in the paper.*

L548 – "Importantly, **from a** remote sensing perspective …"

*Corrected, thanks.*

L650 – 'This is expected caused ….'' reads awkwardly. Please revise.

*Corrected to:*

*"This is expected since PEAK can identify peaks before and after MAX as the s-i interface provided the peak fulfils the requirements"*

L654 – I suggest revising the sentence 'Several inconsistencies are observed between the a-s interface determined by ALS and the radars likely caused by … " to "Several inconsistencies are observed for the as interface determined by ALS and the radars and are likely caused by …"

*Corrected, thanks.*

Part 2
Here, I have two comments for consideration.

1) In Part 2 you make the comparison of your CRYO2ICE derived snow depth to AMSR2, but you have left the AMSR2 background image in Figure 1 of Part 1 (because this was one big manuscript initially and you have decided to not alter the figure … I wouldn't either … it's a really nice figure). So, for all the AMSR2 discussion in Part 2, we are left to viewing a few AMSR2 points on Figure 2 Part 2. As a result, your CRYO2ICE tracks in Figure 1 of Part 2 lack a bit of spatial context to the broader AMSR2 estimates, especially if you are not going to refer to Figure 1 Part 1 in all of the CRYO2ICE to AMSR2 comparison discussion in section 4.2 of Part 2. So, I guess what I'm trying to say is, can you figure out a way of presenting Figure 1 from

Part 1 to show the AMRS2 background image needed for your discussion of AMSR2 in Part 2 and for the larger spatial context of your CRYO2ICE tracks shown in Figure 1 of Part 2. I get that you can't show Figure 1 from Part 1 in both papers, but can you possibly integrate a sub-image of the AMSR2 background image shown in Figure 1 Part 1 into Figure 2 Part 2so that AMSR2 image data presented in Figure 1 Part 1 doesn't go to waste, especially if you are not going to refer to it in section 4.2 Part 2?

*We thank the reviewer for the comment. We include a reference to Figure 1 in Part I for a zoom in, and include the following figure of both products from 13 December 2022 showing the pan-Antarctic snow depth distribution in Part II.*

[Figure]

2) So, brine wicking and its potential for altering dielectrics and scattering was thoroughly discussed in Part 1 (L541-555) and it is introduced in the Introduction of Part 2, but it is not mentioned again in the rest of the manuscript (Part 2). In additional to ice surface roughness, the brine wicking process and basal snow layer brine volume affects the snow dielectrics, and it affects Ka- and Ku-band scattering and attenuation as you rightfully acknowledge in Part 1. As such, I would have expected some additional discussion of its potential affect on CRYO2ICE penetration depth, scattering and attenuation. The fact that the Ka- and Ku-band sensors are on satellites (Part 2) as opposed to aircraft (Part 1) doesn't change the implications for dielectrics and microwave penetration, scattering and attenuation. I strongly suggest you add some additional commentary in either section 4.3 or 5. Or, make stronger reference in Part 2 back to your description in Part 1 (L541-55)

*We thank the reviewer for the comment. Indeed, it does not make a difference whether the sensors are on an aircraft or a satellite in terms of the dielectrics and microwave penetration as such, however, the bandwidth and imaging geometries will govern what is detectable in the data as well as the central frequency of the instruments, and cannot be neglected - hence the impact of scale. Furthermore, for the satellites, we do not have a Ka-band sensor, but the photon-counting laser*

*altimeter instead, which has an even smaller footprint than the Ka-band satellite radars.*

*We have now included the following in the discussion (Sect. 5) of Part II:*

*"Similarly to the airborne data (Part I, Section 5), the snow conditions also matter to the satellite observations, and the retracked scattering horizons are directly related to the instrument specifications of the spaceborne altimeters, as with the airborne. Hence, any snow conditions (e.g., saline snow, icy layers, snow grains, etc.) limiting penetration at airborne scales are also expected to be impactful on the satellite scales, although at different magnitudes, which one may assume when the footprints are larger and the resolution lower. "*

Finally, as an overall comment on both Parts, I agree that there appear to be MAUP (Modifiable Areal Unit Problem) issues at play (L419 of Part 2). Openshaw https://www.uio.no/studier/emner/sv/iss/SGO9010/openshaw1983.pdf is a good reference for trying to understand the problem wrt measuring snow covered sea ice from various microwave sensor characteristics and ground resolutions/footprints as a function of height above the surface. These scale issues are no doubt a result of different altimeter processing techniques as a function of frequency but some which are likely as result of MAUP as a function of the length scales of snow thickness distributions and ice surface properties particularly wrt to ice surface roughness resulting from heterogeneous ice freezing/consolidation processes and dynamics+deformation, etc. I would be curious to hear the authors thoughts on how MAUP can be overcome (or at least minimized) as the sea ice community continues to use both airborne and surface-based multi-frequency microwave measurements towards either calibrating and validating satellite-based estimates.

*We thank the reviewer for the comment. Indeed, MAUP is at play in terms of different spatial distribution of datasets and the way they are aggregated (e.g., going from airborne to satellite comparisons, to comparison with gridded data). However, we would argue that some of the most important aspects relates to the sensor specifications and the surface characteristics retrievable using that specific sensor (so not only as a function of frequency, but as a function of beam width, range resolution, footprint and such). Thus, with future work, we hope to make some sensitivity studies to evaluate the impact of this on the scattering interfaces considering not only the frequency, but also change of resolution, area size (altimeter processing techniques - e.g., low-resolution-mode vs SAR), altitude (related to footprint) and such. In addition, further discussions on how field work could (or maybe "should") be performed to emulate what the airborne and satellite sensors will see - or how we can at least try to reach something comparbale, so that we can actually "validate" them under the same conditions. Some projects are preparing simulators to evaluate different validation scenarios, but we do believe that it will likely not be overcomed (or at least minimised) without a strong collaborative interdisciplinary effort including modelling of ground, airborne, and satellite observation and further exploration of the data we already have, to see if there are things we have overlooked in the process.*

**Reviewer #3**

The paper 'Multi-frequency altimetry snow depth estimates over heterogeneous snow-covered Antarctic summer sea ice – Part I: C/S-, Ku-, and Ka-band airborne observations & Part II: Comparing airborne estimates with near-coincident CryoSat-2 and ICESat-2 (CRYO2ICE)' by Hansen et al. has been carefully reviewed. The paper disucsses how airborne Ka-, Ku- and C/S-band radar signals interact with summer Antarctic sea ice and how snow affects the radar signal propagation affecting snow depth retrievals, which has impact on CryoSat-2, AltiKa and upcoming CRISTAL missions. Although, the paper comprehensively (well-written) describes existing literature, theory of multi-frequency radar signal propagation through snow-covered sea ice and uses unique datasets to investigate its potential to retrieve snow depth on summer Antarctic sea ice, I have some reservations for immediate publication of this paper. Here are my general comments for this round

*We thank the reviewer for their efforts in reviewing our manuscript, the feedback and suggestions. Please find a detailed response to each point below.*

a) I am happy the authors acknowledged the lack of in situ snow depth and geophysical properties data on summer sea ice, but that is itself a downfall in the discussion part (mostly sepulations through lit review) supporting their results. Lines 135-140 show their dependence on ERA-5 reanalysis data of air temperature and precipitation to discuss their findings. This inherently amplifies the uncertainty in what is seen from the results (e.g. Figure 6 showing 'probable' air/snow and snow/sea ice interfaces). I do agree sometimes when there is no in situ data, we have to rely on literature review, but the discussions are way 'too speculative', considering the assumptions of snow properties used in the methodology.

*We thank the reviewer for their comment. Indeed, it is difficult to be more than speculative when there are no in situ observations for comparison. Within the altimetry community, and especially when comparing with satellite data, the "snow radar" has been utilised as a "reference" for the snow depth (in lack of in situ observations, but with great comparison to in situ data nonetheless - although it has its own caveats, as explained in the paper), it remains the only dataset that resembles a reference (aside from the laser for air-snow interface), and it was therefore deemed a unique - and optimal set-up - to have all the radars and lidar on board to evaluate microwave penetration. However, you can argue that the snow-radar data is not sufficient on its own as a reference (even though the community generally agrees to use it as such). To limit the speculative discussion, we have critically re-assessed the text and minimised where we saw fit or referenced further to literature - since, as the reviewer also states, we do still have to rely on literature reviews when in lack of in situ data.*

*To support the comparison in lack of in situ data, we did also compare the airborne estimates and satellite estimates with both AMSR2 and CASSIS, which have both been compared with contemporaneous in situ and/or reference data, which is the next best thing when additional contemporaneous (beyond the airborne data as presented in this study) is not available for further validation. We are keen to hear what the reviewer might suggest of data otherwise to support the study.*

2) Based on 1), the authors use 300 kg/m3 as the bulk snow density for Antarctic snow cover on sea ice, although field observations and past studies show presence of melt/refreeze layers, slush, refrozen snow-ice layers and complex snow layering both during

winters and summers, not just on sea ice that are flooded, but also for positive freeboards. Especially, with daily diurnal fluctuations in meteorological conditions, presence of all these complex snow properties discussed above are common. My problem is that the assumption of a 300 kg/m3 bulk snow density may not work considering these issues, instead, I suggest authors to conduct a sensitivity analysis based on changes in snow density (of course snow density is not the only one error source).

*We thank the reviewer for this comment. Indeed, it is not the only aspect at play, but an important one. It is also why we already included a sensitivity study in Fig. 5 of Part I based on snow density, where it shows that depending on the density used (for most studies often ranging from 280 to 350 kg/m3), the snow depth will not change by more than approximately 2.5 cm for snow depths of 0.6 m (and less for thinner snowpacks of course) using the established assumptions for deriving snow depth from the differences when comparing to using the bulk density applied in our study. However, we apologise that it has not been more clearly presented in the paper, which we have now aimed to do in Sect. 4.3, Part I.*

*As Reviewer #2 previously suggested (in the first review round), other methods could be explored (e.g., what happens when there is slush, etc.), but the current community-agreed-upon method for estimating snow depth from dual-frequency (assuming dry and cold snow conditions) follows the methodology utilised in this study (and which has also been applied for year-round snow depth maps using CryoSat-2 and ICESat-2 in published studies, even with the caveats that the snowpack might not be dry and cold throughout). For now, we have utilised that method too and believe that it is out of scope to conduct radar backscatter modelling or additional sensitivity studies. However, we have planned for future work on radar backscatter modelling to support dual-frequency altimetry approaches for a variety of snow and ice conditions.*

3) Your lines 369-372 'Traditional hypotheses of the radars include Ka-band primarily scattering at the a-s interface, Ku-band at the s-i interface (over cold and dry snow), and that C/S-band should reflect at the s-i interface at maximum scattering. Furthermore, airborne Ka- and Ku-band traditionally assume one surface to primarily contribute (thus, we only track one point of the waveform). In contrast, we assume C/S-band to be influenced enough by both interfaces to re-track both.' My question on this. How do you say C/S-band 'SHOULD' reflect from the snow/sea ice interface. If you have thin (pre-melt time) snow, these long wavelengths can scatter max from the ice volume or even from ice/ocean interface. Even otherwise, if you read my previous comment on snow properties such as slush and snow-ice formations, neither Ku-, or C/S-band will penetrate from the snow/sea ice interfaces or below. Because the density inhomogeneties and salt content from these layers are stronger causing the scattering to be the strongest from these layers. This issue ties up with my initial problem of speculating results without any in situ data. Also, how did you decide the location of the snow/sea ice interface from the '---->' line in Figure 6 labelled as snow-ice interface?

*We thank the reviewer for the comment, which we agree with. In fact, in the manuscript (especially Part I), we discuss several times the impact of various snow conditions such as e.g., slush, snow-ice formation, brine, etc., which will alter the reflective horizon, but we apologise if it has not been more clear and has tried to make clearer in this revised version. It is also true that the C/S-band could reflect within the ice under specific conditions. However, C/S-band has long been used as a snow-radar system, and it has not been a*

*worry. Most of the time, the underlying ice interface is assumed to provide the maximum scattering (this assumption is also what the official OIB data product relies on). We have changed the wording to "tends to" instead, and edited parts of the discussion to include more of the considerations of the reviewer.*

*Regarding the snow-ice interface location in Figure 6: The location of the snow-ice interface is taken from comparison to the C/S-band, which is re-tracked as the snow-ice interface with all of the re-trackers (CWT, PEAK, and MAX). The caption is now rephrased to: "In the Ku-band, what appears to be a s-i interface somewhat apparent based on a qualitative guess corresponding to the location of the s-i interface observed in the C/S-band ; however, it does not represent the maximum scatter or somewhere on the leading edge of Ku-band.".*

These are my major concerns for now. I do understand and you have clearly acknowledged the lack of in situ data in the paper that inhibits what is 'really' happening. But I feel, there should be some sort of radar penetration modeling or a sensitivity analysis to atleast reduce speculative discussions in your paper. For that reason, I think, the authors should be given a chance to respond to this review and come up with a stronger analysis to rebutt the reviewer comments. For that reason, I recommend the paper to go through major revisions for this round.

*We thank the reviewer for the review, and agree (along with the other two reviewers) that in situ data would be ideal. Nonetheless, it is not very often that in situ data is available from the airborne campaigns nor in a format that makes it comparable to the satellites - however, that does not mean that the airborne itself does not have merit, and should not be evaluated - especially when such a unique set-up is available with three different radars and a lidar system. In addition, it is very rare to have collocated in situ observations along both satellite and airborne tracks due to the drift and logistics, as well as weather, meaning that we often are not able to collect the required in situ observations, even if it was planned.*

*However, including a specific sensitivity analysis of radar penetration using a radiative transfer model and complex snow pack structures is out of scope and deserves a study in itself. For these manuscripts (Part I and Part II), the sensitivity analysis based on snow density (which is the only variable in terms of methodology) is included, and future work on such radar backscattering modelling is planned by the authors.*